# Single-cell multi-omics identifies chronic inflammation as a driver of *TP53*-mutant leukemic evolution

Alba Rodriguez-Meira [1,2,23,24,25] ✉, Ruggiero Norfo[1,2,3,25], Sean Wen[1,2,4,25],
Agathe L. Chédeville[5,6,7,8,25], Haseeb Rahman[1,2], Jennifer O'Sullivan[1,2],
Guanlin Wang[1,2,4], Eleni Louka [1,2], Warren W. Kretzschmar[9,10,11],
Aimee Paterson[1,2], Charlotte Brierley[1,2,12], Jean-Edouard Martin[5,6,7,8],
Caroline Demeule[13], Matthew Bashton [14], Nikolaos Sousos [1,2],
Daniela Moralli[15], Lamia Subha Meem[15], Joana Carrelha[1], Bishan Wu[1],
Angela Hamblin[2], Helene Guermouche [16], Florence Pasquier[5,6,7,17],
Christophe Marzac[5,6,7,18], François Girodon [13,19], William Vainchenker [5,6,7],
Mark Drummond[20], Claire Harrison[21], J. Ross Chapman [22], Isabelle Plo[5,6,7],
Sten Eirik W. Jacobsen [1,9,10,11], Bethan Psaila [1,2], Supat Thongjuea[4],
Iléana Antony-Debré [5,6,7,26] ✉ & Adam J. Mead [1,2,26] ✉

Understanding the genetic and nongenetic determinants of tumor protein 53 (*TP53*)-mutation-driven clonal evolution and subsequent transformation is a crucial step toward the design of rational therapeutic strategies. Here we carry out allelic resolution single-cell multi-omic analysis of hematopoietic stem/progenitor cells (HSPCs) from patients with a myeloproliferative neoplasm who transform to *TP53*-mutant secondary acute myeloid leukemia (sAML). All patients showed dominant *TP53* 'multihit' HSPC clones at transformation, with a leukemia stem cell transcriptional signature strongly predictive of adverse outcomes in independent cohorts, across both *TP53*-mutant and wild-type (WT) AML. Through analysis of serial samples, antecedent *TP53*-heterozygous clones and in vivo perturbations, we demonstrate a hitherto unrecognized effect of chronic inflammation, which suppressed *TP53* WT HSPCs while enhancing the fitness advantage of *TP53*-mutant cells and promoted genetic evolution. Our findings will facilitate the development of risk-stratification, early detection and treatment strategies for *TP53*-mutant leukemia, and are of broad relevance to other cancer types.

Tumor protein 53 (*TP53*) is the most frequently mutated gene in human cancer, typically occurring as a multihit process with a point mutation in one allele and loss of the other wild-type (WT) allele[1,2]. *TP53* mutations are also strongly associated with copy number alterations (CNA) and structural variants, reflecting the role of p53 in the maintenance of genomic integrity[2,3]. In myeloid malignancies, the presence of a *TP53* mutation defines a distinct clinical entity[1], associated with complex CNA, lack of response to conventional therapy and dismal outcomes[2,4,5]. Understanding the mechanisms by which *TP53* mutations drive clonal evolution and disease progression is a crucial step toward

---

the development of rational strategies to diagnose, stratify, treat and potentially prevent this condition.

Myeloproliferative neoplasms (MPN) arise in hematopoietic stem cells (HSC) through the acquisition of mutations in JAK/STAT signaling pathway genes (*JAK2*, *CALR* or *MPL*), leading to aberrant proliferation of myeloid lineages[6]. Progression to secondary acute myeloid leukemia (sAML) occurs in 10–20% of MPN and is characterized by cytopenias, increased myeloid blasts, acquisition of aberrant leukemia stem cell (LSC) properties by hematopoietic stem/progenitor cells (HSPCs) and median survival of less than 1 year[7,8]. *TP53* mutations are detected in approximately 20–35% of post-MPN sAML[9–11] (collectively termed *TP53*-sAML), often in association with loss of the remaining WT allele[12] and multiple CNAs[13]. Furthermore, deletion of *Trp53* combined with JAK2 V617F mutation leads to highly penetrant myeloid leukemia in mice[11,14].

Notwithstanding the established role of *TP53* mutation in MPN transformation, *TP53*-mutant subclones are also present in 16% of chronic phase MPN (CP-MPN), and in most cases, this does not herald the development of *TP53*-sAML (ref. 15). However, little is known about the additional genetic and nongenetic determinants of clonal evolution following the acquisition of a *TP53* mutation. Resolving this question requires unraveling multiple layers of intratumoral heterogeneity, including reliable identification of the *TP53* mutation, loss of the WT allele and presence of CNA. Integrating this mutational landscape with cellular phenotype and transcriptional signatures will resolve aberrant hematopoietic differentiation and molecular properties of LSC in *TP53*-sAML. This collectively requires single-cell approaches, which combine molecular and phenotypic analysis of HSPCs with allelic-resolution mutation detection, an approach recently enabled by the TARGET-seq technology[16].

## Results

### Convergent clonal evolution in *TP53* leukemic transformation

To characterize the genetic landscape of *TP53*-sAML, we analyzed 33 *TP53*-sAML patients (Supplementary Table 1) through bulk-level targeted next-generation sequencing and single nucleotide polymorphism (SNP) array (Extended Data Fig. 1). We detected MPN-driver mutations (*JAK2* and *CALR*) in 28 patients (85%), and co-occurring myeloid driver mutations in 24 patients (73%). Multiple *TP53* mutations were present in one-third (*n* = 11) of patients, including 2 patients with 3 *TP53* mutations; 82% (18 of 22) of patients with a single *TP53* mutation showed a high variant allelic frequency (VAF) of >50%. CNAs were present in all patients analyzed, and 87% (20 of 23) had a complex karyotype (≥3 CNAs; Extended Data Fig. 1a–g). Deletion or copy-neutral loss of heterozygosity affecting the *TP53* locus (chr17p13.1) was detectable at the bulk level in 43% of patients (10 of 23; Extended Data Fig. 1b–d). Taken together, these findings support that *TP53*-sAML is associated with complex genetic intratumoral heterogeneity.

To characterize tumor phylogenies and subclonal structures, we performed TARGET-seq analysis[16], a technology that allows allelic-resolution genotyping, whole transcriptome and immunophenotypic analysis from the same single-cell, on 17517 Lin⁻CD34⁺ HSPCs from 14 *TP53*-sAML patients (Extended Data Fig. 1a), 9 age-matched healthy donors (HDs) and 8 previously published myelofibrosis (MF) patients (Fig. 1a, gating strategy shown in Extended Data Fig. 2a). HSPCs WT for all mutations analyzed were present in 10 of 14 patients (Extended Data Fig. 2b–o), providing a valuable population of cells for intrapatient comparison with mutation-positive cells[17]. In all cases, the dominant clone showed loss of WT *TP53* through the following four patterns of clonal evolution: (1) bi-allelic *TP53* mutations by acquisition of a second mutation on the other *TP53* allele, (2) hemizygous *TP53* mutations (deleted *TP53* WT allele), (3) parallel evolution with two clones harboring different *TP53* alterations and (4) a *JAK2* negative dominant clone with bi-allelic *TP53* mutations in patients with previous *JAK2*-mutant MPN[18] (Fig. 1b–e and Extended Data Fig. 2b–o). Bi-allelic mutations were confirmed by single molecule

cloning and computational analysis (Extended Data Fig. 1h–j). Integration of index-sorting data revealed that dominant *TP53* multihit clones were enriched in progenitor populations as previously described in de novo AML[19], whereas *TP53*-mutant cells were less frequent in the HSC compartment (Extended Data Fig. 3a). CNA analysis using single-cell transcriptomes showed that all *TP53* multihit clones harbored at least one highly clonally-dominant CNA, with very few *TP53*-mutant cells without evidence of a CNA (3.4 ± 1.2%) and an additional 5 of 14 (36%) patients also showing cytogenetically-distinct subclones (Fig. 1f,g and Extended Data Fig. 2p,q).

To confirm that dominant HSPC clones were functional LSCs, we established patient-derived xenografts (PDX) for two *TP53*-sAML patients (Fig. 1h). Mice developed leukemia in 27–31 weeks with high numbers of human CD34⁺ myeloid blast cells in the bone marrow (BM; Extended Data Fig. 3b–d), with a progenitor phenotype, *TP53* mutations and CNAs similar to the dominant clone from patients' primary cells (Fig. 1i and Extended Data Fig. 3e–l). In patient IF0131, a monosomy 7 subclone (Fig. 1f) preferentially expanded in PDX models (Fig. 1i). Monosomy 7 cells showed a distinct transcriptional profile with increased WNT, RAS, MAPK signaling and cell cycle associated transcription (Extended Data Fig. 3m,n). Together, these data are compatible with a fitness advantage of monosomy 7 cells, a recurrent event in *TP53*-sAML (Extended Data Fig. 1b,c), driven by activation of signaling pathways that may relate to deletion of chromosome 7 genes such as *EZH2* (ref. 20). In summary, the dominant leukemic clones in *TP53*-sAML were invariably characterized by multiple hits affecting *TP53* (multihit state), indicating strong selective pressure for complete loss of WT *TP53*, together with the gain of CNAs and complex cytogenetic evolution, with very few *TP53* multihit cells with a normal karyotype (Fig. 1j).

### Molecular signatures of *TP53*-mutant-mediated transformation

To understand the cellular and molecular framework through which *TP53* mutations drive clonal evolution, we next analyzed single-cell RNA-seq data from 10,459 *TP53*-sAML HSPCs alongside 2,056 MF and 5,002 HD HSPCs passing quality control. Force-directed graph analysis revealed separate clustering of *TP53*-mutant HSPC in comparison with healthy and MF donor cells, with a high degree of interpatient heterogeneity (Extended Data Fig. 4a) as observed in other hematopoietic malignancies[21]. This could potentially be explained by patient-specific cooperating mutations and cytogenetic alterations (Extended Data Fig. 1). TARGET-seq analysis uniquely enabled the comparison of *TP53* multihit HSPC to *TP53*-WT preleukemic stem cells ('pre-LSCs') from the same *TP53*-sAML patients as well as healthy and MF donors, to derive a specific *TP53* multihit signature including known p53-pathway genes (Extended Data Fig. 4b,c).

Integration of single-cell transcriptomes and diffusion map analysis of HSPCs from *TP53*-sAML patients showed that *TP53* multihit HSPCs clustered separately from *TP53*-WT pre-LSCs in two distinct populations with enrichment of LSC and erythroid-associated transcription, respectively (Fig. 2a and Supplementary Table 3), and a differentiation trajectory toward the erythroid-biased population (Fig. 2b), an unexpected finding given that erythroleukemia is uncommon in *TP53*-sAML (refs. 22,23). Sorted CD34⁺ *TP53*-multi-hit cells exhibited potential for erythroid differentiation in vivo and in vitro, supporting that this occurs downstream of the LSC population (Extended Data Fig. 5a–c). *TP53* multihit LSCs showed enrichment of cell cycle, inflammatory, signaling pathways and LSC-associated transcription, whereas *TP53* multihit erythroid cells were depleted of the latter (Extended Data Fig. 4d).

To further explore this erythroid-biased population, we projected *TP53* multihit cells onto a previously published HD hematopoietic hierarchy[24]. *TP53*-sAML differed from de novo AML with an enrichment into HSC and early erythroid populations, whereas de novo AML was enriched in myeloid progenitors (Fig. 2c,d)[25]. A similar enrichment was observed for *TP53* multihit cells when mapped on a Lin⁻CD34⁺ MF

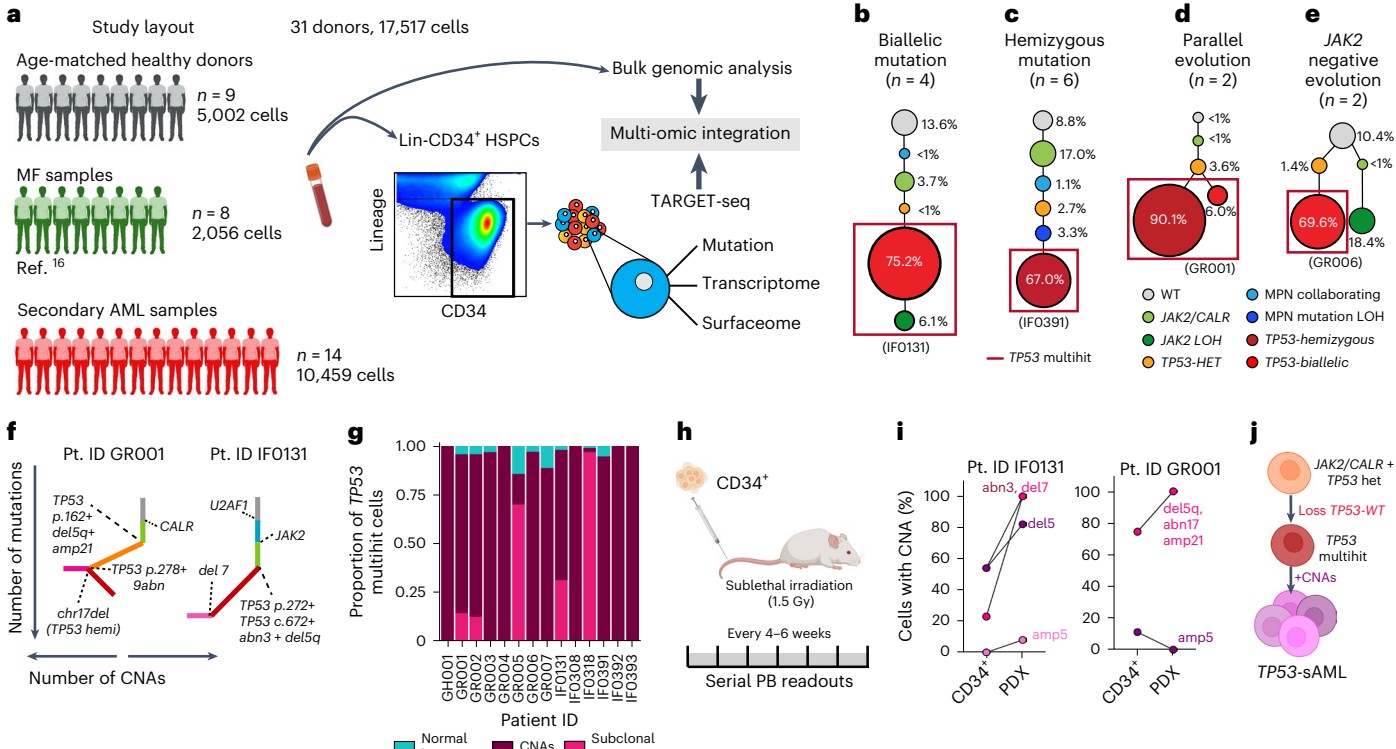

**Fig. 1 | Clonal evolution of *TP53*-sAML. a**, Schematic study layout for TARGET-seq profiling of 17517 Lin⁻CD34⁺ HSPCs from 31 donors. **b–e**, Representative examples of the four major patterns of clonal evolution in *TP53*-sAML patients: bi-allelic mutations (**b**), hemizygous mutations (**c**), parallel evolution (**d**) and *JAK2* negative bi-allelic evolution (**e**). The numbers in parenthesis indicate the number of patients in each category. The size of the circles is proportional to each clone's size, indicated as a percentage of total Lin⁻CD34⁺ cells for one representative patient in each group; each clone is colored according to its genotype (related to Extended Data Fig. 2b–o) and red boxes indicate *TP53* multihit clones. **f**, Representative examples from integrated mutation and CNA-based clonal hierarchies. Solid lines indicate the acquisition of a genetic hit (that is point mutation or CNA), whereas dotted lines indicate the specific genetic hit acquired in each step of the hierarchy (related to Extended Data Fig. 2p,q). **g**, Proportion of *TP53* multihit cells classified as carrying clonal or subclonal CNAs in each patient, using a transcriptomic-based CNA clustering approach (inferCNV). **h**, Experimental strategy for xenotransplantation of CD34⁺ cells from *TP53*-sAML patients in immunodeficient mice. **i**, Percentage of cells carrying CNAs found in each PDX and corresponding Lin⁻CD34⁺ cells from the primary *TP53*-sAML sample transplanted (related to Extended Data Fig. 3; *n* = 1). **j**, Model of *TP53*-sAML genetic evolution. Created with BioRender.com.

cellular hierarchy (Extended Data Fig. 5d,e), with erythroid-biased populations being highly enriched in immunophenotypically defined MEPs (Extended Data Fig. 5f). Taken together, these findings support an aberrant erythroid-biased differentiation trajectory in *TP53*-sAML.

To determine whether upregulation of erythroid-associated transcription was a more widespread phenomenon in *TP53*-mutant AML, we investigated erythroid–myeloid-associated transcription in the BeatAML and The Cancer Genome Atlas (TCGA) cohorts[26,27]. Erythroid scores were increased in *TP53* mutant compared to *TP53*-WT AML, whereas there was no significant difference in myeloid scores (Fig. 2e–f, Extended Data Fig. 5g–j and scores described in Supplementary Table 3). Concomitantly, patients with high erythroid scores also showed decreased *TP53*-target gene expression (Extended Data Fig. 5k). We next investigated the expression of key transcription factors for erythroid/granulomonocytic commitment and found increased *GATA1* expression in Lin⁻CD34⁺ *TP53* multihit HSPCs, whereas *CEBPA* was only expressed at low levels (Fig. 2g). Analysis of the BeatAML cohort revealed increased *GATA1* and reduced *CEBPA* expression in association with *TP53* mutation (Extended Data Fig. 5l), with consequent reduction in the *CEBPA/GATA1* expression ratio (Fig. 2h). Similar findings were observed in *TP53* knock-out or mutant isogenic MOLM13 cell lines (Extended Data Fig. 5m)[28]. These observations suggest that the *CEBPA/GATA1* expression ratio, an important transcription factor balance that affects erythroid versus myeloid differentiation in leukemia[29,30], is disrupted by *TP53* mutation.

To determine whether p53 directly influences myeloid–erythroid differentiation, we knocked down *TP53* in *JAK2V617F* CD34⁺ cells from MPN patients (Extended Data Fig. 5n). *TP53* knock-down led to increased erythroid (CD71⁺CD235a⁺) and decreased myeloid (CD14⁺/CD15⁺/CD11b⁺) differentiation in vitro (Fig. 2i), and consequently decreased *CEBPA/GATA1* expression ratio (Fig. 2j), suggesting that p53 may directly contribute to the aberrant myelo-erythroid differentiation observed.

As 'stemness scores' have previously been applied to determine prognosis in AML[31], we next asked whether a single-cell defined *TP53* multihit LSC signature might identify AML patients with adverse outcomes. Single-cell multi-omics allowed us to derive a 44-gene 'p53LSC-signature' (Supplementary Table 4) by comparing gene expression of HD, *JAK2*-mutant MF HSPC and *TP53* WT pre-LSC to transcriptionally defined *TP53*-mutant LSCs (Fig. 2a,k). High p53LSC-signature score (Extended Data Fig. 6a,b) was strongly associated with *TP53* mutation status, although some *TP53*-WT patients also showed a high p53LSC score. A high p53LSC score predicted poor survival in the independent BeatAML and TCGA cohorts, irrespective of *TP53* mutational status (Fig. 2l and Extended Data Fig. 6c–e). The p53LSC signature performed well as a predictor of survival, including in sAML patients, as compared to the previously published LSC17 score[31] and p53-mutant score generated using all *TP53*-mutant HSPC rather than LSCs (Extended Data Fig. 6f,g and Supplementary Table 4), providing a powerful tool to aid risk stratification in AML.

## *TP53*-WT cells display self-renewal and differentiation defects

TARGET-seq uniquely enabled phenotypic and molecular characterization of rare *TP53*-WT cells, referred to as pre-LSCs, which include both residual HSPCs that were WT for all mutations analyzed, as well as HSPCs that form part of the antecedent MPN clone. These pre-LSCs were obtained in sufficient numbers (>20 cells) from 9 of 14 *TP53*-sAML patients, including all patterns of clonal evolution (Fig. 3a and Extended Data Fig. 7a). Pre-LSCs representing the antecedent MPN clone (positive for MPN-associated driver mutations) were more frequent (60.5%) than pre-LSCs that were WT for all mutations (39.5%). Pre-LSCs were enriched in HSC-associated genes and mapped onto HSC clusters in healthy and MF donor hematopoietic hierarchies (Fig. 3a,b). Index sorting revealed that pre-LSCs were strikingly enriched in the phenotypic HSC compartment, unlike *TP53* multihit HSPCs (Fig. 3c and Extended Data Fig. 3a). Pre-LSCs were rare, as reflected by a reduction in the numbers of phenotypic HSCs present within the Lin⁻CD34⁺ HSPC compartment in *TP53*-sAML compared to HDs (Extended Data Fig. 7b).

We reasoned that the HSC phenotype of pre-LSCs, with reduced frequency in progenitor compartments, might reflect impaired differentiation. To explore this hypothesis, we carried out scVelo analysis, which showed the absence of a transcriptional differentiation trajectory in pre-LSCs, unlike HD HSCs (Fig. 3d). Pre-LSCs showed increased expression of HSC and Wnt β-catenin genes and decreased cell cycle genes as compared to HD and MF cells (Fig. 3e–g and Supplementary Table 3). To functionally confirm these findings, we sorted phenotypic HSCs (to purify pre-LSCs), as well as other progenitor cells, from HDs, MF and *TP53*-sAML patients for long-term culture-initiating cell (LTC-IC) and short-term cultures (Fig. 3h and Extended Data Fig. 7c). Pre-LSC LTC-IC activity was similar to HDs and increased compared to MF patients, with preserved terminal differentiation capacity and confirmed *TP53* WT genotype (Fig. 3i and Extended Data Fig. 7d–g). In short-term liquid culture, pre-LSCs showed reduced clonogenicity, with retained CD34 expression and decreased proliferation (Fig. 3j and Extended Data Fig. 7h–i). In summary, we identified rare and phenotypically distinct pre-LSCs from *TP53*-sAML samples, which were characterized by differentiation defects and distinct stemness, self-renewal and quiescence signatures. As these cells were *TP53* WT and showed normal differentiation after prolonged ex vivo culture, we reasoned that these functional and molecular abnormalities are likely to be cell-extrinsically mediated. Indeed, pre-LSCs showed enrichment of gene signatures associated with certain cell-extrinsic inflammatory mediators (TNFα, IFNγ, TGFβ and IL2; Fig. 3k).

## Inflammation promotes *TP53*-associated clonal dominance

To understand the transcriptional signatures associated with leukemic progression, we analyzed samples from 5 CP-MPN patients who subsequently developed *TP53*-sAML (pre-*TP53*-sAML) alongside 6 CP-MPN patients harboring *TP53*-mutated clones who remained in CP (CP *TP53*-MPN, median 4.43 years (2.62–5.94) of follow-up; Fig. 4a and Extended Data Fig. 8). Compared to *TP53*-sAML samples, CP *TP53*-MPN had a lower VAF and number of *TP53* mutations (Extended Data

Fig. 8a–d). The type, distribution and pathogenicity score of *TP53* mutations were similar between chronic and acute stages (Extended Data Fig. 8e,f). All five pre-*TP53*-sAML samples and four of the six CP *TP53*-MPN were then analyzed by TARGET-seq (Fig. 4a). HSPC immunophenotype was similar for pre-*TP53*-sAML and CP *TP53*-MPN patients (Extended Data Fig. 9a–c), and clearly distinct from the *TP53*-sAML stage (Extended Data Fig. 9d). Heterozygous *TP53* clones were identified in 3 pre-*TP53*-sAML patients and all 4 CP *TP53*-MPN (Fig. 4b and Extended Data Fig. 9e–m). A minor homozygous *TP53*-mutated clone initially present in one CP *TP53*-MPN patient was undetectable after 4 years (Extended Data Fig. 9h). As *TP53*-heterozygous mutant HSPCs represent the direct genetic ancestors of *TP53* 'multihit' LSCs, we compared gene expression of heterozygous *TP53*-mutant HSPC from pre-*TP53*-sAML (*n* = 296) to CP *TP53*-MPN (*n* = 273; Fig. 4b, blue box) to identify putative mediators of transformation. *TP53*-heterozygous HSPC from pre-*TP53*-sAML patients showed downregulation of TNFα- and TGFβ-associated gene signatures, both of which are known to be associated with HSC attrition[32,33], with upregulated expression of oxidative phosphorylation, DNA repair and interferon (IFN) response genes (Supplementary Table 5 and Fig. 4c–e), without changes in IFN receptor expression levels or concurrent IFN treatment (Extended Data Fig. 9n and Supplementary Table 1). Upregulation of inflammatory signatures was detected in *TP53*-homozygous cells from the same pre-*TP53*-sAML patients at a higher level than in *TP53*-heterozygous cells (Extended Data Fig. 9o). Collectively, these findings raise the possibility that inflammation might contribute to preleukemic clonal evolution toward *TP53*-sAML.

To evaluate the role of inflammation in *TP53*-driven leukemia progression, we performed competitive mouse transplantation experiments between CD45.1⁺ Vav-iCre *Trp53*^R172H/+ and CD45.2⁺ *Trp53*^+/+ BM cells followed by repeated poly(I:C) or lipopolysaccharide (LPS) intraperitoneal injections. These experiments recapitulate chronic inflammation through induction of multiple pro-inflammatory cytokines[34,35] known to be increased in the serum of patients with MPN[36], including IFNγ (Fig. 5a and Extended Data Fig. 10a). *Trp53*-mutant peripheral blood (PB) myeloid cells, BM HSCs (Lin⁻Sca1⁺c-Kit⁺CD150⁺CD48⁻) and LSKs (Lin⁻Sca1⁺c-Kit⁺) were selectively enriched upon poly(I:C) treatment (Fig. 5b,c and Extended Data Fig. 10b–e). Crucially, the fitness advantage of *Trp53*-mutant HSCs and LSKs was exerted both through an increase in the numbers of *Trp53*^R172H/+ HSPCs and a reduction in the numbers of WT competitors (Fig. 5d,e and Extended Data Fig. 10f,g). Treatment of chimeric mice with LPS (Fig. 5a), which induces an inflammatory response mediated through the release of IL1β and IL6 (ref. 37), among others, led to a similar increase in the number of *Trp53*-mutant PB myeloid cells and LSKs (Fig. 5f,g). These results indicate that a variety of inflammatory stimuli can promote expansion of the *Trp53*-mutant clone.

To determine how inflammation might alter hematopoietic differentiation and exert a selective pressure to drive the expansion of the *Trp53*-mutant clone, we established an inducible SCL-CreER^T *Trp53*^R172H/+ mouse model (Fig. 5h). Poly(I:C) treatment led to

**Fig. 2 | Distinct differentiation trajectories and molecular features of *TP53*-sAML. a**, Three-dimensional diffusion map of 8988 Lin⁻CD34⁺ cells from 14 sAML samples colored by *TP53* genotype (left), LSC score (middle) and erythroid transcription score (right). **b**, Monocle3 pseudotime ordering of the same single cells as in **a**. **c,d**, UMAP representation of an HD hematopoietic hierarchy (**c**; ref. 24) and latent semantic index projection of *TP53* multihit cells from 14 sAML patients (**d**, top) and cells from de novo AML patients (**d**, bottom; ref. 25) onto the HD hematopoietic hierarchy atlas (**c**). **e,f**, Expression of an erythroid (**e**) and myeloid (**f**) gene score in AML patients from the BeatAML dataset stratified by *TP53* mutational status (*n* = 329 *TP53* WT; *n* = 31 *TP53* mutant). **g**, *CEBPA* (top) and *GATA1* (bottom) expression in the same cells as in **a** and **b**. **h**, *CEBPA* and *GATA1* expression ratio in the same patient cohort as in **e** and **f**. **i,j**, Proportion of immature erythroid (CD235a⁺CD71⁺) and myeloid (CD14⁺,

CD15⁺ or CD11b⁺) cells (**i**) and ratio of *CEBPA* to *GATA1* expression in total cells (**j**) after 12 d of differentiation of peripheral blood CD34⁺ cells from patients with MPN transduced with shRNA targeting *TP53* or shCTR. *n* = 5 patients, three independent experiments. Barplot indicates mean ± s.e.m. and two-tailed paired *t*-test *P* value (related to Extended Data Fig. 5n). **k**, Schematic representation of the key analytical steps to derive a 44-gene *TP53*-LSC sAML signature. **l**, Kaplan–Meier analysis of AML patients (*n* = 322) from the BeatAML cohort stratified by p53-LSC signature score (high, above median; low, below median) derived in **k** (related to Extended Data Fig. 6). *P* indicates log-rank test *P* value and HR, hazard ratio. All boxplots represent the median, first and third quartiles, and whiskers correspond to 1.5 times the interquartile range; '*P*' indicates Wilcoxon rank sum two-sided test *P* value in panels **e**,**f**,**h**.

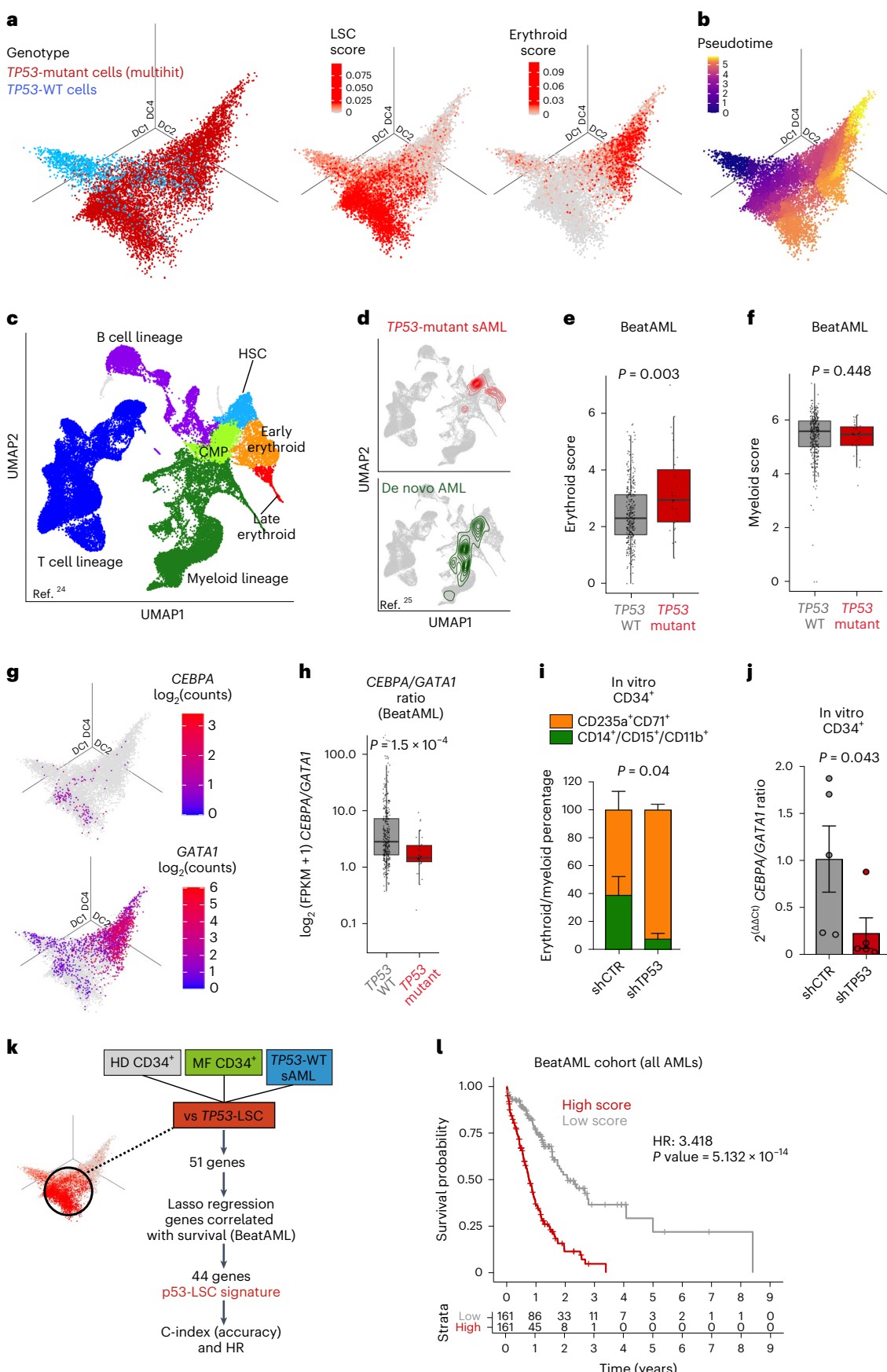

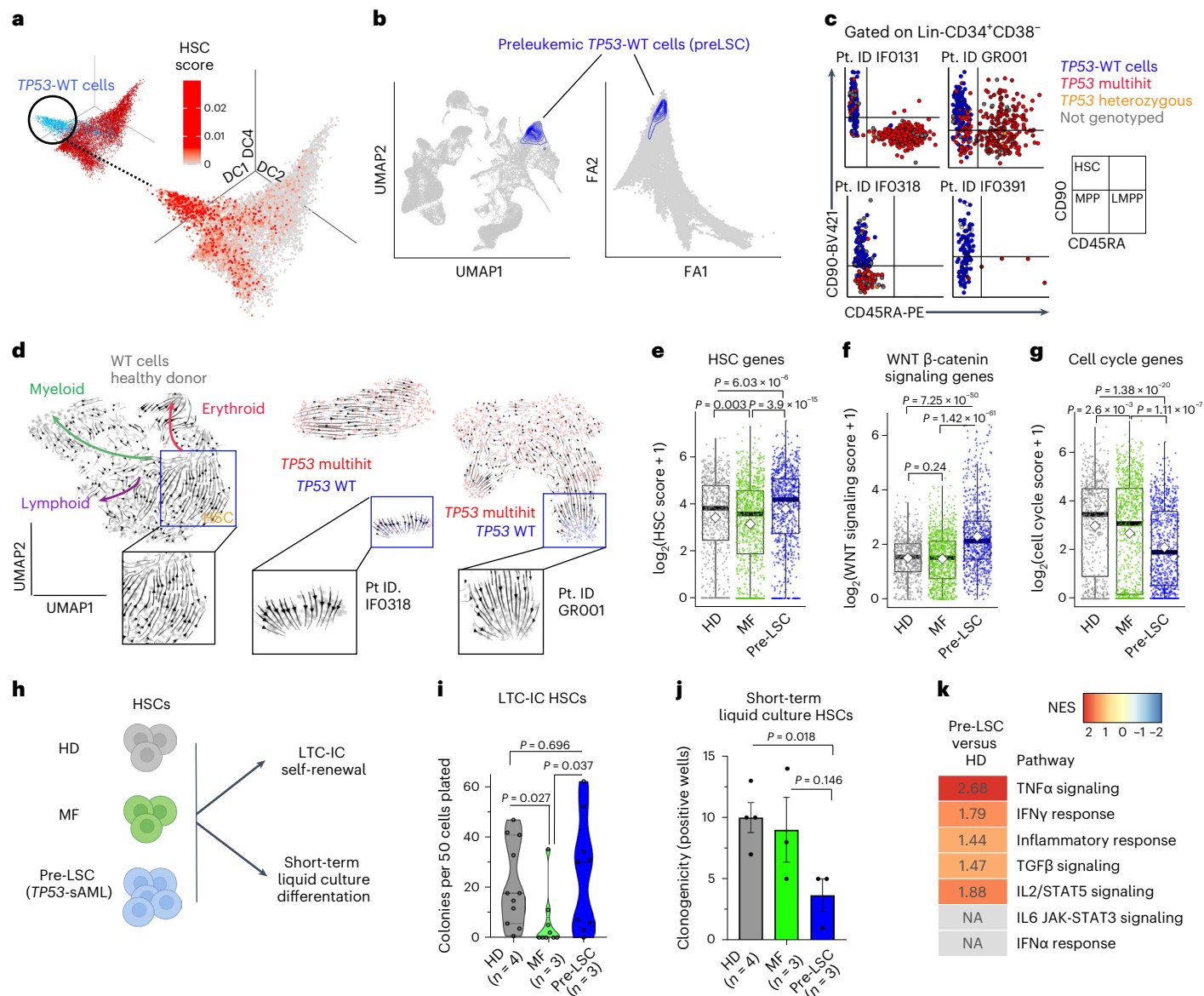

**Fig. 3 | Molecular and functional analysis of pre-LSCs in *TP53*-sAML patients.** **a**, Three-dimensional diffusion map of 8,988 Lin⁻CD34⁺ cells from *TP53*-sAML patients (related to Fig. 2a) colored by expression of an HSC signature (Supplementary Table 3). **b**, Projection of *TP53*-WT (n = 880) pre-LSCs on HD (left) and MF (right) hematopoietic hierarchy (related to Fig. 2c and Extended Data Fig. 5d). **c**, Immunophenotype of Lin⁻CD34⁺CD38⁻ cells from four representative sAML patients colored by genotype. Lin⁻CD34⁺CD38⁻CD90⁺CD45RA⁻ cells (HSCs) were enriched using the sorting strategy outlined in Extended Data Fig. 2a. **d**, scVelo analysis of differentiation trajectories of Lin⁻CD34⁺ cells from one HD (left) and two representative *TP53*-sAML patients (middle and right). Insets show HSC or pre-LSCs clusters. **e–g**, Scores of HSC (**e**), WNT β-catenin signaling (**f**) and cell-cycle (**g**) associated transcription in Lin⁻CD34⁺CD38⁻ cells from HDs (n = 730 cells), MF (n = 1,106 cells) and pre-LSCs from *TP53*-sAML patients (n = 880 cells). Boxplots represent the median, first and third quartiles,

and whiskers correspond to 1.5 times the interquartile range; the white square indicates the mean for each group. *P* indicates the Wilcoxon rank sum test *P* value. **h–j**, Functional analysis of pre-LSCs. Schematic representation of HSC in vitro assays (**h**), LTC-IC colony forming unit activity (**i**) and short-term in vitro liquid culture clonogenicity (**j**) of HSC from HDs (n = 4), MF (n = 3) and pre-LSCs from *TP53*-sAML patients (n = 3, samples used (IFO131, IFO391 and GR001) were known to have *TP53*-WT pre-LSC in the HSC compartment). Violin plot indicates points' density; dashed lines, median and quartiles, two independent experiments (**i**); barplot indicates mean ± s.e.m., three independent experiments with 30 colonies plated per experiment (**j**). *P* indicates two-tailed *t*-test *P* value. **k**, GSEA analysis of HALLMARK inflammatory pathways in pre-LSCs compared to HDs; positive NES in the heatmap represents significant (FDR *q* value < 0.25) enrichment in pre-LSCs, values indicate NES for each pathway.

inflammation-associated changes in blood cell parameters, including anemia, leucopenia and thrombocytopenia (Extended Data Fig. 10h–j). Similar to the Vav-iCre model, poly(I:C) treatment promoted the selection of myeloid *Trp53*-mutant cells in the PB (Extended Data Fig. 10k), with a myeloid bias induced by the inflammatory stimulus in PB leukocytes specifically associated with *Trp53*-mutation (Fig. 5i,j and Extended Data Fig. 10l). Analysis of HSPCs showed the expected selection for *Trp53*-mutant HSCs and LSKs following Poly(I:C)

treatment (Extended Data Fig. 10m). Numbers of WT competitor erythroid progenitors were reduced upon poly(I:C) treatment as expected[38], whereas *Trp53*-mutation was associated with an increase in erythroid progenitors that was not impacted by inflammation (Fig. 5k and Extended Data Fig. 10n) in line with the erythroid bias detected in patient samples. Finally, to determine the mechanisms by which inflammation might promote a fitness advantage for *Trp53*-mutated cells, we performed cell cycle and apoptosis

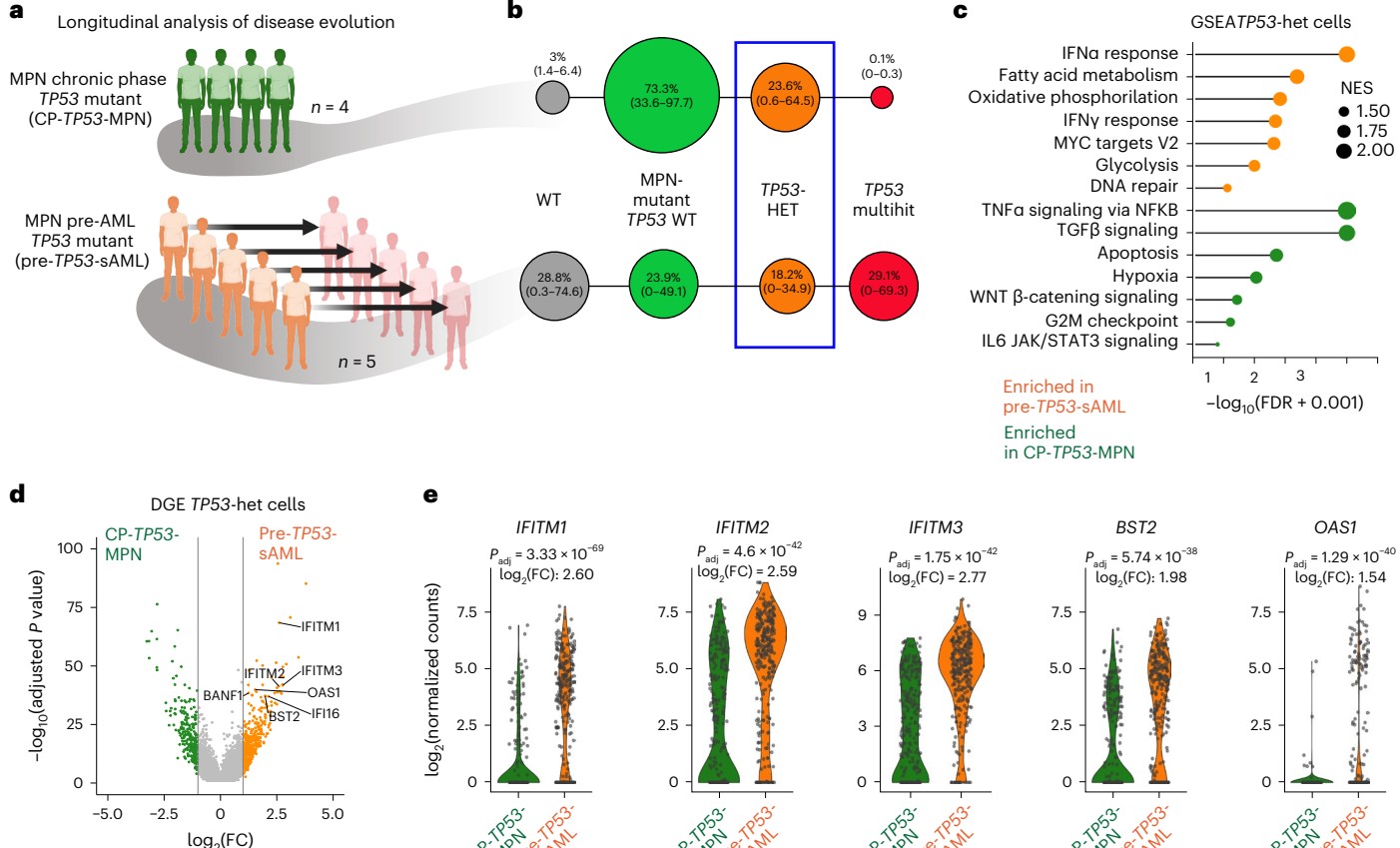

**Fig. 4 | Inflammatory pathways are upregulated in *TP53*-mutant HSPCs before transformation. a**, Schematic study layout of the CP and paired samples patient cohort selected for TARGET-seq analysis. Created with BioRender.com. **b**, Clonal evolution of *TP53*-mutant CP patient samples without clinical evidence of transformation (CP-*TP53*-MPN, *n* = 4) and pre-*TP53*-sAML (patients who subsequently transformed to *TP53*-sAML) samples (*n* = 5). The size of the circles is proportional to the average percentage of cells mapping to each clone, and each clone is colored according to its genotype (related to Extended Data Fig. 9e–m). *TP53*-heterozygous cells selected for subsequent transcriptional analysis are indicated by the blue box. **c,d**, GSEA of selected differentially expressed pathways

(**c**) and volcano plot of differentially expressed genes (**d**) in *TP53*-mutant heterozygous cells from CP *TP53*-MPN (green; *n* = 273 cells) and pre-*TP53*-sAML (orange; *n* = 296 cells). In **d**, genes involved in the IFN response are labeled. **e**, Expression of key IFN-response genes in *TP53*-heterozygous cells from the same patients as in **c** and **d**. In **d** and **e**, $P_{adj}$ indicates adjusted *P* value from combined Fisher's exact test and Wilcoxon tests, calculated using Fisher's method and adjusted using Benjamini–Hochberg procedure; FC indicates fold-change. Violin plots indicate $\log_2$(counts) distributions and each point represents the expression value of a single cell.

analysis following chronic poly(I:C) treatment. Cell cycle was similarly increased in poly(I:C)-treated WT and *Trp53*-mutated LSKs[39]; however, *Trp53*-mutated LSKs were resistant to inflammation-induced apoptosis[40] in comparison with their WT counterparts (Fig. 5l,m).

### Inflammation promotes the genetic evolution of *Trp53*-mutant HSPC

As exit from dormancy promotes DNA-damage-induced HSPCs attrition[41], we reasoned that *Trp53* mutation might rescue HSPCs that acquire DNA damage (and would otherwise undergo apoptosis) driven by chronic inflammation-associated proliferative stress. To explore this possibility, we carried out multiplex fluorescence in situ hybridization (M-FISH) karyotype analysis of *Trp53*[+/+] LSKs expanded in vitro from mice following poly(I:C) treatment and *Trp53*[R172H/+] LSKs from mice with or without poly(I:C) treatment. WT competitor LSK-derived cells from poly(I:C) treated mice were karyotypically normal. In contrast, we observed a striking increase in the frequency and number of karyotypic abnormalities in *Trp53*-mutated LSK-derived cells upon poly(I:C) treatment (Fig. 6a–d). Collectively, these results support a model whereby chronic inflammation promotes the survival and genetic evolution of *TP53*-mutated cells while suppressing WT hematopoiesis, ultimately promoting the clonal expansion of *TP53*-mutant HSPCs (Fig. 6e).

## Discussion

Here we unravel multilayered genetic, cellular and molecular intra-tumoral heterogeneity in *TP53* mutation-driven disease transformation through single-cell multi-omic analysis. Allelic resolution genotyping of leukemic HSPCs revealed a strong selective pressure for gain of *TP53* missense mutation, loss of the *TP53*-WT allele and acquisition of complex CNAs, including cases with parallel genetic evolution during *TP53*-sAML LSC expansion. Despite the known dominant negative and/or gain of function effect of certain *TP53* mutations[28,42], loss of the *TP53*-WT allele, a genetic event associated with a particularly dismal prognosis[2], confers an additional fitness advantage to *TP53*-sAML LSCs. As CNA were universally present in *TP53*-sAML with a very high clonal burden, it is not possible, even with high-resolution single-cell analyses, to disentangle the impact of *TP53*-multi-hit mutation versus the effects of patient-specific CNA that were inextricably linked in all patients analyzed.

Three distinct clusters of HSPCs were identified in *TP53*-sAML, including one characterized by overexpression of erythroid genes, of particular note as erythroleukemia is a rare entity, associated with adverse outcomes and *TP53* mutation[43,44]. Analysis of a large AML cohort also revealed overexpression of erythroid genes as a more wide-spread phenomenon in *TP53*-mutant AML, with disrupted balance of

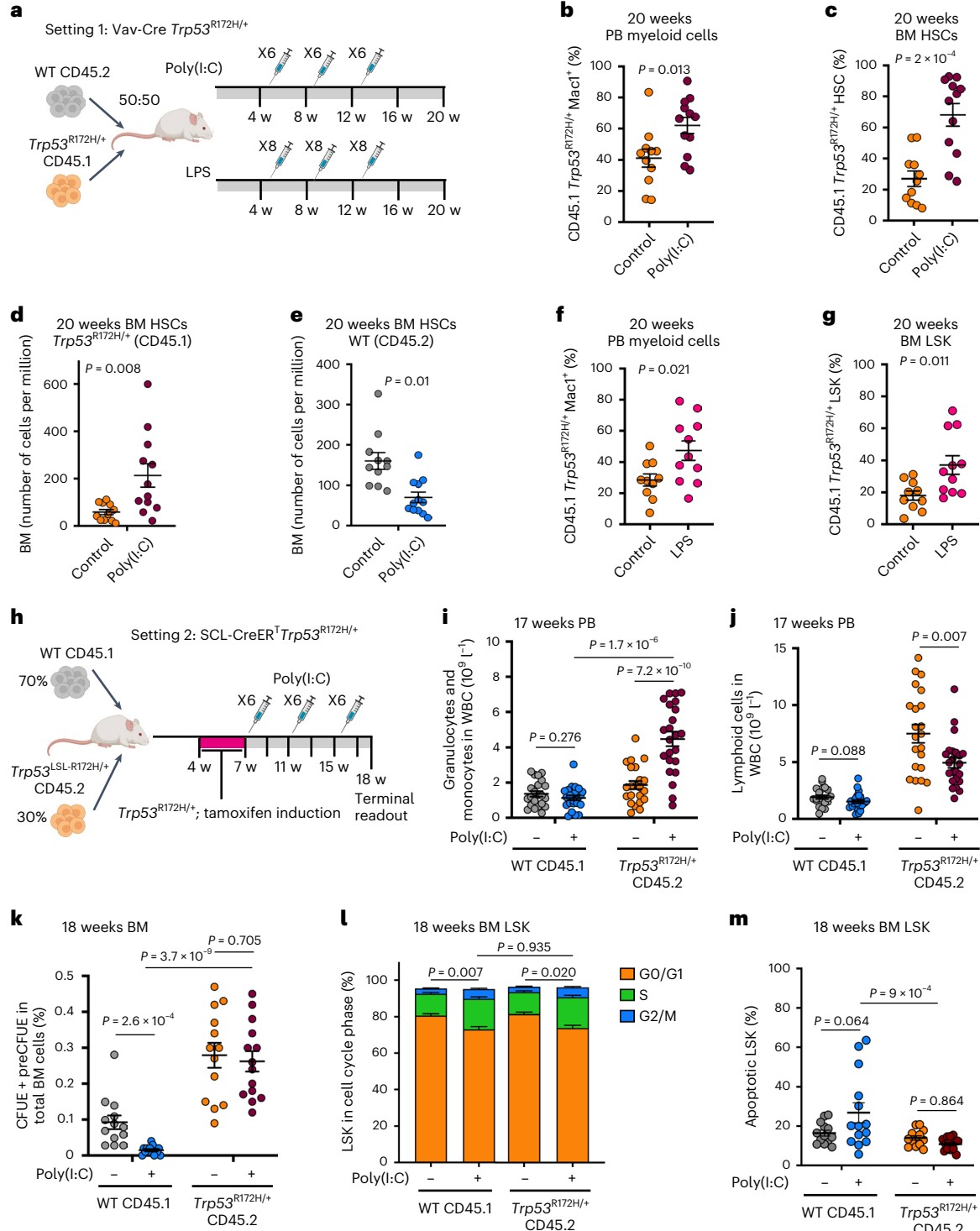

**Fig. 5 | Inflammation promotes *TP53*-associated clonal dominance.**
**a**, Experimental design of *Vav-iCre* WT:*Trp53*[R172H/+] chimera serial poly(I:C) and LPS treatment. **b–e**, Analysis of chimera mice 20 weeks post-transplantation following three cycles of six poly(I:C) injections. Percentage of CD45.1 *Trp53*[R172H/+] Mac1+ cells in the PB (**b**) or BM HSCs (Lin-Sca-1+c-Kit+CD150+CD48−; **c**), number of BM CD45.1 *Trp53*[R172H/+] HSC (**d**) and CD45.2 WT HSC (**e**) per million BM cells. *n* = 11–12 mice per group in two independent experiments and three biological replicates. Mean ± s.e.m. is shown and *P* indicates a two-tailed unpaired *t*-test *P* value. **f,g**, Analysis of chimera mice 20 weeks post-transplantation following three cycles of eight LPS injections. Percentage of CD45.1 *Trp53*[R172H/+] Mac1+ cells in the PB (**f**), or BM LSKs (Lin-Sca-1+c-Kit+; **g**). *n* = 10–11 mice per group in two independent experiments and two biological replicates. Mean ± s.e.m. is shown and *P* indicates a two-tailed unpaired *t*-test *P* value. **h**, Experimental design of

*SCL-Cre-ER*[T] WT:*Trp53*[R172H/+] chimera serial poly(I:C) treatment. **i,j**, Absolute counts of CD45.1 WT or CD45.2 *Trp53*[R172H/+] granulo-monocytic (Ly6G+ and/or Mac1+; **i**) and lymphoid (B220+/NK1.1+/CD3+; **j**) PB cells at 17 weeks post-transplant. **k**, Percentage of CD45.1 WT or CD45.2 *Trp53*[R172H/+] erythroid progenitors (Lin-Sca-1-c-Kit+CD41-FcgRII/III-CD105+) in total BM MNC at 18 weeks post-transplant. *n* = 22 control, *n* = 23 poly(I:C) groups (**i**,**j**) or *n* = 13 control, *n* = 14 poly(I:C) groups (**k**) from two independent experiments. Bars indicate mean ± s.e.m. and *P* indicates two-tailed unpaired *t*-test *P* value. **l,m**, Analysis of cell cycle (**l**) and apoptosis (**m**) in BM LSK cells from chimeric mice 18 weeks post-transplantation following three cycles of six poly(I:C) injections as in **h**. *n* = 13 control, *n* = 17 poly(I:C) groups (**l**) or *n* = 13 control, *n* = 14 poly(I:C) groups (**m**) from two independent experiments, mean ± s.e.m. is shown and *P* indicates adjusted *P* value from one-way Anova (in **l**, the *P* value was calculated using G0/G1 cell cycle phase).

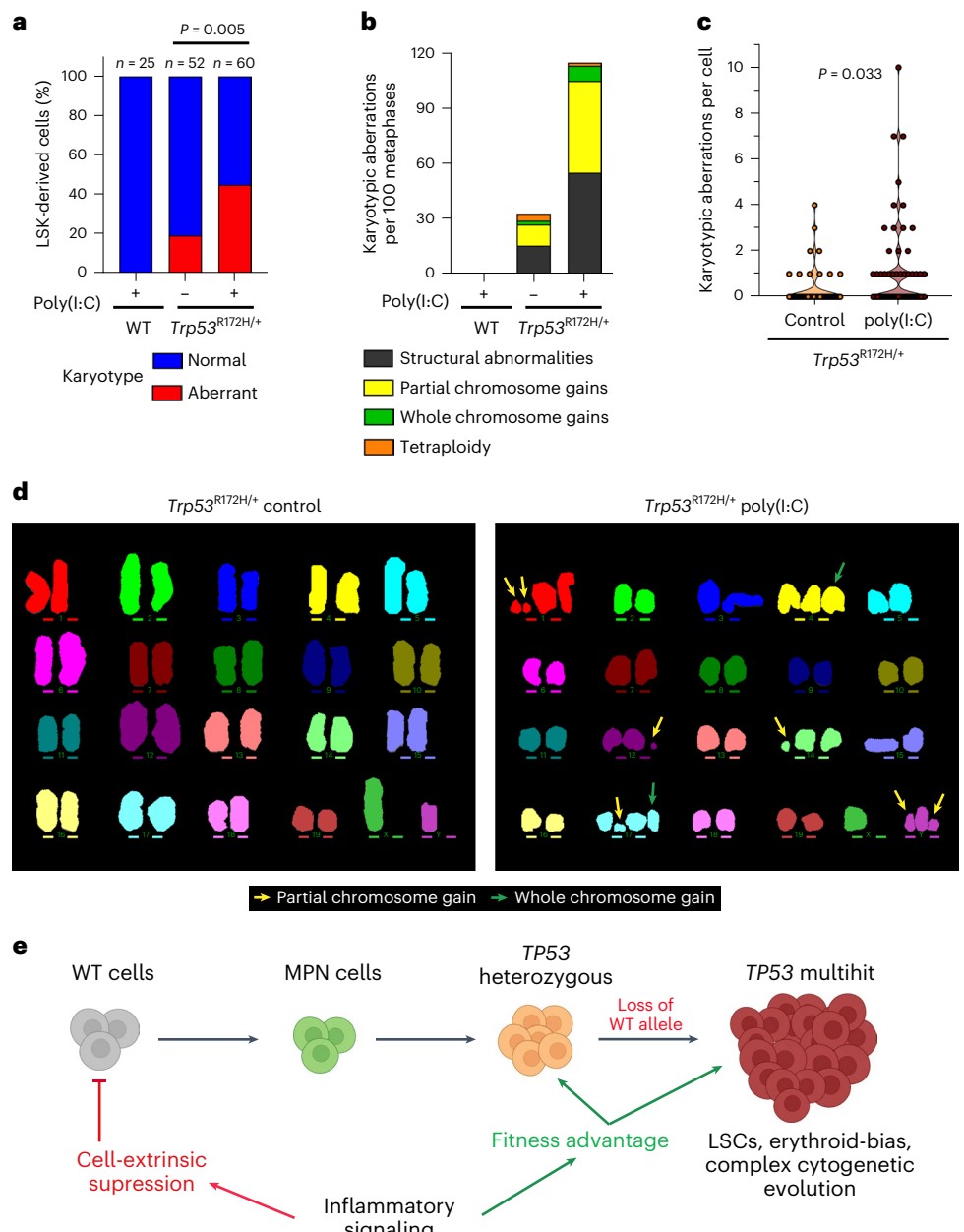

**Fig. 6 | Inflammation leads to genetic instability in *Trp53*-mutant cells.**
**a**–**d**, M-FISH karyotype analysis of LSK-derived cultured cells from CD45.1 (*Trp53*R172H/+) or CD45.2 (WT) LSKs obtained at terminal end-point from chimeric control or poly(I:C) treated mice as in Fig. 5a (*n* = 3 mice per group, *n* = 2 independent experiments). **a**, Percentage of normal and abnormal karyotypes in each experimental group. At the top of each bar, *n* indicates number of metaphases scored. **b**, Type of karyotypic aberrations per hundred metaphases. **c**, Violin plot of the number of karyotypic aberrations per single *Trp53*R172H/+ cell stratified by treatment. **d**, Representative karyotypes from *Trp53*R172H/+ cells obtained from control or poly(I:C) chimeras (yellow arrows indicate partial chromosome gains and green arrows indicate whole chromosome gains). Two-sided Fisher's exact test was carried out to calculate *P* values; in **c**, Fisher's exact test was calculated by testing metaphases with 3 or more aberrations versus metaphases with 0–2 aberrations. **e**, Schematic of the proposed model of *TP53*-mutant-driven transformation in MPN. Created with BioRender.com.

*GATA1* and *CEBPA* expression. *CEBPA* knockout or mutation is reported to cause a myeloid to erythroid lineage switch with increased expression of *GATA1* (refs. 29,30) and, in addition, GATA1 interacts with and inhibits p53 (ref. 45). Notably, our data do not distinguish whether this lineage-switch is primarily an instructive versus permissive effect of *TP53*-mutation[46]. A second '*TP53*-sAML LSC' cluster allowed us to identify a p53LSC-signature, which can predict outcomes in AML independently of *TP53* status. This powerful approach could be more broadly applied in cancer, whereby single multi-omic cell-derived gene scores can be used to stratify larger patient cohorts using bulk gene expression data.

A third *TP53* WT 'pre-LSC' HSPC cluster was characterized by quiescence signatures and defective differentiation, reflecting the impaired hematopoiesis observed in patients with *TP53*-sAML. Through the integration of single-cell multi-omic analysis with in vitro and in vivo functional assays, we show that *TP53*-WT pre-LSCs are cell-extrinsically suppressed while chronic inflammation promotes the fitness advantage of *TP53*-mutant cells, ultimately leading to clonal selection (Fig. 6e). Inflammation is a cardinal regulator of HSC function with many effects on HSC fate and function[47], including proliferation-induced DNA-damage and depletion of HSCs[41]. There is emerging evidence that clonal HSCs can become inflammation-adapted[47–49] and by altering

the response to inflammatory challenges, mutations can thus confer a fitness advantage to HSCs. Here we demonstrate a hitherto unrecognized effect of *TP53* mutations, which conferred a marked fitness advantage to HSPCs in the presence of chronic inflammation induced with both poly(I:C) as well as LPS. We provide evidence that *TP53*-mutant HSPCs showing dysregulated inflammation-associated gene expression are enriched in patients who will develop *TP53*-sAML. We propose that HSCs that would otherwise undergo inflammation-associated and DNA-damage-induced attrition are rescued by *TP53* mutation, ultimately leading to the accumulation of HSCs that have acquired DNA damage, thus promoting genetic evolution that underlies disease progression. This hypothesis was strongly supported through in vivo experiments in which inflammation promoted genetic evolution of *Trp53*-mutant mouse HSPCs. Further studies are required to characterize the key inflammatory mediators and molecular mechanisms involved, which we believe are unlikely to be restricted to a single axis, with a myriad of inflammatory mediators overexpressed in MPN[50]. Furthermore, loss of the *Trp53*-WT allele confers an additional fitness advantage to *Trp53*-mutant HSPC following DNA damage as previously described[28], providing an explanation for the selection for multihit *TP53*-mutant clones observed in patients. Consequently, we believe that approaches that target the inflammatory state, rather than a specific cytokine, are likely to be required to restrain disease progression, as reported for bromodomain inhibitors, which, when combined with JAK2 inhibition, markedly reduce the serum levels of inflammatory cytokines[51]. Collectively, our findings provide a crucial conceptual advance relating to the interplay between genetic and nongenetic determinants of *TP53*-mutation-associated disease transformation. This will facilitate the development of early detection and treatment strategies for *TP53*-mutant leukemia. Because *TP53* is the most commonly mutated gene in human cancer[3,52], we anticipate that these findings will be of broader relevance to other cancer types.

## Online content

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

[1]Haematopoietic Stem Cell Biology Laboratory, Medical Research Council Molecular Haematology Unit, Medical Research Council Weatherall Institute of Molecular Medicine, University of Oxford, Oxford, UK. [2]NIHR Biomedical Research Centre, University of Oxford, Oxford, UK. [3]Centre for Regenerative Medicine 'Stefano Ferrari', Department of Biomedical, Metabolic and Neural Sciences, University of Modena and Reggio Emilia, Modena, Italy. [4]Medical Research Council Centre for Computational Biology, Weatherall Institute of Molecular Medicine, University of Oxford, Oxford, UK. [5]INSERM, UMR 1287, Villejuif, France. [6]Gustave Roussy, Villejuif, France. [7]Université Paris Saclay, Gif-sur-Yvette, France. [8]Université Paris Cité, Paris, France. [9]Department of Cell and Molecular Biology, Karolinska Institutet, Stockholm, Sweden. [10]Karolinska University Hospital, Stockholm, Sweden. [11]Center for Hematology and Regenerative Medicine, Department of Medicine Huddinge, Karolinska Institutet, Karolinska University Hospital, Stockholm, Sweden. [12]Center for Hematological Malignancies, Memorial Sloan Kettering Cancer Center, New York City, NY, USA. [13]Laboratoire d'Hématologie, CHU Dijon, Dijon, France. [14]The Hub for Biotechnology in the Built Environment, Faculty of Health and Life Sciences, Northumbria University, Newcastle upon Tyne, UK. [15]The Wellcome Centre for Human Genetics, Oxford, UK. [16]Sorbonne Université, INSERM, Centre de Recherche Saint-Antoine, AP-HP, Hôpital Saint-Antoine, Service d'hématologie biologique, Paris, France. [17]Département d'Hématologie, Gustave Roussy, Villejuif, France. [18]Laboratoire d'Immuno-Hématologie, Gustave Roussy, Villejuif, France. [19]INSERM, UMR 1231, Centre de Recherche, Dijon, France. [20]Beatson Cancer Centre, Glasgow, UK. [21]Guy's and St Thomas' NHS Foundation Trust, London, UK. [22]Genome Integrity Laboratory, Medical Research Council Molecular Haematology Unit, Medical Research Council Weatherall Institute of Molecular Medicine, University of Oxford, Oxford, UK. [23]Present address: Department of Cancer Biology, Dana Farber Cancer Institute, Boston, MA, USA. [24]Present address: Broad Institute, Cambridge, MA, USA. [25]These authors contributed equally: Alba Rodriguez-Meira, Ruggiero Norfo, Sean Wen, Agathe L. Chédeville. [26]These authors jointly supervised this work: Iléana Antony-Debré, Adam J. Mead. ✉e-mail: albarmeira@gmail.com; ileana.antony-debre@gustaveroussy.fr; adam.mead@imm.ox.ac.uk

## Methods

### Ethical approval, banking and processing of human samples

Primary human samples (PB or BM; described in Supplementary Table 1) were analyzed with approvals from the Inserm Institutional Review Board Ethical Committee (project C19-73, agreement 21-794, CODECOH DC-2020-4324) and from the INForMeD Study (REC: 199833, 26 July 2016, University of Oxford). Patients and normal donors provided written informed consent in accordance with the Declaration of Helsinki for sample collection and use in research. For secondary AML patients, we specifically selected samples from patients with known *TP53* mutation.

Cells were subjected to Ficoll gradient centrifugation and for some samples, CD34 enrichment was performed using immunomagnetic beads (Miltenyi). Total mononuclear cells (MNCs) or CD34$^+$ cells were frozen in FBS supplemented with 10% dimethyl sulfoxide for further analysis.

### Targeted bulk sequencing

Bulk genomic DNA from patient samples' mononuclear or CD34$^+$ cells was isolated using DNeasy Blood & Tissue Kit (Qiagen) or QIAamp DNA Mini Kit (Qiagen) as per the manufacturer's instructions. Targeted sequencing was performed using a TruSeq Custom Amplicon panel (Illumina) or a Haplex Target Enrichment System (Agilent Technologies) with amplicons designed around 32, 44 or 77 genes[53]. Targets were chosen based on the genes/exons most frequently mutated and/or likely to alter clinical practice (diagnostic, prognostic, predictive or monitoring capacity) across a range of myeloid malignancies (for example, MDS/AML/MPN). Targets covered in all panels include *ASXL1, CALR, CBL, CEBPA, CSF3R, DNMT3A, EZH2, FLT3, HRAS, IDH1, IDH2, JAK2, KIT, KRAS, MPL, NPM1, NRAS, PHF6, RUNX1, SETBP1, SF3B1, SRSF2, TET2, TP53, U2AF1, WT1* and *ZRSR2*. Sequencing was performed with a MiSeq sequencer (Illumina), according to the manufacturer's protocols. Raw sequence data in FASTQ format were analyzed using the following variant callers and as previously described[16,53]: BWA v-0.7.12 (read alignment); Picard-tools (marking duplicates); samtools v-1.2; v-1.139 (BAM file creation); GATK HaplotypeCaller v-3.4-46 GRVC v-1.1; snpEff v-4.0 (variant calling). Run quality control included %DP_100X (>95%), %DP_200X (>90%), number of reads per sample and % reads q30 forward and reverse (>85%), read quality mean (>30) and percentage of mapped reads (>75%). A minimum of ten reads was required for variant calling. Results were analyzed after alignment of the reads using two dedicated pipelines, SOPHiA DDM (Sophia Genetics) and an in-house software GRIO-Dx. All pathogenic variants were manually checked using Integrative Genomics Viewer software. The analysis is presented in Extended Data Figs. 1a and 8a.

Pathogenic scores for each *TP53* variant (Extended Data Fig. 8e) were derived from the Catalog of Somatic Mutations in Cancer using the FATHMM-MKL algorithm. The FATHMM-MKL algorithm integrates functional annotations from ENCODE with nucleotide-based hidden Markov models to predict whether a somatic mutation is likely to have functional, molecular and phenotypic consequences. Scores greater than 0.7 indicate that a somatic mutation is likely pathogenic, while scores less than 0.5 indicate a neutral classification.

The type and location of *TP53* mutations from this study, de novo AML patients and CHIP individuals represented in Extended Data Fig. 8f were generated using Pecan Portal[54]. De novo AML *TP53* mutations were downloaded from ref. 55 and ref. 27; CHIP-associated *TP53* mutations were obtained from refs. 56–58.

### Sanger sequencing of patient-associated mutations in PDX models

Genomic DNA from PDX sorted populations (LMPP: hCD45$^+$Lin$^-$CD34$^+$CD38$^-$CD45RA$^+$CD90$^-$ and GMP: hCD45$^+$Lin$^-$CD34$^+$CD38$^+$CD45RA$^+$CD123$^+$) was extracted using QIAamp DNA Mini Kit (Qiagen). Sanger sequencing was performed with forward or reverse primers (Supplementary

Table 6a) targeting mutations identified by targeted bulk sequencing in the corresponding primary samples using Mix2seq kit (Eurofins Genomics) and sequences were analyzed with the ApE editor.

### SNP array sample preparation, copy number variant and loss of heterozygosity analysis

Bulk genomic DNA from patients' MNCs was isolated using DNeasy Blood & Tissue Kit (Qiagen) as per the manufacturer's instructions. 250 ng of gDNA was used for hybridization on an Illumina Infinium OmniExpress v1.3 BeadChips platform.

To call mosaic copy number events in primary patient samples, genotyping intensity data generated were analyzed using the Illumina Infinium OmniExpress v1.3 BeadChips platform. Haplotype phasing, calculation of log R ratio (LRR) and B-allele frequency (BAF), and calling of mosaic events were performed using MoChA WDL pipeline v2021-01-20 (MoChA: a BCFtools extension to call mosaic chromosomal alterations starting from phased VCF files with either BAF and LRR or allelic depth) as previously described[59,60]. In brief, MoChA comprises the following steps: (1) filtering of constitutional duplications; (2) use of a parameterized hidden Markov model to evaluate the phased BAF for variants on a per-chromosome basis; (3) deploying a likelihood ratio test to call events; (4) defining event boundaries; (5) calling copy number and (6) estimating the cell fraction of mosaic events. A series of stringent filtering steps were applied to reduce the rate of false positive calls. To eliminate possible constitutional and germline duplications, we excluded calls with lod_baf_phase <10, those with length <500 kbp and rel_cov >2.5, and any gains with estimated cell fraction >80%, $\log(R) > 0.5$ or length <24 Mb. Given that interstitial LOH are rare and likely artefactual, all LOH events <8 Mb were filtered[59]. Events on genomic regions reported to be prone to recurrent artifact[59] (chr6 < 58 Mb, chr7 > 61 Mb and chr2 > 50 Mb) were also filtered, and those where manual inspection demonstrated noise or sparsity in the array.

To find common genomic lesions on a focal and arm level, Infinium OmniExpress arrays were initially processed with Illumina Genome Studio v2.0.4. Following this, LRR data were extracted for all probes and array annotation was obtained from Illumina (InfiniumOmniExpress-24v1-3_A1). LRR data were then smoothed and segmentation called using the CBS algorithm from the DNACopy[61,62] v1.60.0 package in R. A minimum number of five probes was required to call a segment, and segments were analyzed using GenomicRanges[63] v1.38.0. Definitions of amplification, gain, loss and deletion events were as outlined in ref. 64. Segmentation data were then analyzed in GISTIC[65] v2.023.

For PDX models, genomic DNA from sorted populations (LMPP: hCD45$^+$Lin$^-$CD34$^+$CD38$^-$CD45RA$^+$CD90$^-$ and GMP: hCD45$^+$Lin$^-$CD34$^+$CD38$^+$CD45RA$^+$CD123$^+$) was extracted using QIAamp DNA Mini Kit (Qiagen). SNP-CGH array hybridization was performed using the Affymetrix Cytoscan HD (Thermo Fisher Scientific) according to the manufacturer's recommendations. DNA amplification was checked using BioSpec-nano spectrophotometer (Shimadzu) with expected concentrations between 2,500 ng μl$^{-1}$ and 3,400 ng μl$^{-1}$. DNA length distribution post fragmentation was checked using D1000 ScreenTapes on Tapestation 4200 instrument (Agilent Technologies). Cytoscan HD array includes 2.6 million markers combining SNP and nonpolymorphic probes for copy number evaluation. Raw data CEL files were analyzed using the Chromosome Analysis Suite software package (v4.1, Affymetrix) with genome version GRCh37 (hg19) only if achieving the manufacturer's quality cut-offs. Only CNAs >10 kb were reported in the analysis presented in Extended Data Fig. 3k,l.

### Single-molecule cloning and sequencing of patient-derived cDNA

To experimentally verify the bi-allelic nature of *TP53* mutations in *TP53*-sAML patients, cDNA from a selected patient with putative

*TP53* bi-allelic status (patient ID GR004) was PCR-amplified using cDNA-specific primers spanning both *TP53* mutations (fwd: 5′-GACCCTTTTTGGACTTCAGGTG-3′ and rev: 5′-CCATGAGCGCTG CTCAGATAG-3′). PCR amplification was performed with KAPA 2X Ready Mix (Roche), a Taq-derived enzyme with A-tailing activity, for direct cloning into a TA vector (pCR2.1 TOPO vector, TOPO TA Cloning Kit, Invitrogen) as per the manufacturer's instructions. Sanger sequencing for 10 different colonies was performed using M13 forward and reverse primers; a representative example is shown in Extended Data Fig. 1h.

### Fluorescence-activated cell sorting (FACS) and single-cell isolation

Single-cell FACS-sorting was performed as previously described[16], using BD Fusion I and BD Fusion II instruments (Becton Dickinson) for 96-well plate experiments or bulk sorting experiments, and SH800S or MA900 (SONY) for 384-well plate experiments. Experiments involving the isolation of human HSPCs included single color stained controls (CompBeads, BD Biosciences) and Fluorescence Minus One controls (FMOs). Antibodies used for HSPC staining are detailed in Supplementary Table 7a (combinations indicated as Panel A or B).

Briefly, single cells were directly sorted into 384-well plates containing 2.07 µl of TARGET-seq lysis buffer[66]. Lineage-CD34+ cells were indexed for CD38, CD90, CD45RA, CD123 and CD117 markers, which allowed us to record the fluorescence levels of each marker for each single cell. The 7-aminoactinomycin D (7-AAD) was used for dead cell exclusion. Flow cytometry profiles of the human HSPC compartment (Extended Data Figs. 2 and 9) were analyzed using FlowJo software (version 10.1, BD Biosciences).

### Single-cell TARGET-seq cDNA synthesis

Reverse transcription (RT) and PCR steps were performed as previously described[66]. Briefly, SMARTScribe (Takara, 639537) retrotranscriptase, RNASe inhibitor (Takara, 2313A) and a template-switching oligo were added to the cell lysate to perform the retrotranscription step. Immediately after, a PCR mix comprised of SeqAMP (Takara, 638509) and ISPCR primer (binding to a common adapter sequence in all cDNA molecules) was used for the PCR step with 24 cycles of amplification. Target-specific primers spanning patient-specific mutations were also added to RT and PCR steps (Supplementary Table 6a). After cDNA synthesis, cDNA from up to 384 single-cell libraries was pooled, purified using Ampure XP Beads (0.6:1 beads to cDNA ratio; Beckman Coulter) and resuspended in a final volume of 50 µl of EB buffer (Qiagen). The quality of cDNA traces was checked using a high-sensitivity DNA kit in a Bioanalyzer instrument (Agilent Technologies).

### Whole transcriptome library preparation and sequencing

Pooled and bead-purified cDNA libraries were diluted to 0.2 ng µl$^{-1}$ and used for tagmentation-based library preparation using a custom P5 primer and 14 cycles of PCR amplification[66]. Each indexed library was purified twice with Ampure XP beads (0.7:1 beads to cDNA ratio), quantified using Qubit dsDNA HS Assay Kit (Invitrogen, Q32854) and diluted to 4 nM. Libraries were sequenced on a HiSeq4000, HiSeqX or NextSeq instrument using a custom sequencing primer for read1 (P5_seq: GCCTGTCCGCGGAAGCAGT GGTATCAACGCAGAGTTGC*T, PAGE purified) with the following sequencing configuration: 15 bp R1; 8 bp index read; 69 bp R2 (NextSeq) or 150 bp R1; 8 bp index read; 150 bp R2 (HiSeq).

### TARGET-seq single-cell genotyping

After RT-PCR, cDNA + amplicon mix was diluted 1:2 by adding 6.25 µl of DNAse/RNAse free water to each well of 384-well plate. Subsequently, a 1.5 µl aliquot from each single-cell derived library was used as input to generate a targeted and Illumina-compatible library for single-cell genotyping[66]. In the first PCR step, target-specific primers containing a plate-specific barcode (Supplementary Table 6b) were used to amplify the target regions of interest. In a subsequent PCR step, Illumina compatible adapters (PE1/PE2) containing single-direction indexes (Access Array Barcode Library for Illumina Sequencers-384, Single Direction, Fluidigm) were attached to pre-amplified amplicons, generating single-cell barcoded libraries. Amplicons from up to 3,072 libraries were pooled and purified with Ampure XP beads (0.8:1 ratio beads to product; Beckman Coulter). These steps were performed using Biomek FxP (Beckman Coulter), Mosquito (TTP Labtech) and VIAFLO 96/384 (INTEGRA Biosciences) liquid handling platforms. Purified pools were quantified using Qubit dsDNA HS Assay Kit (Invitrogen, Q32854) and diluted to a final concentration of 4 nM. Libraries were sequenced on a MiSeq or NextSeq instrument using custom sequencing primers as previously described[66] with the following sequencing configuration: 150 bp R1; 10 bp index read; 150 bp R2.

### Targeted single-cell genotyping analysis

**Data preprocessing.** For each cell, the FASTQ file containing both targeted gDNA and cDNA-derived sequencing reads was aligned to the human reference genome (GRCh37/hg19) using Burrows–Wheeler Aligner (BWA v0.7.17) and STAR[67] (v2.6.1d). Custom perl scripts were used to demultiplex the gDNA and mRNA reads in the BAM file into separate SAM files based on targeted-sequencing primer coordinates (https://github.com/albarmeira/TARGET-seq). Next, Samtools[68] (v1.9) was used to concatenate the BAM header to the resulting SAM files before reconverting the SAM file to BAM format, which was subsequently sorted by genomic coordinates and indexed. Both gDNA and mRNA reads were tagged with the cell's unique identifier using Picard (v2.3.0) 'AddOrReplaceReadGroups' and duplicate reads were subsequently marked using Picard 'MarkDuplicates'. The sequencing reads overhanging into intronic regions in the mRNA reads were additionally hard-clipped using GATK (v4.1.2.0) SplitNCigarReads[69,70].

**Variant calling.** Variants were called from the processed BAM files using GATK *Mutect2* with the options (--tumor-lod-to-emit 2.0 --disable-read-filter NotDuplicateReadFilter --max-reads-per-alignment-start) to increase the sensitivity of detecting low-frequency variants. The frequency of each nucleotide (A, C, G, T) and indels at each predefined variant site were also called using a Samtools mpileup as previously described[16]. Lastly, the coverage at each predefined variant site was computed using Bedtools[71] (v2.27.1).

To determine the coverage threshold of detection for each variant site, the coverage for 'blank' controls (empty wells) was first tabulated. A cut-off coverage outlier value was computed as having a coverage exceeding 1.5 times the length of the interquartile range from the 75th percentile. Next, a value of 30 was added to this outlier value to yield the final coverage threshold to be used for genotype assignment.

**Genotype assignment.** For each predefined variant site, the number of reads representing the reference and alternative (variant) alleles for indels (insertion and deletions) and single nucleotide variants (SNVs) were tabulated from the outputs of GATK *Mutect2* and Samtools mpileup, respectively.

Here a genotype scoring system was introduced to assign each variant site into one of the following three possible genotypes: WT, heterozygous or homozygous mutant. Chi-square ($\chi^2$) test was first used to compare the observed frequency of reference and alternative alleles against the expected fraction of reference and alternative alleles corresponding to the three genotypes. The expected fraction of the reference alleles was 0.999, 0.5 and 0.001, and the expected fraction of the alternative alleles was 0.001, 0.5 and 0.999 for WT, heterozygous and homozygous mutant genotype, respectively. The $\chi^2$ statistics were then tabulated for each fitted model and converted to genotype scores using the following formula:

$$\text{Score}_{\text{genotype}} = \frac{1}{\log_{10}(\chi^2 + 1)}$$

The genotype assigned to the variant site was based on the genotype model with the highest score.

Next, the variant (alternative) allele frequency (VAF) was computed and variant sites with 2 < VAF < 4 and 96 < VAF < 98 were reassigned as 'ambiguous'. For cells with no variants detected at the specific variant sites by the mutation callers (either due to the absence of the variants, that is WT genotype, or that such variants were present below the detection limit), a 'WT' genotype was assigned to those cells with coverage above the specific threshold and 'low coverage' to those cells with coverage below such threshold.

Taken together, each variant site was assigned one of the five following genotypes: WT, heterozygous, homozygous mutant, ambiguous or low coverage. Variants with ambiguous or low coverage assignments for a particular cell were excluded from the analysis.

### Computational reconstruction of clonal hierarchies

Genotypes for each single cell were recoded for input to SCITE in a manner inspired by ref. 72; each mutation in each gene was coded as two loci, representing two different alleles. In the first recoded loci, all homozygous calls from each mutation where coded as heterozygous genotype calls. In the second recoded loci, all heterozygous and homozygous genotype calls in the original mutation matrix were coded as homozygous reference (that is, WT) and heterozygous, respectively. For example, if for a certain mutation 0 represents WT status, 1 encodes heterozygous and 2 refers to homozygous status, these would be encoded as (0,0), (1,0) and (1,1), respectively, where the first term in the parenthesis corresponds to the first loci and the subsequent to the second loci.

Then, SCITE was used (git revision 2016b31, downloaded from https://github.com/cbg-ethz/SCITE.git; ref. 73) to sample 1,000 mutation trees from the posterior for every single-cell genotype matrix corresponding to a particular patient, where all possible mutation trees are equally likely a priori. For patients in which several disease time points were available, all time points were merged for SCITE analysis. As parameters for every SCITE run '-fd 0.01' (corresponding to the allelic dropout (ADO) rate of reference allele in our adapted SCITE model), '-ad 0.01' (corresponding to the ADO of the alternate allele), a chain length (-l) of 1e6 and a thinning interval of 1 while marginalizing out cell attachments (-p 1 -s) were used.

To summarize the posterior tree sample distribution, the number of times a particular sample matched each tree was computed. For each patient, the most common tree topology in the posterior tree samples is reported (Extended Data Figs. 2b–o and 9e–m), where 'pp' is the proportion of samples that match this tree. For each clade in the most common posterior tree, clade probabilities were estimated as the proportion of trees in the posterior that contained the clade. These are indicated in each square for each mutation in Extended Data Figs. 2b–o and 9e–m.

### Clone assignment.

For every patient's most common posterior tree, we assigned every cell to the tree node that matches the genotype of that particular cell. If an exact match was not found, then for every tree node, the loss of assigning a cell to that node was calculated using the following loss function:

$$l(M) = \log(ADO)(M[1,2] + M[3,2]) + \log(FD)(M[2,1] + M[2,3])$$

$$+ \log(ADO^2 FD)(M[1,3] + M[3,1])$$

where $M$ is a confusion matrix generated across all loci of a cell in which the first index represents the genotype that was measured for that particular cell (1 = homozygous reference, 2 = heterozygous, 3 = homozygous alternate), and the second index represents the genotype implied by the tree node. ADO = 0.01 and FD = 0.001 were used. Every cell was assigned to the node with the lowest loss $l$. For the trees

presented in Extended Data Figs. 2b–o and 9e–m, only the numbers of cells with exact genotype matches were reported.

### Testing for evidence of homozygous genotypes.

Due to the nature of our loci-specific mutation encoding (each gene is encoded as two loci), homozygous mutations are placed in the clonal hierarchy independently of their accuracy. Therefore, for every patient and at every locus with observed homozygous alternate genotype calls, the tested null hypothesis was that all homozygous alternate genotype calls are due to ADO at a level not exceeding 0.05 using a one-tailed binomial test. The total number of draws for the test is the number of heterozygous and homozygous alternate genotype calls at the locus, the number of successful draws is the number of homozygous alternate calls and the success rate is 0.05. Only homozygous alternate genotype calls below this 0.05 cut-off were reported in Extended Data Figs. 2b–o and 9e–m; the results of the binomial test are reported for each patient and mutation in Supplementary Table 8.

### Computational validation of *TP53* bi-allelic status from single-cell targeted genotyping datasets

To further validate the bi-allelic status of *TP53* mutations in our dataset, the patterns of ADO in TARGET-seq single-cell genotyping data from patients carrying at least two different *TP53* mutations were investigated (*n* = 6; Extended Data Fig. 1j).

To test the hypothesis that the observed *TP53*-WT/*TP53*-homozygous (*TP53*-WT/HOM; or (0,2)) cells are the result of a chromosomal loss (and therefore, in different alleles), the following null hypothesis ($H_0$) was formulated: observed *TP53*-WT/HOM cells are double ADO events. Under $H_0$, every *TP53*-WT/HOM cell (0,2), *TP53*-HOM/WT cell (2,0), *TP53*-HOM/HOM (2,2) as well as an unknown number of *TP53*-WT/WT (0,0) are the result of a *TP53*-HET/HET (1,1) cell undergoing ADO at both sites. The following assumptions were made: (1) ADO is unbiased toward HOM or WT and (2) ADO events at each *TP53* site are independent. The null hypothesis was then tested with a binomial test, where the number of (2,2) events should be half the sum of (0,2) + (2,0) events (Extended Data Fig. 1j). (0,0) events were disregarded.

If *TP53* mutations are bi-allelic, the expected number of WT/HOM and HOM/WT would be higher than HOM/HOM cells considering TARGET-seq expected ADO rates (1–5%).

### Single-cell 3′-biased RNA-sequencing data preprocessing

FASTQ files for each single cell were generated using bcl2fastq (version 2.20) with default parameters and the following read configuration: Y8N*, I8, Y63N*. Read 1 corresponds to a cell-specific barcode, index read corresponds to an i7 index sequence from each cDNA pool and read 2 corresponds to the cDNA molecule. PolyA tails were trimmed from demultiplexed FASTQ files with TrimGalore (version 0.4.1). Reads were then aligned to the human genome (hg19) using STAR (version 2.4.2a), and counts for each gene were obtained with FeatureCounts (version 1.4.5-p1; options–primary). Counts were then normalized by dividing each gene count by the total library size of each cell and multiplying this value by the median library size of all cells processed, as implemented in the 'normalize_UMIs' function from the SingCellaR package[74] (version 1.2.1; https://github.com/supatt-lab/SingCellaR). A summary of the preprocessing pipeline can be found at https://github.com/albarmeira/TARGET-seq-WTA.

Quality control was performed using the following parameters: number of genes detected >500, percentage of ERCC-derived reads <35%, percentage of mitochondrial reads <0.25% and percentage of unmapped reads <75%. Cells with less than 2,000 reads in batch1, 5,000 reads in batch2 and 20,000 reads in batch3 were further excluded. This QC step was performed independently for each sequencing batch owing to differences in sequencing depth (mean library size: 42,949 batch 1, 93,580 batch 2 and 171,393 batch 3). After these QC steps, 7,123 cells passed QC for batch 1, 5,779 for batch 2 and 6,319 for batch 3 (79.3%,

68.9% and 80.3% of cells processed, respectively). Then, 2,734 cells from a previously published study[16] corresponding to 8 MF patients and 2 normal donor controls were further integrated, encompassing a final dataset of 21,955 cells in total.

## Identification of highly variable genes

Highly variable genes above technical noise were identified by fitting a gamma generalized linear model (GLM) of the $\log_2$(mean expression level) and coefficient of variation for each gene, using the 'get_variable_genes_by_fitting_GLM_model' from SingCellaR package and the following options: mean_expr_cutoff = 1, disp_zscore_cutoff = 0.1, quantile_genes_expr_for_fitting = 0.6 and quantile_genes_cv2_for_fitting = 0.2. Those genes with a coefficient of variation above the fitted model and expression cut-off were selected for further analysis, excluding those annotated as ribosomal or mitochondrial genes.

## CNA inference from single-cell transcriptomes

InferCNV (v1.0.4) was used to identify CNAs in single-cell transcriptomes[75] (https://github.com/broadinstitute/inferCNV/wiki). Briefly, inferCNV creates genomic bins from gene expression matrices and computes the average level of expression for each of these bins. The expression across each bin is then compared to a set of normal control cells, and CNAs are predicted using a hidden Markov model. For each patient, inferCNV was performed with the following parameters: 'cutoff = 0.1, denoise = T, HMM = T', compared to the same set of normal donor control cells ($n$ = 992). To identify CNA subclones, inferCNV in analysis_mode = 'subclusters' was used. CNAs identified by inferCNV were manually curated by removing those with size <10 kb, merging adjacent CNA calls with identical CNA status into larger CNA intervals and comparing them to SNP-Array bulk CNA calls. Finally, to generate combined TARGET-seq single-cell genotyping and CNA-based clonal hierarchies, the CNA status from each inferCNV cluster was assigned to its predominant genotype.

## Dimensionality reduction, data integration and clustering

PCA was performed using 'runPCA' function from the SingCellaR R package, and Force-directed graph analysis was subsequently performed using the 'runFA2_ForceDirectedGraph' with the top 30 PCA dimensions to generate the plots in Extended Data Fig. 4a.

For the analysis of patient IF0131 presented in Extended Data Fig. 3m, PCA was performed using 'runPCA' function from the *SingCellaR* R package and then UMAP was performed using the 'runUMAP' function with the top ten PCA dimensions and the following options: n.neighbors = 20, uwot.metric = 'correlation', uwot.min.dist = 0.30, n.seed = 1.

Integration of TARGET-seq single-cell transcriptomes from 10,459 cells corresponding to 14 *TP53*-sAML samples was performed using 'runHarmony' function implemented in the SingCellaR package, using the patient ID as covariate and the following options: n.dims.use = 20, harmony.theta = 1, n.seed = 1. Diffusion map analysis was performed using 'runDiffusionMap' with the integrative Harmony embeddings and the following parameters: n.dims.use = 20, n.neighbors = 5, distance = 'euclidean'. Signature scores were calculated using 'plot_diffusionmap_label_by_gene_set' to generate the plots in Figs. 2a and 3a. Only cells with assigned genotypes '*TP53* multihit' and '*TP53*-WT' are shown.

## Pseudotime trajectory analysis

Monocle3 (ref. 76; https://cole-trapnell-lab.github.io/monocle3/) was used to infer differentiation trajectories from single-cell transcriptomes. Raw UMI count matrix and clustering annotations were extracted from the SingCellaR object to build a Monocle3 'cds' object. Next, we retrieved the first two components of the diffusion map (DC1 and DC2), and the 'learn_graph' function was used to calculate the trajectory on the two-dimensional diffusion map, using *TP53*-WT preleukemic cell cluster as the root node. Pseudotime was calculated using

'order_cells' function and overlaid on the diffusion map embeddings to generate the plot in Fig. 2b.

## Differential expression analysis

Differentially expressed genes from TARGET-seq datasets were identified using a combination of nonparametric Wilcoxon test, to compare the expression values for each group, and Fisher's exact test, to compare the frequency of expression for each group, as previously described[17]. Logged normalized counts were used as input for this comparison, including genes expressed in at least two cells. Combined $P$ values were calculated using Fisher's method and adjusted $P$ values were derived using Benjamini–Hochberg procedure. Significance level was set at $P$-adjusted < 0.05. For the analysis presented in Extended Data Fig. 4b and Supplementary Table 2, the top 100 differentially expressed genes with $\log_2$(FC) > 0.3 and at least 20% expressing cells are shown. Differentially expressed genes identified between *TP53*-multihit versus *TP53*-WT cells were further assessed for the enrichment of known p53 target genes (337 curated p53 target genes from ref. 77) for the analysis presented in Extended Data Fig. 4c. We assessed the extent of overlap of these gene lists using the R package GeneOverlap. The overlapping genes were further assessed for the enrichment of p53-related pathways using the R package clusterProfiler.

For the analysis presented in Fig. 2k,l, only genes overexpressed in *TP53* multihit cells and $\log_2$(FC) > 0.75 were included; for Fig. 4d, only those with $\log_2$(FC) > 1 were considered. Violin plots (Fig. 4e and Extended Data Fig. 9n) from selected differentially expressed genes were generated using 'ggplot2' package in R.

## Gene-set enrichment analysis

For analysis involving <600 cells (Fig. 4c and Supplementary Table 5), GSEA was performed using GSEA software version 4.0.3 (www.gsea-msigdb.org/gsea/index.jsp) with default parameters and 1,000 permutations on the phenotype, using $\log_2$(normalized counts).

For analysis involving >600 cells per group (Fig. 3k and Extended Data Figs. 4d and 9o), GSEA was performed with 'identifyGSEAPrerankedGene' function from SingCellaR R package with default options. Briefly, differential expression analysis was performed between two cell populations using the Wilcoxon rank sum test, and the resulting $P$ values were adjusted for multiple testing using the Benjamini–Hochberg approach. Before the differential expression analysis, down-sampling was performed so that both cell populations had the same number of cells. Next, $-\log_{10}(P$ value) transformation was performed and the resulting $P$ values were multiplied by +1 or −1 if the corresponding $\log_2$(FC) was >0.1 or <−0.1, respectively. The gene list was ranked using this statistic in ascending order and used as input for GSEA analysis using 'fgsea' function from the fgsea R package with default options.

MSigDB HALLMARK v7.4 50-gene sets or previously published signatures (https://www.gsea-msigdb.org/gsea/msigdb/cards/GENTLES_LEUKEMIC_STEM_CELL_UP) were used for all analysis. Normalized enrichment scores were displayed in a heatmap using pheatmap R package. Gene sets with false discovery rate (FDR) $q$ value lower than 0.25 were considered significant.

## Projection of single-cell transcriptomes

A previously published human hematopoietic atlas was downloaded from https://github.com/GreenleafLab/MPAL-Single-Cell-2019 and used as a normal hematopoietic reference to project *TP53*-sAML and de novo AML transcriptions using Latent Semantic Index Projection[24]. Common genes to all datasets were selected, and then *TP53*-sAML or previously published de novo AML cells[25] were projected using 'projectLSI' function for the analysis presented in Fig. 2c,d. A previously published human MF atlas[78] was used as a reference to project *TP53*-sAML multihit cells in the analysis presented in Extended Data Fig. 5d,e, using previously defined force-directed graph embeddings.

## Velocyto analysis

Loom files were generated for each single cell using velocyto (v0.17.13) with options -c and -U, to indicate that each BAM represents an independent cell and reads are counted instead of molecules (UMIs), respectively[79]. The individual loom files were subsequently merged using the combine function from the loompy Python module.

HDs with at least 300 cells with RNA-sequencing data and patients with at least 300 cells consisting of >50 preleukemic (*TP53* WT) cells and >50 *TP53* multihit cells were included for analysis. For each individual, the Seurat object was created from the merged loom file and processed for downstream RNA-velocity analysis[80]. Specifically, for each patient, the spliced RNA counts were normalized using regularized negative binomial regression with the SCTransform function[81]. Next, linear dimension reduction was performed using RunPCA function and the first 30 principal components were further used to perform nonlinear dimension reduction using the RunUMAP function. Ninety-six multiple rate kinetics (MURK) genes previously shown to possess coordinated step-change in transcription and hence violate the assumptions behind scVelo were removed[82]. The processed and MURK gene-filtered Seurat object was then saved in h5Seurat format using the SaveH5Seurat function and finally converted to h5ad format using the 'Convert' function.

AnnData object was created from the h5ad file using the scvelo python module for RNA velocity analysis[83]. Highly variable genes were identified and the corresponding spliced and unspliced RNA counts were normalized and $\log_2$-transformed using the scvelo.pp.filter_and_normalize function. Next, the first- and second-order moments were computed for velocity estimation using the scvelo.pp.moments function. The velocities (directionalities) were computed based on the stochastic model as defined in the scvelo.t1.velocity function, and the velocities were subsequently projected on the UMAP embeddings generated from Seurat above. Finally, the UMAP embeddings were annotated using the HSPC and erythroid lineage signature scores[74] and *TP53* mutation status. For each cell, the cell lineage signature score was computed using the average SCTransform expression values of the individual cell lineage genes.

## Analysis of bulk BeatAML and TCGA gene expression datasets

**Data retrieval and preprocessing.** Two publicly available AML cohorts with genetic mutation and RNA-sequencing data available were used to validate findings from our single-cell analysis, namely BeatAML[26] and TCGA[27]. Gene expression values in FPKM (fragments per kilobase of transcript per million mapped reads) were retrieved from the National Cancer Institute (NIH) Genomic Data Commons (GDC)[84]. Gene expression values were then offset by 1 and $\log_2$ transformed. *TP53* point mutation status was retrieved from the cBio Cancer Genomics Portal (cBioPortal)[85]. Clinical data including survival data for BeatAML and TCGA were retrieved from the BeatAML data viewer (Vizome) and NIH GDC, respectively.

We selected samples from the BeatAML cohort with an AML diagnosis (540 de novo AML and 96 secondary AML) collected within 1 month of the patient's enrollment in the study, and with both *TP53* mutation status and RNA-sequencing data available. For patients for whom multiple samples were available, samples were collapsed to obtain patient-level data. Specifically, the mean gene expression value for each gene from multiple samples was used to represent patient-level gene expression value. Furthermore, patients with at least one sample with a *TP53* mutation were considered *TP53*-mutant. Analysis of *TP53* VAF and reported karyotypic abnormalities indicated that the vast majority of patients could be classified as 'multi-hit', and therefore patients were classified as *TP53*-mutant or WT without taking into account *TP53* allelic status. In total, 360 patients with *TP53* mutation status (329 *TP53* WT and 31 *TP53* mutant) and RNA-sequencing data available were included for analysis. Of these, 322 patients had concomitant survival data available (294 *TP53* WT and 28 *TP53* mutant).

The TCGA cohort consisted of 200 de novo AML patients represented by one sample each, of which 151 patients had *TP53* mutation status (140 *TP53* WT and 11 *TP53* mutant) and RNA-sequencing data available, and were included for analysis. Of these, 132 patients had concomitant survival data available (124 *TP53* WT and 8 *TP53* mutant).

**Cell lineage gene signature scores.** For each sample, a given cell lineage gene signature score was computed as the mean expression values of the individual genes belonging to the cell lineage gene signature. Here the gene signature scores for two cell lineages were computed, namely myeloid and erythroid populations. Two gene sets for each cell lineage were compiled. The first gene set was based on cell lineage markers previously reported in the literature, whereas the second gene set was based on cell lineage markers derived from analyzing a published single-cell dataset[24]. Genes from each score are described in Supplementary Table 3.

For the former approach, six erythroid genes (*KLF1, GATA1, ZFPM1, GATA2, GYPA* and *TFRC*; Fig. 2e and Extended Data Figs. 5k,m and seven myeloid genes (*FLI1, SPI1, CEBPA, CEBPB, CD33, MPO* and *IRF8*; Fig. 2f) were identified. For the latter approach, the expression values of erythroid and myeloid cell clusters were first compared separately against all other cell clusters using Wilcoxon ranked sum test. The erythroid cluster consisted of the early and late erythroid populations, while the myeloid cluster consisted of granulocyte, monocyte and dendritic cell populations. Erythroid and myeloid-specific gene signatures were defined as genes having FDR values <0.05 and $\log_2(FC) > 0.5$ in ≥20 and 17 comparisons, respectively. In total, 100 erythroid genes and 135 myeloid genes were identified from this single-cell dataset (Supplementary Table 3) and were used to compute the scores presented in Extended Data Fig. 5g–j.

**TP53 target gene score.** Genes downregulated in *TP53*-multihit compared to *TP53*-WT cells (defined as per 'differential expression analysis' section above; related to Extended Data Fig. 4b) and p53 targets positively regulated from ref. 77 were used to compute a *TP53*-target gene score presented in Extended Data Fig. 5k.

## Prognostic signatures and Cox-regression survival models

**LSC signature score.** The 17-gene LSC17 gene set was retrieved from ref. 31. For each sample, the LSC17 score was defined as the linear combination of gene expression values weighted by their respective regression coefficients.

To identify *TP53*-sAML LSC signatures from our TARGET single-cell dataset, two different approaches were used. First, differentially expressed genes were identified as overexpressed in all Lin⁻CD34⁺ *TP53*-multihit cells regardless of their transcriptional classification (p53-all-cells) versus MF, HD and *TP53*-WT preleukemic cells; this gene set consists of 29 genes (Supplementary Table 4a). For the second approach, the same analysis was performed, but *TP53*-multihit cells transcriptionally defined as LSCs (falling in the LSC-like cluster; Fig. 2a, middle) were specifically selected; this gene-set is comprised of 51 genes (p53LSC; Supplementary Table 4a).

Next, lasso cox regression with tenfold cross-validation implemented in the glmnet R package (version 4.1-1) was used to identify p53-all-cells and p53-LSC genes that were associated with survival and to estimate their respective regression coefficients[86]. Specifically, Harrel's concordance measure (C-index) was used to assess the performance of each fitted model during cross-validation. The best model was defined as the fitted model with the highest C-index. Subsequently, the coefficient for each gene estimated using the best model was used to compute the gene signature scores. Only genes with nonzero coefficient values were included in the final gene set. In total, 9 and 44 genes were retained from the p53-all-cells and p53-LSC gene sets, respectively. For each sample, the gene signature score for each gene set was defined as the linear combination of gene expression values

weighted by their respective regression coefficients[31,86]. The list of p53-LSC and p53-all-cells gene signatures is provided in Supplementary Table 4b.

**Survival analysis.** For each gene expression signature, patients were first split using the median gene expression signature score. This resulted in two groups of patients, namely patients with high expression scores (greater than or equal to the median) and patients with low expression scores (lower than the median), exemplified in Extended Data Fig. 6a,b.

The Cox proportional hazards regression model implemented by the survival R package (version 3.5−5) was fitted to estimate the hazard ratio associated with each feature. The log-rank test was used to test the differences between survival curves. The features analyzed here were LSC17, p53-all-cells and p53-LSC signatures. Patients with low gene expression signature scores (below median) and patients with *TP53* WT status were specified as the reference groups in the model. Kaplan−Meier curves were plotted using the survminer R package (version 0.4.9) to visualize the probability of survival and sample size at a respective time interval.

### In vitro assays

**Short-term liquid culture experiments.** For short-term liquid culture differentiation experiments (Fig. 3j and Extended Data Fig. 7h,i), single cells from different Lineage-CD34+ HSPC populations (HSC: CD34+CD38-CD45RA-CD90+, MPP: CD34+CD38-CD45RA-CD90-, LMPP: CD34+CD38-CD45RA+ and more committed progenitors CD34+CD38+) were directly sorted into a 96-well tissue culture plate containing 100 µl of differentiation media: StemSpan (StemCell Technologies, 09650), 1% penicillin+streptomycin, 20% BIT9500 (StemCell Technologies, 9500), 10 ng ml$^{-1}$ SCF (Peprotech, 300-07), 10 ng ml$^{-1}$ FLT3L (Peprotech, 300-19), 10 ng ml$^{-1}$ TPO (Peprotech, 300-18-10), 5 ng ml$^{-1}$ IL3 (Peprotech, 200-03), 10 ng ml$^{-1}$ G-CSF (Peprotech, 300-23), 10 ng ml$^{-1}$ GM-CSF (Peprotech, 300-03), 1 IU ml$^{-1}$ EPO (Janssen, erythropoietin alpha, clinical grade) and 10 ng ml$^{-1}$ IL6 (Peprotech, 200-06).

For all liquid culture experiments, 50 µl of fresh 1× differentiation media was added on day 4. Readout was performed by flow cytometry after 12 d of culture using the antibodies detailed in Supplementary Table 7c (combination indicated as Panel D).

**LTC-IC assay.** Fifty cells from each Lin-CD34+ population (HSC; MPP; LMPP and CD38+) and donor type (HDs, MF and *TP53*-sAML) were sorted in triplicate. Cells were resuspended in 100 µl of MyeloCult (StemCell Technologies, H5150) supplemented with hydrocortisone (10$^{-6}$ M; StemCell Technologies, 74142) and plated into an irradiated supportive stromal cell layer (5,000 SI/SI cells and 5,000 M2-10B4 cells per well) in a 96-well tissue-culture plate coated with Collagen type I (Corning, 354236).

The medium was changed weekly and after 6 weeks of culture, cells were washed in IMDM + 20% fetal calf serum (FCS) and plated into 1.4 ml of cytokine-rich methylcellulose (StemCell Technologies, MethoCult H4435). Colonies were scored 14 days later under an inverted microscope, and each colony was classified according to its morphology as BFU-E (Burst-forming unit erythroid), CFU-G (Colony Forming Unit granulocyte), CFU-GM (granulocyte-macrophage), CFU-M (macrophage) or CFU-GEMM (granulocyte, erythrocyte, macrophage and megakaryocyte). Selected colonies were used for cytospin and genotyping as outlined below.

**LTC-IC colony genotyping.** LTC-IC colonies were picked from methylcellulose media, washed, resuspended in 10 µl of PBS and transferred to individual wells in a 96-well PCR plate. 15 µl of lysis buffer (Triton X-100 0.4%, Qiagen Protease 0.1 AU ml$^{-1}$) was added to each well, and samples were incubated at 56 °C for 10 min and 72 °C for 20 min. A 3 µl aliquot from each lysate was used as input to generate a targeted and Illumina-compatible library for colony genotyping. The preparation

of single-cell genotyping libraries involves three PCR steps. In the first PCR step, target-specific primers spanning each mutation of interest are used for amplification (Supplementary Table 6a); in the second PCR step, nested target-specific primers (Supplementary Table 6b) attached to universal CS1/CS2 adapters (forward adapter− CS1: ACACTGACGACATGGTTCTACA; reverse adapter−CS2: TACG-GTAGCAGAGACTTGGTCT) further enrich for target regions; and in the third PCR step, Illumina-compatible adapters containing sample-specific barcodes are used to generate sequencing libraries.

**TP53 knockdown and differentiation of human CD34+ cells.** shRNA sequence for p53 knockdown has been previously cloned into the lentiviral vector pRRLsin-PGK-eGFP-WPRE and validated[87]. Primary human CD34+ cells from patients with MPN (Supplementary Table 1) were infected twice with scramble (shCTL) or shTP53 with a multiplicity of infection of 15 and sorted 48 h later on CD34 and GFP expression. Cells were cultured in serum-free medium with a cocktail of human recombinant cytokines containing EPO (1 IU ml$^{-1}$, Amgen), FLT3-L (10 ng ml$^{-1}$, Celldex Therapeutics), G-CSF (20 ng ml$^{-1}$, Pfizer), IL-6 (10 ng ml$^{-1}$, Miltenyi), GM-CSF (5 ng ml$^{-1}$, Peprotech), IL-3 (10 ng ml$^{-1}$, Miltenyi), TPO (10 ng ml$^{-1}$, Kirin Brewery) and SCF (25 ng ml$^{-1}$, Biovitrum AB).

On day 12 of the culture, cells were stained with the antibodies detailed in Supplementary Table 7c (combination indicated as Panel C). DAPI was used for dead cell exclusion before acquisition on a FACSCanto II (BD Biosciences) instrument and on a BD FACS Diva software (version 8.0.2). Analysis of FACS data was performed using Kaluza (version 2.1, Beckman Coulter) software.

### Quantitative real-time PCR in shRNA experiments

In *TP53* knockdown experiments, RNA from either CD34+ cells sorted after transduction or bulk cells at day 12 of culture was extracted using Direct-Zol RNA MicroPrep Kit (Zymo Research) and reverse transcription was performed with SuperScript Vilo cDNA Synthesis Kit (Invitrogen). Quantitative RT−PCR was performed on a 7,500 real-time PCR Machine using SYBR-Green PCR Master Mix (Applied Biosystems). Expression levels were normalized to *PPIA* (housekeeping gene). Primers used are listed in Supplementary Table 6c.

### Xenotransplantation

Purified CD34+ cells from AML patients were transplanted via retroorbital vein injection in sublethally irradiated (1.5 Gy) NOD.CB17-*Prkdcscid IL2rgtm1*/Bcgen mice (B-NDG, Envigo) (female, 8 weeks old, *n* = 1 for IF0131, *n* = 3 for GR001). All experiments were approved by the French National Ethical Committee on Animal Care (2020-007-23589). Blood cell counts were performed monthly by submandibular sampling of mice with blood chimerism assessed by flow cytometry using hCD34, hCD45 and mCD45 antibodies (Supplementary Table 7b, PDX PB panel). The following endpoints were applied: >50% of human blast cells in the blood, abnormalities of blood cell count (hemoglobin <7 g dl$^{-1}$, platelets <150 × 10$^9$ l$^{-1}$ or white blood cells >60 × 10$^9$ l$^{-1}$), altered general conditions or >15% of weight loss. At sacrifice, BM was stained with the antibodies listed in Supplementary Table 7b (PDX BM panel) and HSPC fractions were sorted on an Influx Cell sorter (BD Biosciences).

### Evaluation of cell morphology

Cell morphology from PDX models (Extended Data Fig. 3d) and in vitro LTC-IC cultures (Extended Data Fig. 7f) was assessed after cytospin of 50−100,000 cells onto a glass slide (5 min at 500 r.p.m.) and May−Grünwald Giemsa staining, according to standard protocols. Images were obtained using an AxioPhot microscope (Zeiss).

### Mouse bone marrow chimeras and ethical approval

*Trp53*$^{tm2Tyj}$ *Commd10*$^{Tg(Vav1-icre)A2Kio}$ or *Trp53*$^{tm2Tyj}$ Tg$^{(Tal1-cre/ERT)42-056Jrg}$ C57/BL6 mice (hereafter referred to as Vav-iCre *Trp53*$^{R172H/+}$ or SCL-CreER$^T$ *Trp53*$^{R172H/+}$, respectively) and C57/BL6 WT mice used for BM

chimera experiments were bred and maintained in accordance to UK and France Home Office regulations. All experiments carried out were performed under Project License P2FF90EE8 approved by the University of Oxford Animal Welfare and Ethical Review Body or under Project License no. 2020-007-23589, approved by the French National Ethical Committee on Animal Care. *Trp53*[tm2Tyj] (ref. [88]), *Commd10*[Tg(Vav1-icre)A2Kio] (ref. [89]; Jackson Laboratory, 008610) and Tg[(Tal1-cre/ERT)42-056Jrg] (ref. [90]) have been previously described.

For in vivo experiments, two different chimera settings were used. For the first setting (Fig. 5a), 1 million BM cells from Vav-iCre *Trp53*[R172H/+] CD45.1 mice and 1 million BM CD45.2 WT cells from competitor mice were transplanted intravenously into lethally irradiated (10 Gy total body irradiation, split dose) congenic CD45.2 mice. Male and female recipient CD45.2 mice were used and were 6–8 weeks old at transplantation, while male and female CD45.1 experimental BM donors were 5–6 weeks old at the time of BM collection. For the second setting (Fig. 5h), 0.9 million BM cells from *Trp53*[LSL-R172H/+] CD45.2 mice (two males for two independent experiments, 8 and 13 weeks old) and 2.1 million BM CD45.1 WT competitor mice (four males for two independent experiments, 11 and 17 weeks old) were transplanted intravenously into lethally irradiated (9.5 Gy total body irradiation) congenic CD45.2 mice (females, 8 weeks old) and *Trp53* mutation was induced 4 weeks after transplantation by tamoxifen (gavage 200 mg kg$^{-1}$, Sigma-Aldrich) during 4 d, followed by tamoxifen feeding during 2 weeks (Ssniff Diet). In each cohort, a selection of mice was injected intraperitoneally with three rounds of six injections, each of 200 μg poly(I:C) (first setting) or 100 μg poly(I:C) (second setting; GE Healthcare, 27-4732-01) or placebo (PBS1X). Alternatively, Vav-iCre *Trp53*[R172H/+] mice were injected with three rounds of eight injections, each of 35 μg LPS from *Escherichia Coli* O111:B4 (LPS; L4391-1MG and L5293-2ML; Sigma-Aldrich).

Poly(I:C) and LPS were administered during weeks 6–8, 10–12 and 14–16 (setting 1), or during weeks 7–8, 11–12 and 15–16 (setting 2) post-transplantation. Within each round, injections were spaced one or two days apart. Blood cell counts and analysis of PB chimerism along with mature lymphoid and myeloid populations were performed every 2–4 weeks by submandibular sampling of mice, while BM chimerism and HSPC populations were analyzed 18–20 weeks after transplantation. The antibodies used are detailed in Supplementary Table 7d. For dead cell exclusion, 7-AAD (Sigma-Aldrich) or DAPI (BD Biosciences) were used. FACS analyses were carried out on BD Fortessa or BD Fortessa X20 (BD Biosciences) and profiles were later analyzed using FlowJo (version 10.1, BD Biosciences) or Kaluza (version 2.1, Beckman Coulter) software.

### LSK apoptosis and cell cycle
BM LSK cells (setting 2) were stained with Annexin-V and DAPI in Annexin V binding buffer 1X (BD Biosciences) for apoptosis analysis. BM LSK cell cycle was assessed by flow cytometry using Ki-67 and DAPI staining, after fixation and permeabilization (BD Cytofix/Cytoperm and Permeabilization Buffer Plus, BD Biosciences).

### M-FISH
Fifty CD45.1 (*Trp53*[R172H/+]) or CD45.2 (WT) LSK (Lin⁻Sca1⁺c-Kit⁺) cells from poly(I:C)-treated and control recipient mice were sorted and cultured for 1 week into Complete X-vivo15 media (BE-04-418Q, Lonza) supplemented with 10% FCS (Sigma-Aldrich, F9665), 0.1 mM 2-mercaptoethanol (Gibco, 21985023), 1% penicillin-streptomycin (PAA laboratories), 2 ng ml$^{-1}$ mouse stem cell factor (mSCF; PeproTech, 250-03), 10 ng ml$^{-1}$ mouse granulocyte–monocyte colony-stimulating factor (mGM–CSF; Immunex), 5 ng ml$^{-1}$ human thrombopoietin (hTPO; PeproTech, 300-18-10), 10 ng ml$^{-1}$ human granulocyte colony-stimulating factor (hG-CSF; Neopogen) 5 ng ml$^{-1}$ human FLT3 ligand (hFL; Immunex, 300-19), 5 ng ml$^{-1}$ mouse interleukin 3 (mIL-3; PeproTech, 213-13). Cells were cultured at 37 °C 5% $CO_2$. On day 7 of culture, metaphase spreads were collected following synchronization with Colcemid (KaryoMAX; Thermo Fisher Scientific, 11519876) 50 ng ml$^{-1}$, for 3 h at 37 °C. The cells were then incubated with KCl 75 mM for 15 min at 37 °C and spun down. Following this, the cells were fixed in a methanol-acetic acid and then dropped onto glass slides.

M-FISH was performed with the 21XMouse-Multicolor FISH probe kit (Metasystem Probes, D-0425-060-DI), following the manufacturer's instructions. For microscopy analysis, slides were mounted in Vectashield Antifade Mounting Medium with DAPI (2BScientific, H-1200). Images were acquired and analyzed using Leica Cytovision software (v7.3.1), on an Olympus BX-51 epifluorescence microscope equipped with a JAI CVM4⁺ progressive-scan 24 fps B&W fluorescence CCD camera. All cells were karyotyped, excluding metaphases severely damaged for technical reasons.

The analysis of the M-FISH hybridized cells was blinded. The cells on each slide were scored for the presence of structural aberrations (translocations, and/or derivative chromosomes and fragments) and/or numerical abnormalities. The presence of more than 40 chromosomes per cell was considered a numerical abnormality, except for cases where it could clearly be attributed to the presence of adjacent metaphases. Chromosome counts lower than 40 were not scored as numerical abnormalities for the impossibility to rule out technical issues (that is, metaphases bursting at the hypotonic step). We scored as follows: translocations and presence of one chromosome plus one or more extra chromosomal fragment(s)/derivative(s) as 'structural abnormalities' (except for sex chromosomes); presence of two chromosomes (or one in case of sex chromosomes) plus one or more extra chromosomal fragment(s)/derivative(s) as 'partial chromosome gains'; two chromosomes (or one in case of sex chromosomes) plus one or more extra chromosomes as 'whole chromosome gains'; two chromosomes plus two chromosomes with at least five different chromosomes present in number = $4n$ as 'tetraploidy or sub-tetraploidy'. Counts of numbers of karyotypic aberrations per cell were performed scoring every type of event occurring on one chromosome as a single event (that is, presence of four chromosomes is counted as one aberration).

### IFNγ ELISA assay
WT mice were injected intraperitoneally with a single dose of 200 μg poly(I:C) and spleens were collected from injected mice and non-treated controls 4 h after injection. Spleens were processed into a single-cell suspension in 200 μl PBS, spun down at 500 g for 5 min and supernatant was collected and used as spleen serum. IFNγ levels were assessed using mouse IFNγ Quantikine ELISA assay (R&D Systems, MIF00) following the manufacturer's instructions. Optical densities of 450 nm and 540 nm were determined using Clariostar microplate reader (BMG Labtech).

### Statistical analysis
Statistical analyses are detailed in figure legends (Figs. 2–6 and Extended Data Figs. 4–10) and performed using GraphPad Prism software (7 or later version) or R (version 3.6.1 and 4.0.5) software. The number of independent experiments, donors and replicates for each experiment are detailed in figure legends.

### Reporting summary
Further information on research design is available in the Nature Portfolio Reporting Summary linked to this article.

## Data availability
Subsets of the single-cell genotyping and RNA-sequencing data were part of a previous study[16] (GSE105454). Raw single-cell genotyping sequencing data from this study are deposited at the Sequence Reads Achieve (SRA; https://www.ncbi.nlm.nih.gov/sra) under the accession number PRJNA930152. Processed single-cell genotyping

data is available at https://zenodo.org/record/8060602 (https://doi.org/10.5281/zenodo.8060602) (metadata_MPNAMLp53_with_index_genotype.revised.txt; columns 'Genotype_curated', 'Genotype_labels' and 'genotype.classification'). Raw and processed (counts matrix) single-cell RNA-sequencing data from this study are deposited at the Gene Expression Omnibus (GEO; https://www.ncbi.nlm.nih.gov/geo/) under the accession number GSE226340. Single-cell dataset from this study is also available as an interactive Shiny app (https://wenweixiong.shinyapps.io/TP53_MPN_AML_Single_Cell_Atlas/). Raw and processed SNP array data are available at https://zenodo.org/record/8073857 (https://doi.org/10.5281/zenodo.8073857). Requests for material(s) should be addressed and will be fulfilled by corresponding authors. Source data are provided with this paper.

## Code availability

Scripts to reproduce data preprocessing and all figures are available on GitHub (https://github.com/albarmeira/p53-transformation) and source data to reproduce the scripts is available at: https://zenodo.org/record/8038152 (https://doi.org/10.5281/zenodo.8038152; data preprocessing) and https://zenodo.org/record/8060602 (https://doi.org/10.5281/zenodo.8060602; source data for figures).

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

## Acknowledgements

We are grateful to patients and donors as without their generosity, this study would not have been possible. We also thank S. Knapper, clinical

study teams and other investigators involved in supporting sample collection, and King's Health Partners Biobank for providing access to samples. We thank Z. Ren, T. Denaes and H. Duparc for their help with mouse experiments and S.-A. Clark for help with sorting. We also thank C. Soblechero for the help with computational analysis. This work was funded by a Medical Research Council (MRC) Senior Clinical Fellowship (MR/L006340/1 to A.J.M.), CRUK Senior Cancer Research Fellowship (C42639/A26988 to A.J.M.), Cancer Research UK (CRUK) DPhil Prize Studentship (C5255/A20936 to A.R.-M.), Sir Henry Wellcome Postdoctoral Fellowship from the Wellcome Trust (222800/Z/21/Z to A.R.-M.), British Spanish Society Scholarship (to A.R.-M.), MRC Confidence in Concept/MLSTF Grant (MC_PC_19049 to A.R.-M. and A.J.M.), the MRC Molecular Haematology Unit core award (MC_UU_12009/5 to A.J.M. and S.E.W.J.), Emergence Cancéropôle Ile de France 2017 (to I.A.-D.), Association pour la Recherche contre le Cancer 2018 (to I.A.-D.), Gustave Roussy Siric-Socrate 2019 (to I.A.-D.), Institut National du Cancer INCA-PLBIO 2020 (to I.A.-D.) and La Ligue Nationale Contre le Cancer (Equipe labellisée 2019 and 2022 to I.P. and I.A.-D.). A.L.C. was supported by Paris University (MENRT grant), J.R.C. by a CRUK Senior Cancer Research Fellowship (RCCSCF-Nov21\100004) and S.E.W.J. by the Swedish Research Council, Torsten Söderberg Foundation and Knut and Alice Wallenberg Foundation. F.G. was supported by grants from the association 'Chalon sur Saône-Tulipes contre le cancer' and the Centre de Ressources Biologiques Ferdinand Cabanne. The authors would like to acknowledge the flow cytometry facility at the MRC Weatherall Institute of Molecular Medicine (WIMM), which is supported by the MRC Human Immunology Unit; MRC Molecular Haematology Unit (MC_UU_12009); National Institute for Health Research (NIHR), Oxford Biomedical Research Centre (BRC); Kay Kendall Leukemia Fund (KKL1057), John Fell Fund (131/030 and 101/517), the EPA fund (CF182 and CF170) and by the MRC WIMM Strategic Alliance awards (G0902418 and MC_UU_12025). The authors acknowledge the contributions of N. Ashley at the MRC Weatherall Institute of Molecular Medicine (WIMM) Single Cell Facility and MRC-funded Oxford Consortium for Single-Cell Biology (MR/M00919X/1). The authors would also like to acknowledge the contribution of the WIMM Sequencing Facility, supported by the MRC Human Immunology Unit and by the EPA fund (CF268), the Gustave Roussy flow cytometry platform and mouse facility. We also thank the Oxford Genomics Centre at the Wellcome Centre for Human Genetics (funded by Wellcome Trust grant reference 203141/Z/16/Z) for the generation and initial processing of the OmniExpress SNP array data. The results published here are in whole or part based upon data generated by the TCGA Research Network (https://www.cancer.gov/tcga) and the BeatAML team. The views expressed are those of the authors and not necessarily those of the National Health Service (NHS), the NIHR or the Department of Health. The funders had no role in study design, data collection and analysis, decision to publish or preparation of the manuscript.

## Author contributions

A.R.M. conceived the project, designed and performed experiments, performed computational analysis, analyzed data and wrote the manuscript. R.N., A.L.C., H.R., J.O.S., E.L., A.P., J.C. and B.W. designed and performed experiments and analyzed data. S.W., G.W. and W.W.K. performed computational analysis. J.E.M. collected primary samples and clinical and bibliographic data. C.D. provided clinical data. C.B. and M.B. analyzed SNP-array data. J.O.S., C.B., N.S., F.G., F.P., I.P., M.D. and C.H. provided patient samples, clinical data and scientific input. C.M. and H.G. analyzed and provided patients and PDX biological data (NGS and SNP-array). A.H. performed and analyzed patients' targeted sequencing. D.M. and L.S.M. performed M-FISH experiments and analyzed M-FISH data. W.V., J.R.C., S.E.W.J., B.P. and S.T. provided scientific input and conceptualization. S.T. supervised computational analysis. I.A.-D. conceived and supervised the project, analyzed data and wrote the manuscript. A.J.M. conceived and supervised the project, provided clinical care and wrote the manuscript.

## Competing interests

A.R.M. and A.J.M. are authors on a patent related to TARGET-seq (US Patent App. 17/038,548). A patent relating to the TARGET-seq technique is licensed to Alethiomics, a spin-out company from the University of Oxford with equity owned by B.P. and A.J.M. and research funding to B.P. and A.J.M. The other authors declare no competing interests.

## Additional information

**Extended data** is available for this paper at https://doi.org/10.1038/s41588-023-01480-1.

**Correspondence and requests for materials** should be addressed to Alba Rodriguez-Meira, Iléana Antony-Debré or Adam J. Mead.

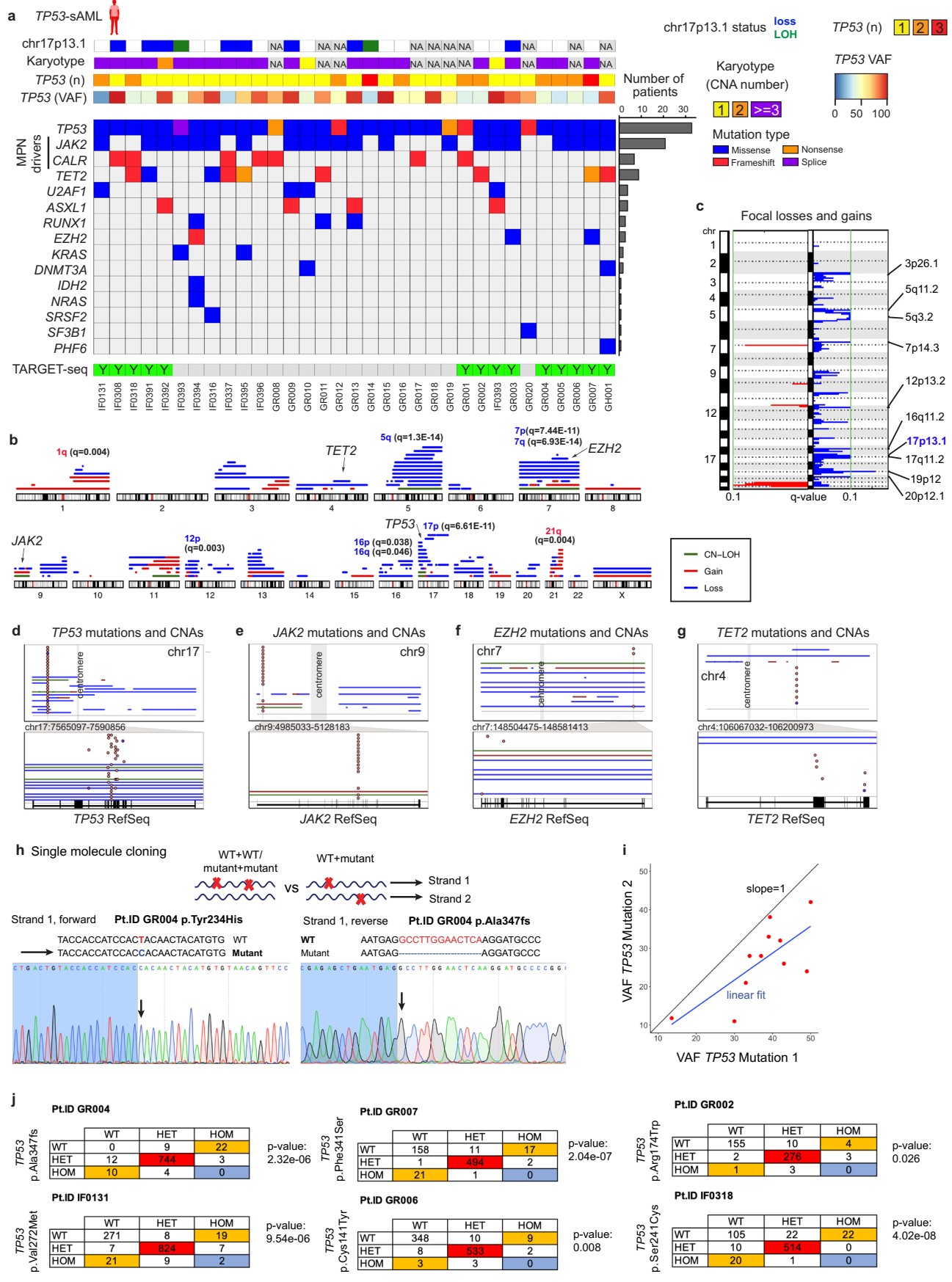

**Extended Data Fig. 1 | See next page for caption.**

**Extended Data Fig. 1 | Genetic landscape of *TP53*-sAML. a**, Mutations, CNAs, *TP53* VAF and allelic status identified in a cohort of 33 *TP53*-sAML patients by bulk sequencing. The barplot on the right indicates the frequency of each mutation in the cohort. The panel at the bottom indicates samples processed by TARGET-seq. **b-c**, Graphical representation of all CNAs identified by MoChA (**b**) and GISTIC analysis of recurrently lost (blue) and amplified (red) focal regions (**c**) in the same patients as in (**b**). In **b**, GISTIC q-values of arm-level gains (red) and loses (blue) are indicated for each chromosome arm. In **c**, *TP53* chromosomal location is indicated in blue (17p13.1). **d-g**, Summary of CNA events spanning recurrently mutated genes *TP53* (**d**), *JAK2* (**e**), *EZH2* (**f**) and *TET2* (**g**), with evidence of deletion or loss of heterozygosity in the single-cell phylogenies computed in Extended Data Fig. 2b-o. For each gene, top panel shows a whole chromosome view and the bottom one, the gene-level view and RefSeq track. Points indicate the location of each point mutation and solid lines indicate CNA status (blue:loss; red:gain; green:LOH). **h**, Sanger sequencing of single-molecule patient-derived *TP53* cDNA showing mutually exclusive alleles in the same cDNA molecule. **i**, VAF of *TP53* mutations in patients in which at least two *TP53* mutations were detected. Blue line represents the linear fit of the points, which deviates from the indicated slope that would be expected if mutations were on the same allele. When more than 2 mutations were present, the 2 with the highest VAF were analyzed. **j**, Contingency table of *TP53* zygosity status in single cells from patients carrying two *TP53* mutations. Double-mutant heterozygous cells are colored in red, mutually exclusive WT/homozygous or homozygous/WT genotypes in orange and homozygous/homozygous cells, in blue. "p" indicates exact one-sided binomial test p-value.

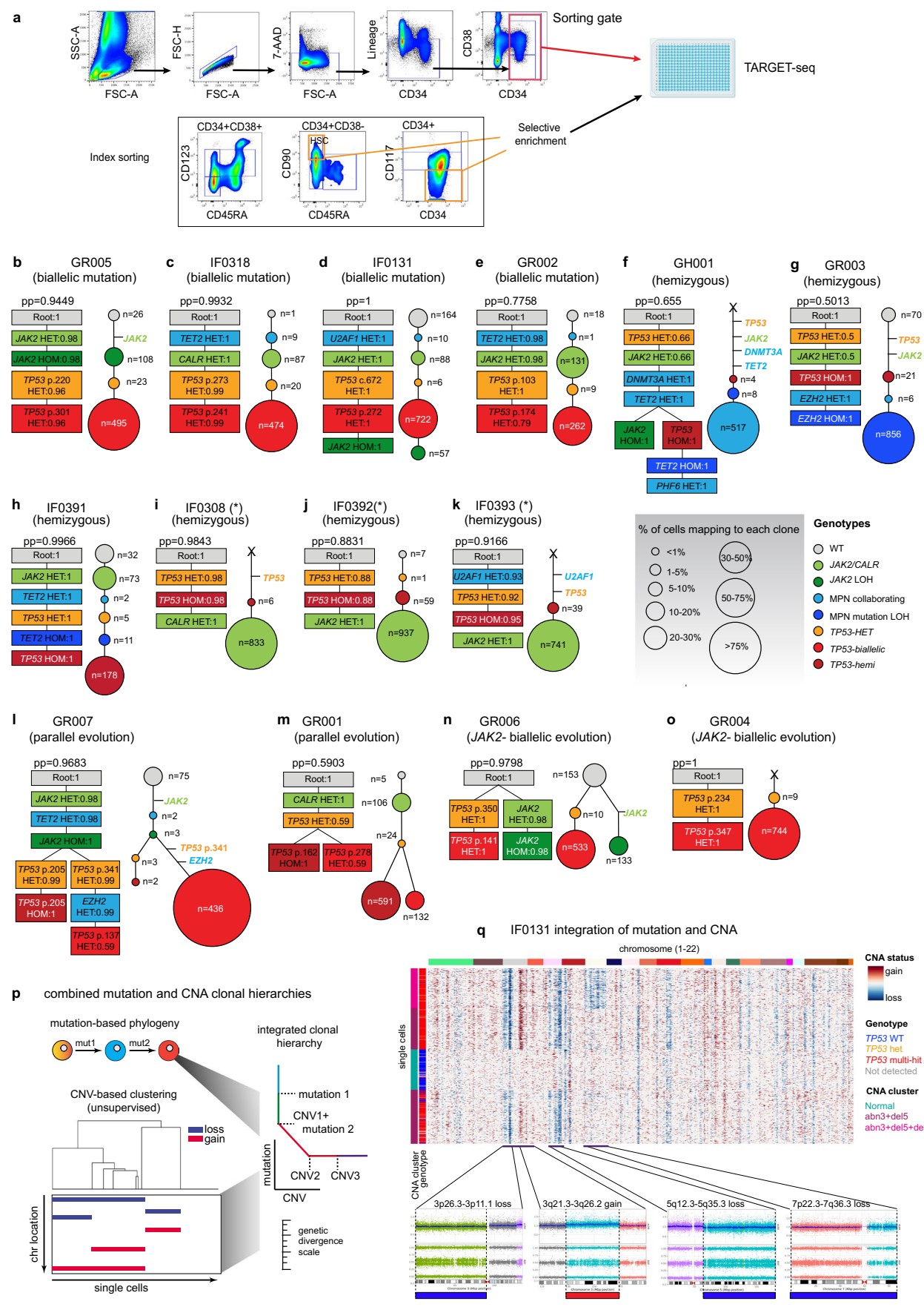

**Extended Data Fig. 2 | See next page for caption.**

**Extended Data Fig. 2 | TARGET-seq sorting strategy and phylogenetic reconstruction of clonal hierarchies in *TP53*-sAML patients using a Bayesian model. a**, Sorting strategy for TARGET-seq: Lineage⁻CD34⁺ cells were sorted into 384-well plates for subsequent library preparation. Selective enrichment of immunophenotypically defined populations (HSC: CD38⁻CD90⁺CD45RA⁻; CD117⁻) is indicated with orange boxes. **b-o**, In each panel, corresponding to a different patient sample, the phylogenetic tree computed using SCITE is shown on the left and the number of cells mapping to each clone on the right. "pp" indicates the posterior probability of each consensus mutation tree, and the probability of each genotype transition is indicated inside each square for each mutation. The size of the circles is proportional to the size of each clone and is colored according to the genotype indicated. The number of cells mapping to each clone is indicated in each circle and the type of *TP53* clonal evolution (biallelic mutation, hemizygous, parallel or *JAK2*-negative) below each patient's ID. (*) indicates patients for which the high clonality of the sample prevented the faithful reconstruction of the order of mutation acquisition. Horizontal lines indicate mutation acquisition where none of the experimentally-detected clones matched that particular combination of mutations and therefore the order of mutations cannot be reliably determined. Due to selective enrichment of certain subpopulations of cells (**a**), the numbers of cells assigned to each subclone in this figure is not necessarily representative of overall clonal burden, and early clones are likely over-represented due to selective enrichment of preleukemic HSCs. In contrast, the relative subclone percentages displayed in Fig. 1 for the same patients have been corrected according to each populations' frequency in the Lin⁻CD34⁺ compartment. **p**, Schematic representation of the strategy to reconstruct integrated clonal hierarchies based on single-cell TARGET-seq genotyping and inferCNV transcriptomic-based CNAs. **q**, Representative example of combined mutation and CNA hierarchies for patient IF0131, in which three cytogenetically-distinct subclones were detected. Corresponding cytogenetic lesions detected at the bulk level through high-density SNP arrays are shown in the bottom panels.

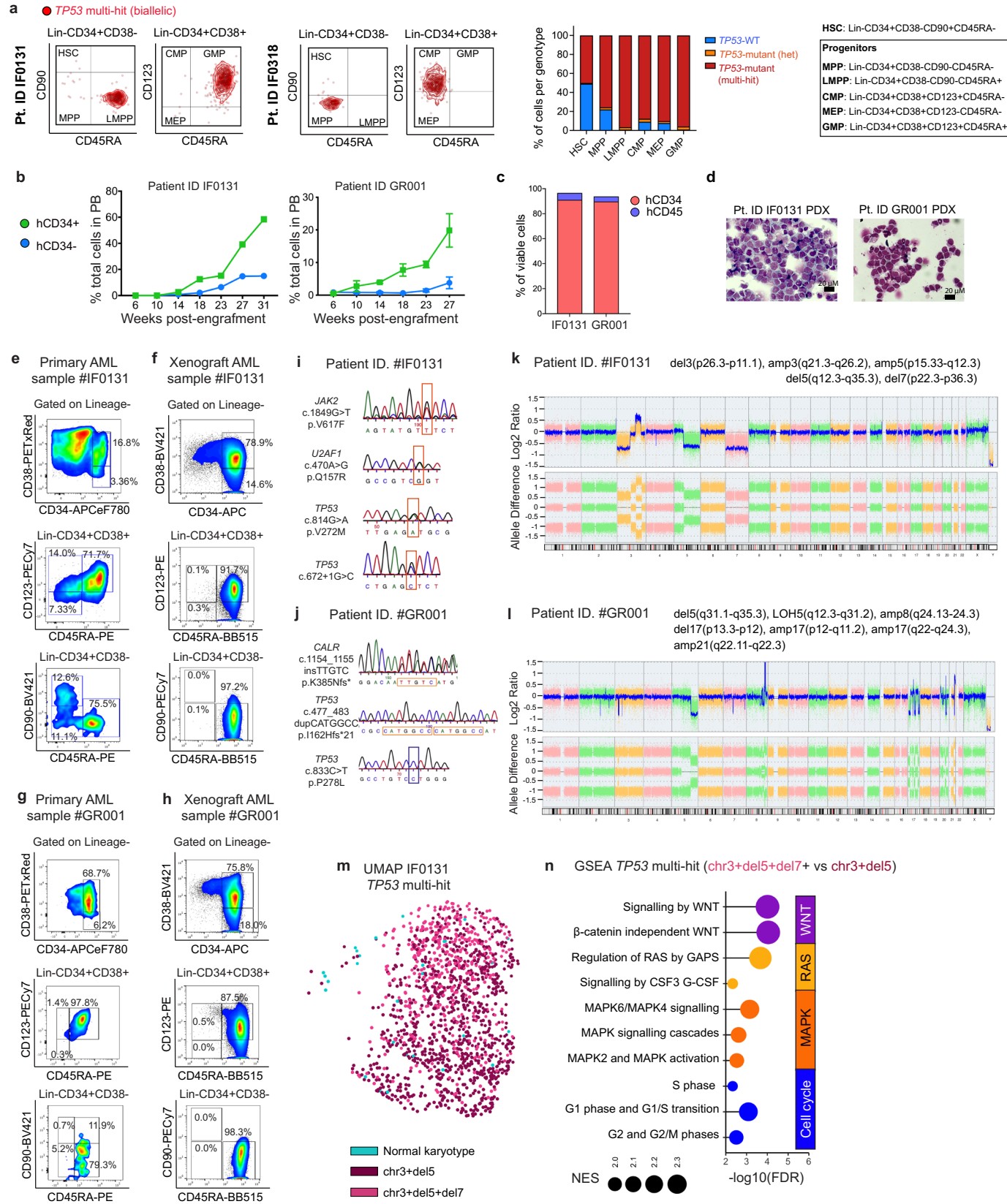

**Extended Data Fig. 3 | See next page for caption.**

**Extended Data Fig. 3 | *TP53*-sAML xenograft characteristics. a**, Integration of index sorting and single cell genotyping of *TP53* multi-hit HSPC from two representative patients (left) and quantification of genotypes across HSPC populations (right). **b**, Serial readouts of human chimerism based on hCD34 and hCD45 expression in mouse PB for IF0131 (n = 1) and GR001 (n = 3, mean ± s.e.m. indicated). **c**, Proportion of hCD45 and hCD34-positive cells in total bone marrow (BM) from each PDX sample. **d**, Representative images from BM blasts isolated from PDX models **e-f**, Representative HSPC flow cytometry profiles of patient IF0131 PB mononuclear cells (MNCs) **(e)** and BM engrafted cells in immunodeficient mice at 31 weeks post transplantation **(f)**. **g-h**, Representative HSCP flow cytometry profiles of patient GR001 PB MNCs **(g)** and BM engrafted cells in immunodeficient mice at 27 weeks post-transplantation **(h)**. **i-l**, Mutations **(i,j)** and CNAs **(k,l)** detected in sorted LMPPs (Lin⁻CD34⁺CD38⁻CD90⁻CD45RA⁺) from indicated PDX samples **(f,h)**. Boxes indicate location of each mutation (orange for mutant allele and blue, for WT) **m**, UMAP representation of *TP53* multi-hit cells from patient IF0131; cells are colored according to their CNA status as in Fig. 1g. **n**, GSEA analysis of cytogenetically distinct subclones in patient IF0131. Pathways enriched in *TP53* multi-hit abn3+del5+monosomy7 versus *TP53* multi-hit abn3+del5 Lin⁻CD34⁺ are shown and colored according to pathway's functional category. NES: Normalized Enrichment Score. FDR: False Discovery Rate.

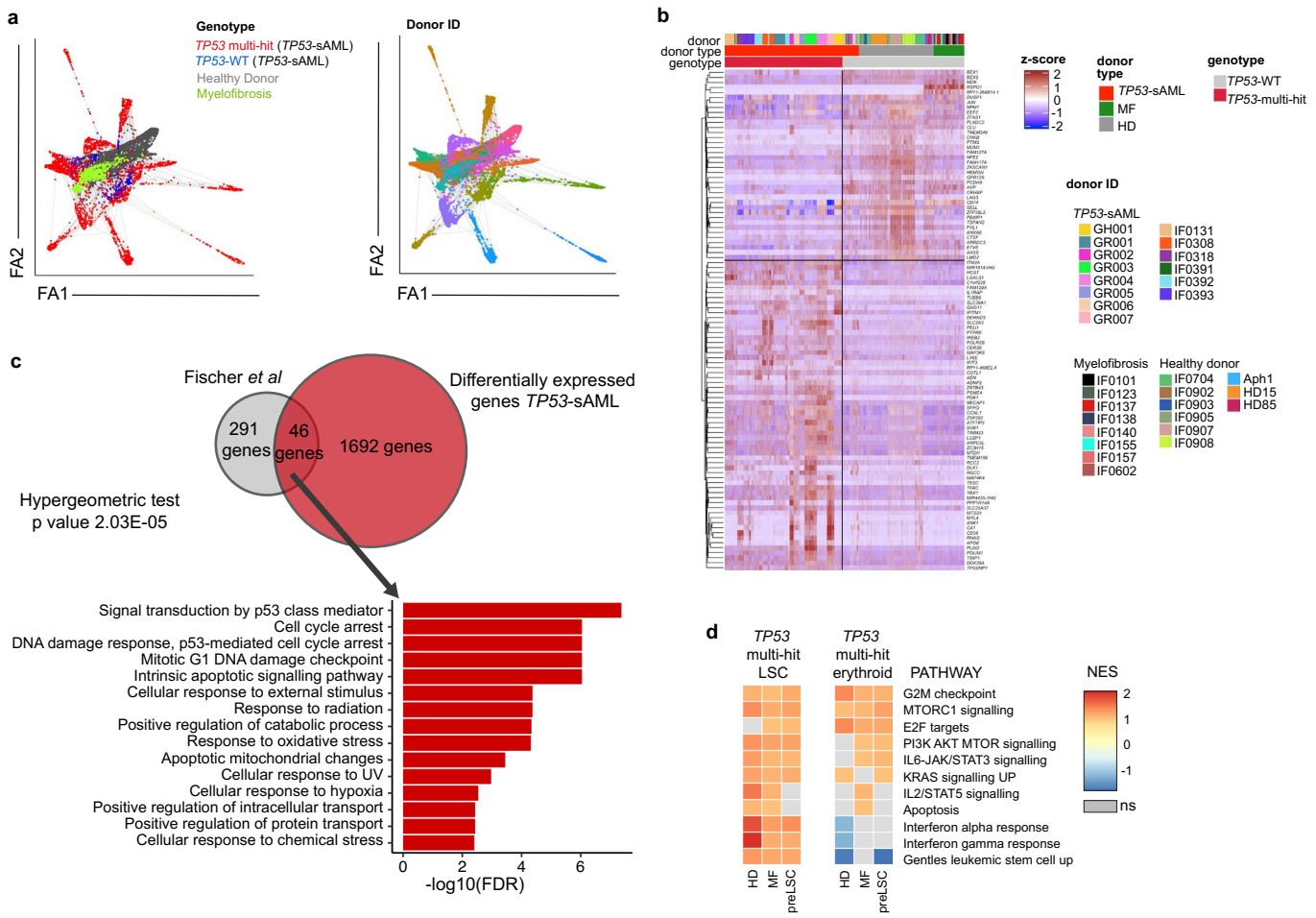

**Extended Data Fig. 4 | Single cell transcriptomic analysis of healthy donor, MF and *TP53*-sAML HSPCs. a**, Force-atlas representation of 17517 cells from healthy donor (n = 9), MF (n = 8) and *TP53*-sAML patients (n = 14; preleukemic: *TP53*-WT, leukemic: *TP53* multi-hit) according to patient type (left) or donor (right). **b**, Heatmap of the top 100 differentially expressed genes identified between *TP53* multi-hit cells and preleukemic (*TP53*-WT; "preLSCs"), MF and healthy donor (HD) cells. The type of donor, donor ID and *TP53* genotype is indicated on the top of the heatmap for each single cell. **c**, Venn diagram of the overlapping p53-target genes from Fischer et al[85] and differentially expressed genes between *TP53*-multi-hit cells and *TP53*-WT cells. "p" indicates hypergeometric test p-value. **d**, GSEA analysis of Lin⁻CD34⁺ *TP53* multi-hit LSC or erythroid-biased cells (Related to Fig. 2a) from *TP53*-sAML patients, compared to Lin⁻CD34⁺ healthy donor, MF and preLSCs. Heatmap indicates NES from selected genesets with FDR q-value < 0.25. NES: Normalized Enrichment Score; FDR: False Discovery Rate. NS: non-significant.

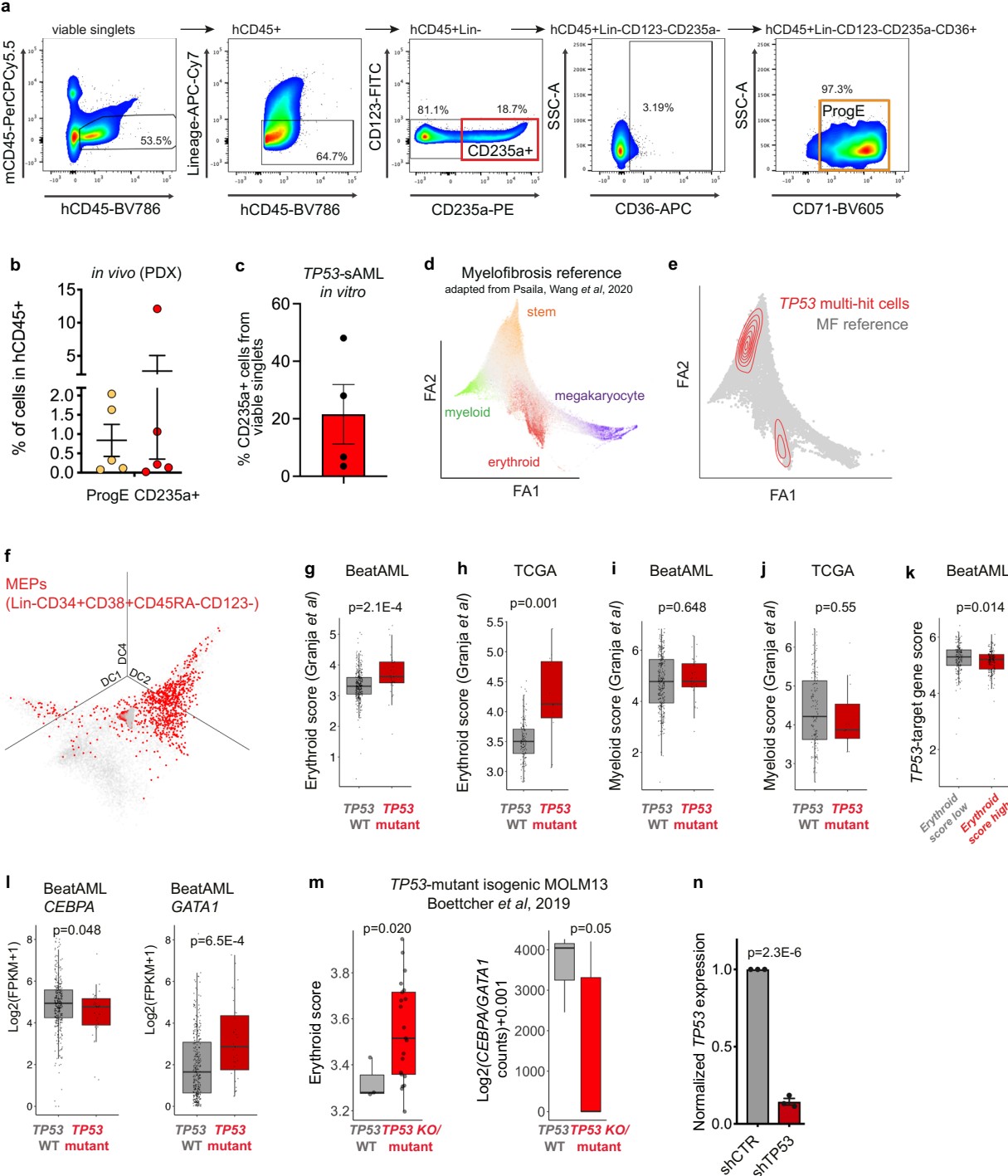

**Extended Data Fig. 5 | See next page for caption.**

**Extended Data Fig. 5 | Aberrant erythroid differentiation in *TP53* mutant AML. a-b**, Analysis of erythroid populations in *TP53*-sAML PDX models. Gating strategy used to identify CD253a+ and erythroid progenitor cells ("ProgE") **(a)** and percentage of each erythroid population in hCD45+ bone marrow cells from PDX models **(b)**. n = 5, bars indicate mean ± s.e.m. **c**, Percentage of cells expressing erythroid markers after culturing CD34 + *TP53*-sAML cells in conditions promoting myelo-erythroid differentiation *in vitro*. n = 4, bars indicate mean ± s.e.m. **d-e**, Force Atlas representation of a CD34⁺ myelofibrosis (MF) atlas (**d**; Psaila et al, 2020) and latent-semantic index projection of *TP53* multi-hit cells from *TP53*-sAML patients into the MF cellular hierarchy **(e)**. **f**, Projection of immunophenotypically-defined MEPs into a diffusion map of the single cells from all 14 *TP53*-sAML patients (as in Fig.2a). **g-j**, Expression of a comprehensive erythroid **(g,h)** and myeloid **(i,j)** gene score derived from a human haematopoietic atlas (Granja et al, 2019) in AML patients from the BeatAML dataset (g,i) and TCGA (h,j) stratified by *TP53* mutational status

(BeatAML: n = 329 *TP53*-WT and n = 31 *TP53*-mutant; TCGA: n = 140 *TP53*-WT, n = 11 *TP53*-mutant). **k**, Expression of a *TP53*-target gene score using the same p53 target genes as in Extended Data Fig. 4c in patients with high (above median) and low (below median) erythroid scores. **l**, *GATA1/CEBPA* gene expression in AML patients from the BeatAML dataset stratified by *TP53* mutational status. In (**g-l**), boxplots represent median, first and third quartiles, and whiskers correspond to 1.5 times the interquartile range. "p" indicates two-sided Wilcoxon rank sum test p-values. **m**, Erythroid score (left) and *CEBPA/GATA1* gene expression ratios (right) in MOLM13 *TP53*-mutant isogenic cell lines (Boettcher et al, 2019). Boxplots represent median, first and third quartiles, and whiskers correspond to 1.5 times the interquartile range. "p" indicates two-tailed unpaired *t*-test p-value. **n**, Fold-change *TP53* expression in CD34⁺GFP⁺ MPN primary cells following transduction with a lentiviral shRNA vector targeting *TP53* compared to a scramble control (shCTR). n = 3 patients, 3 independent experiments. Barplot indicates mean ± s.e.m. and "p", two-tailed unpaired t-test p-value.

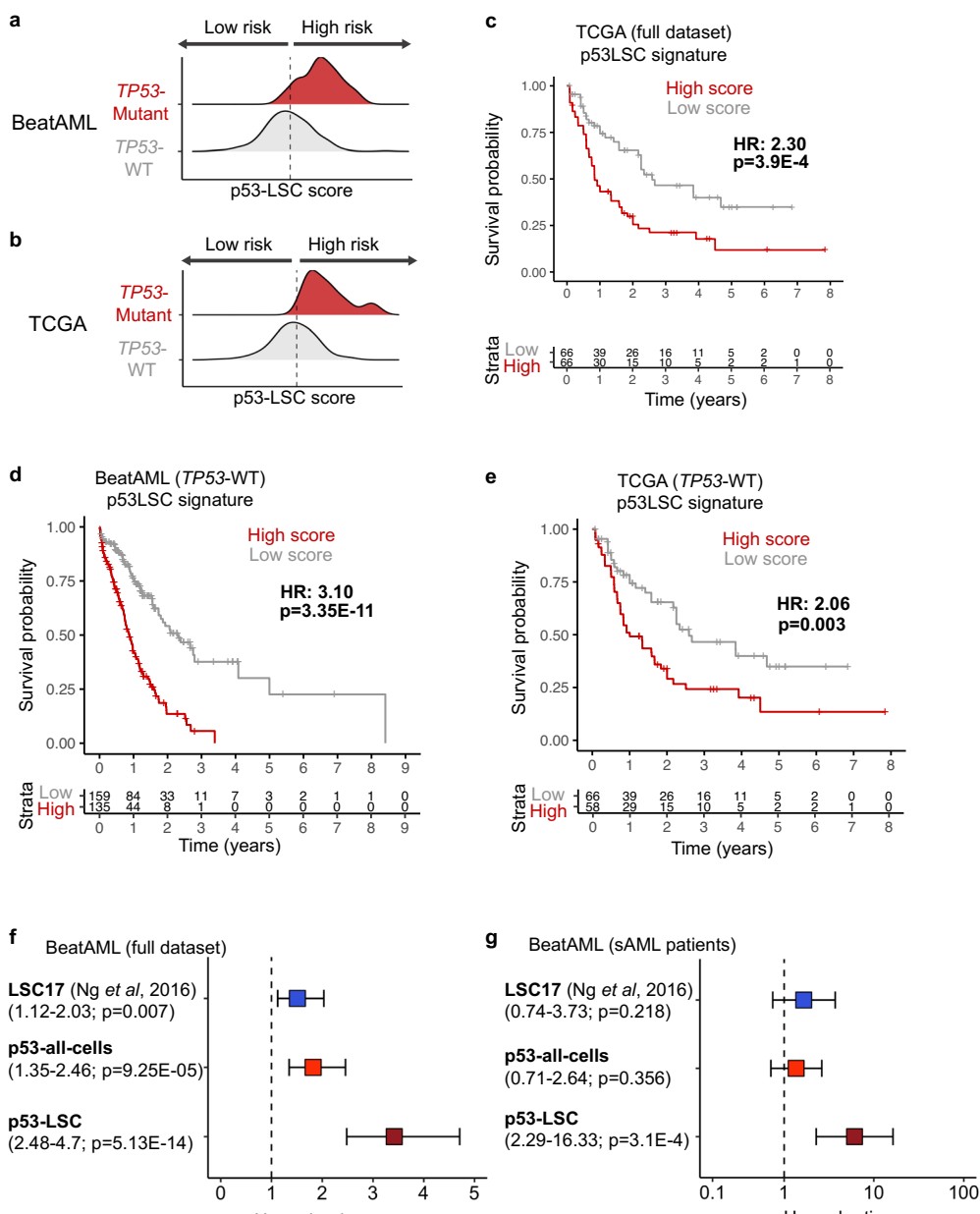

**Extended Data Fig. 6 | Validation of p53-LSC signature score in two independent cohorts. a-b**, Distribution of p53-LSC scores in BeatAML (**a**) and TCGA (**b**) cohorts stratified by *TP53* mutational status. **c-e**, Kaplan-Meier analysis of *de novo* AML patients from the full TCGA AML dataset (n = 132) (**c**), *TP53*-WT AML patients from BeatAML (n = 294) (**d**) and *TP53*-WT AML patients from TCGA (n = 124) (**e**) stratified according to high or low p53 LSC signature score. **f-g**, Hazard ratio of all AML patients (n = 322) (**f**) or secondary AML patients (n = 49) (**g**) from the BeatAML cohort using LSC17 score (Ng et al, 2016), p53-all-cells score (derived from all *TP53*-mutant sAML cells) and p53-LSC signature score (derived from transcriptionally-defined LSCs; related to Fig. 2a). Boxes represent hazard ratios and lower and upper bounds of error bars, 95% confidence intervals. Genes used for each score are listed in Supplementary Table 4. "p" indicates log-rank test p-value.

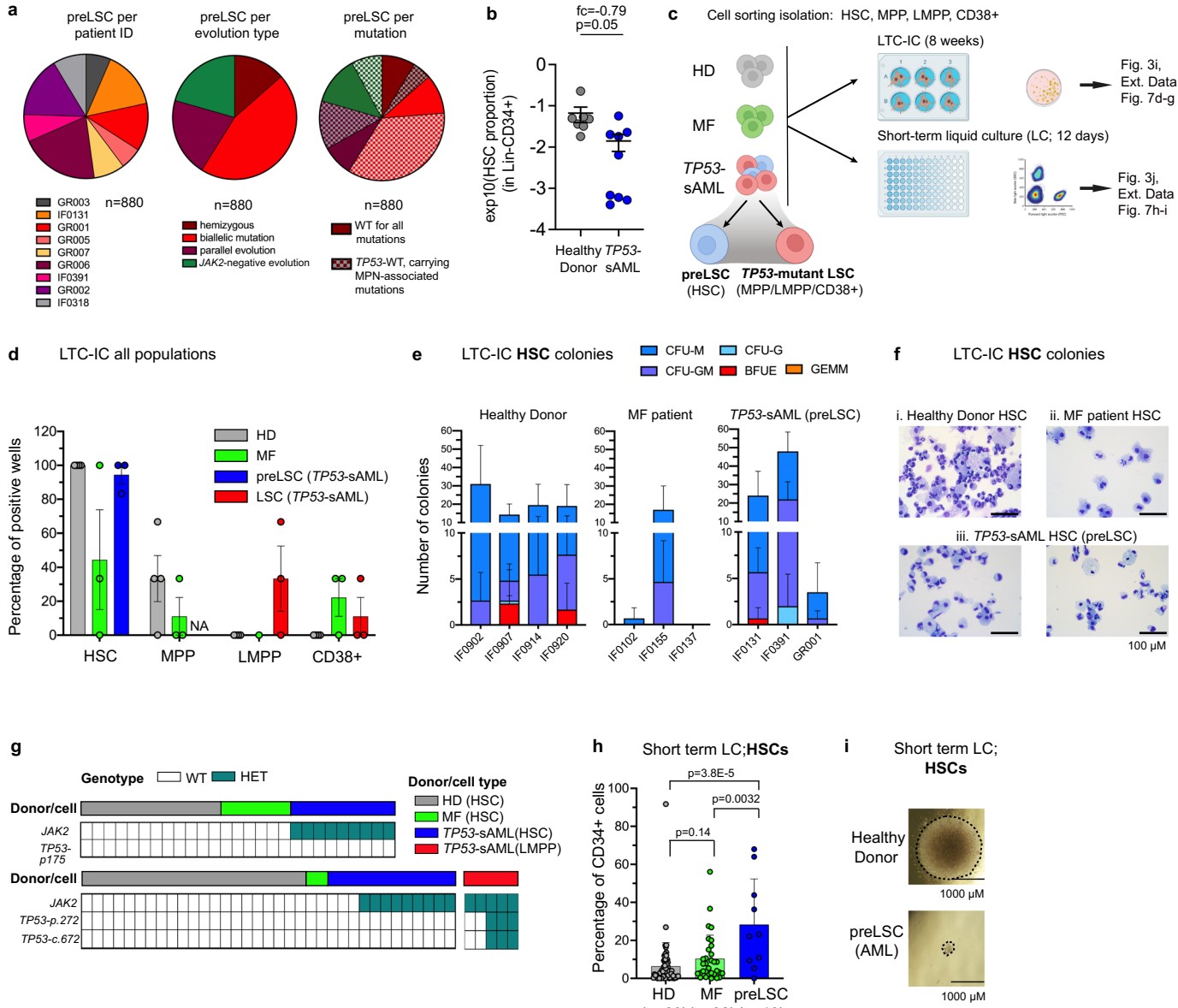

**Extended Data Fig. 7 | *In vitro* assessment of self-renewal and differentiation potential in preleukemic cells from *TP53*-sAML patients. a**, Donor, type of clonal evolution and genotype of the 880 preLSCs identified. **b**, Proportion of HSCs in mobilized PB or BM from healthy donors (n = 7) and *TP53*-sAML patients (n = 9) in which preLSCs were detected. Graph shows mean ± s.e.m, "p" indicates two-tailed Student t-test p-value and "fc", fold-change. **c**, Schematic representation of Lin⁻CD34⁺ cell fractions isolated and *in vitro* assays performed. *TP53*-sAML patient samples used (n = 3: IF0131, IF0391, GR001) were known to have *TP53*-WT preleukemic stem cells (preLSC) in the HSC compartment (Related to Fig. 3c). **d-e**, Long term culture-initiating cell *in vitro* assay. Percentage of positive wells in each immunophenotypic population **(d)** and clonogenic output **(e)** from healthy donor (HD, n = 4), MF (n = 3) and preLSCs from *TP53*-sAML

(n = 3). Barplot indicates mean ± s.e.m. from 2 independent experiments. **f**, Representative cytospin images of HSC-derived colonies from the same patient groups as in **(d-e)**. **g**, Genotyping of HSC and LMPP-derived colonies from the same LTC-IC assay as in **(d-f)**, demonstrating absence of *TP53* mutations in HSC-derived colonies, contrary to LMPPs. **h**, Percentage of CD34⁺ cells from healthy donor (HD, n = 4), MF (n = 3) and preLSCs from *TP53*-sAML (n = 2) after 12 days of liquid culture in conditions promoting hematopoietic differentiation. Barplot indicates mean ± s.e.m from 3 independent experiments, and "p", two-tailed Student t-test p-value. **i**, Representative image of liquid culture HSC-derived colonies for healthy donor and *TP53*-sAML preLSCs, from the same experiment as in **(h)**.

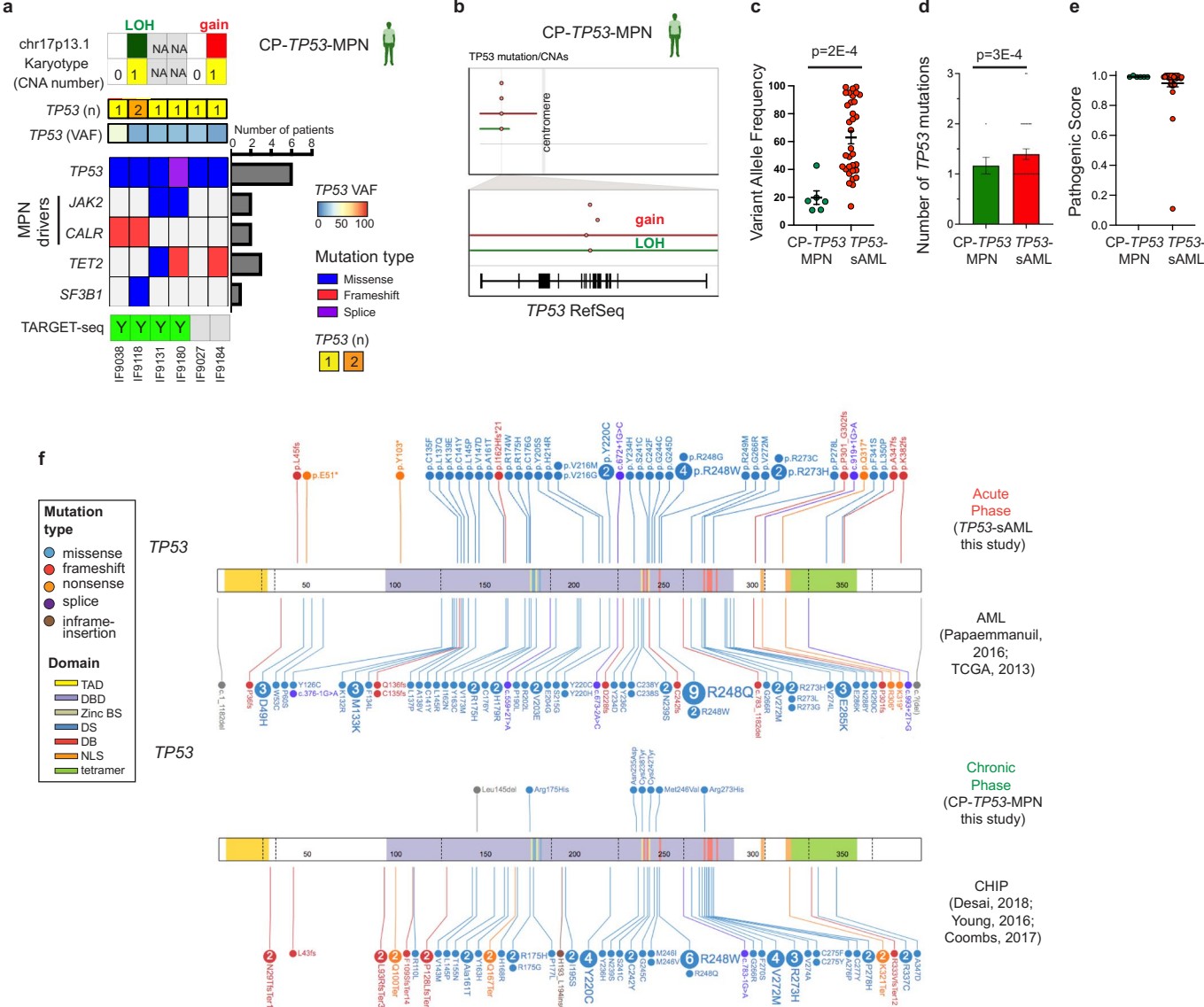

**Extended Data Fig. 8 | Genetic landscape of chronic phase *TP53*-mutant MPN. a**, Point mutations and cytogenetic abnormalities identified in a cohort of 6 CP *TP53*-MPN patients with no evidence of clinical transformation after 4.43 years [2.62-5.94] median follow-up. The number of patients in which each gene is mutated is shown on the barplot on the right and patients processed for TARGET-seq analysis are indicated below the heatmap. **b**, Summary of CNA events in chr17 and *TP53* gene in the 2 CP *TP53*-MPN patients with detectable CNAs. The top panel shows a whole chromosome view and the bottom one, the gene-level view

and RefSeq track. Points indicate the location of each point mutation and solid horizontal lines indicate CNA status. **c-e**, Comparison of variant allele frequency **(c)**, number of *TP53* mutations **(d)** and pathogenic scores **(e)** of *TP53* variants identified in CP-*TP53*-MPN (n = 6) and *TP53*-sAML patients (n = 33). Mean ± s.e.m. is shown; "p" indicates two-tailed Mann-Whitney test p-value. **f**, Location and mutation type stratified by patient group (chronic/acute phase) as compared to previously published CHIP and AML patient cohorts.

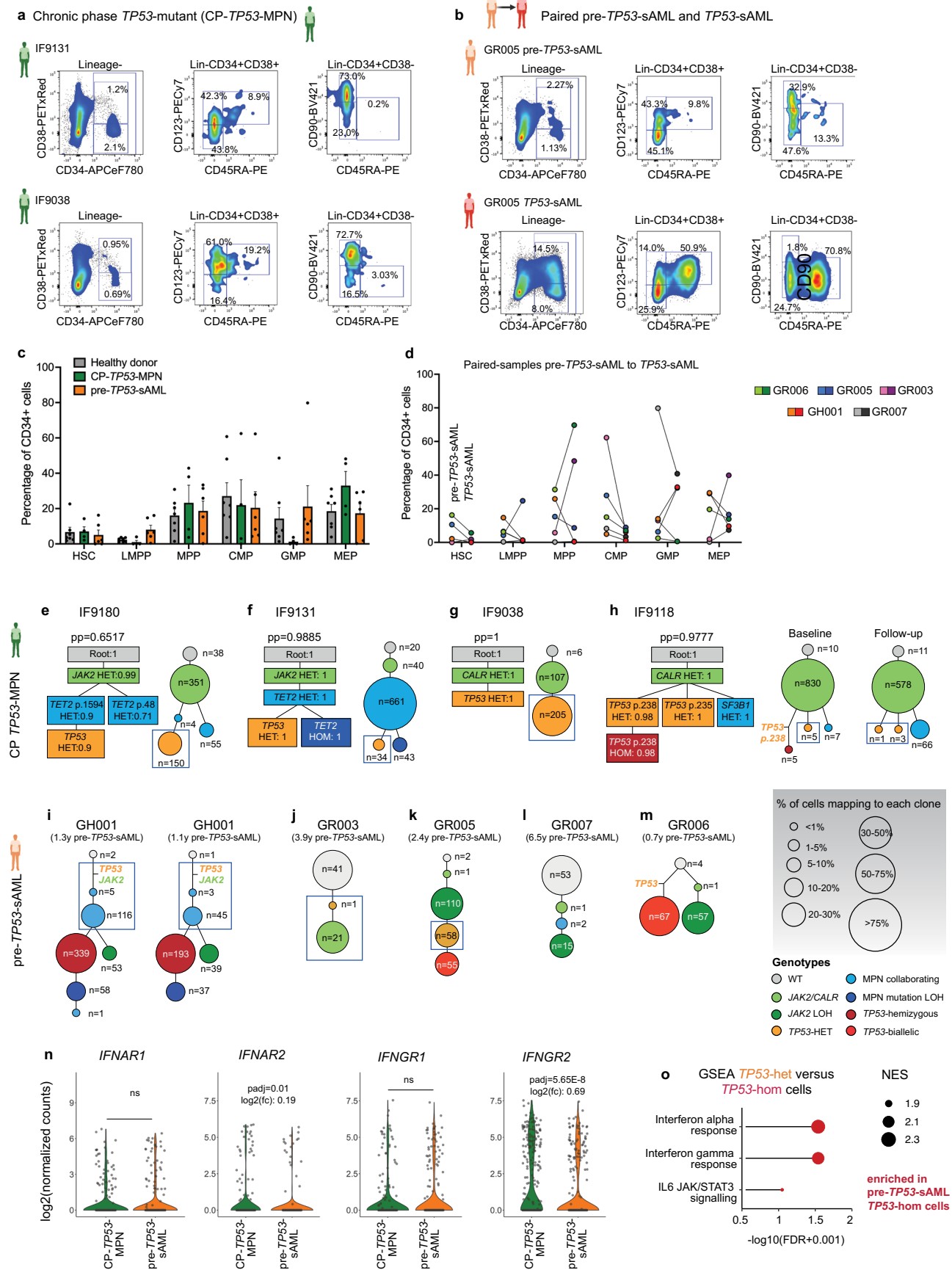

**Extended Data Fig. 9 | See next page for caption.**

**Extended Data Fig. 9 | Clonal evolution and molecular signatures of *TP53*-mutant patients at chronic phase. a-b**, Flow cytometry profiles of the Lin⁻CD34⁺ HSPC compartment in two CP *TP53*-MPN patients without evidence of clinical transformation (**a**) and in a representative paired chronic phase (**b**, up; pre-*TP53*-sAML) and acute phase (**b**, bottom; *TP53*-sAML) sample (Related to Fig. 4a). **c-d**, Percentage of immunophenotypic HSPC populations in normal donors (n = 8), CP *TP53*-MPN (n = 4) and pre-*TP53*-AML patients (n = 5) (**c**) or in the 5 paired pre-*TP53*-AML and *TP53*-AML samples (**d**). None of the population frequencies are significantly different (p < 0.05) between patient groups by multiple unpaired t-test analysis. In (c), barplot indicates mean ± s.e.m. **e-h**, Phylogenetic reconstruction of clonal hierarchies in CP *TP53*-MPN patients from single-cell TARGET-seq genotyping data. In each panel, the phylogenetic tree computed using SCITE is shown on the left, and the number of cells mapping to each clone for each patient, on the right. "pp" indicates posterior probability or each consensus mutation tree, and the probability of each genotype transition is indicated in the square for each mutation. For patient IF9118 (**h**), baseline (left)

and 4 years of follow-up (right) samples are shown separately. **i-m**, Phylogenetic reconstruction of clonal hierarchies in pre-*TP53*-AML patients from single-cell TARGET-seq genotyping data (related to Extended Data Fig. 2). In panels (e-m), the size of the circles is proportional to each clone's size, and is colored according to the genotype indicated in the genotype key. Blue boxes indicate *TP53*-heterozygous clones used for the analysis presented in Fig. 4c-e. **n**, Expression of interferon receptors in *TP53*-heterozygous cells from CP *TP53*-MPN (n = 273 cells) and pre-*TP53*-sAML patients (n = 296 cells). "p-adj" indicates adjusted p-value from combined Fisher's exact test and Wilcoxon tests, calculated using Fisher's method and adjusted using Benjamini & Hochberg procedure; "fc" indicates fold-change (related to Fig. 4d,e). Violin plots indicate log2(counts) distributions and each point represents the expression value of a single-cell. **o**, GSEA of inflammatory pathways in *TP53*-mutant heterozygous (n = 284) and homozygous (n = 622) cells from patients GH001 and GR005 at the pre-*TP53*-sAML stage. NES: Normalized Enrichment Score. FDR: False Discovery Rate.

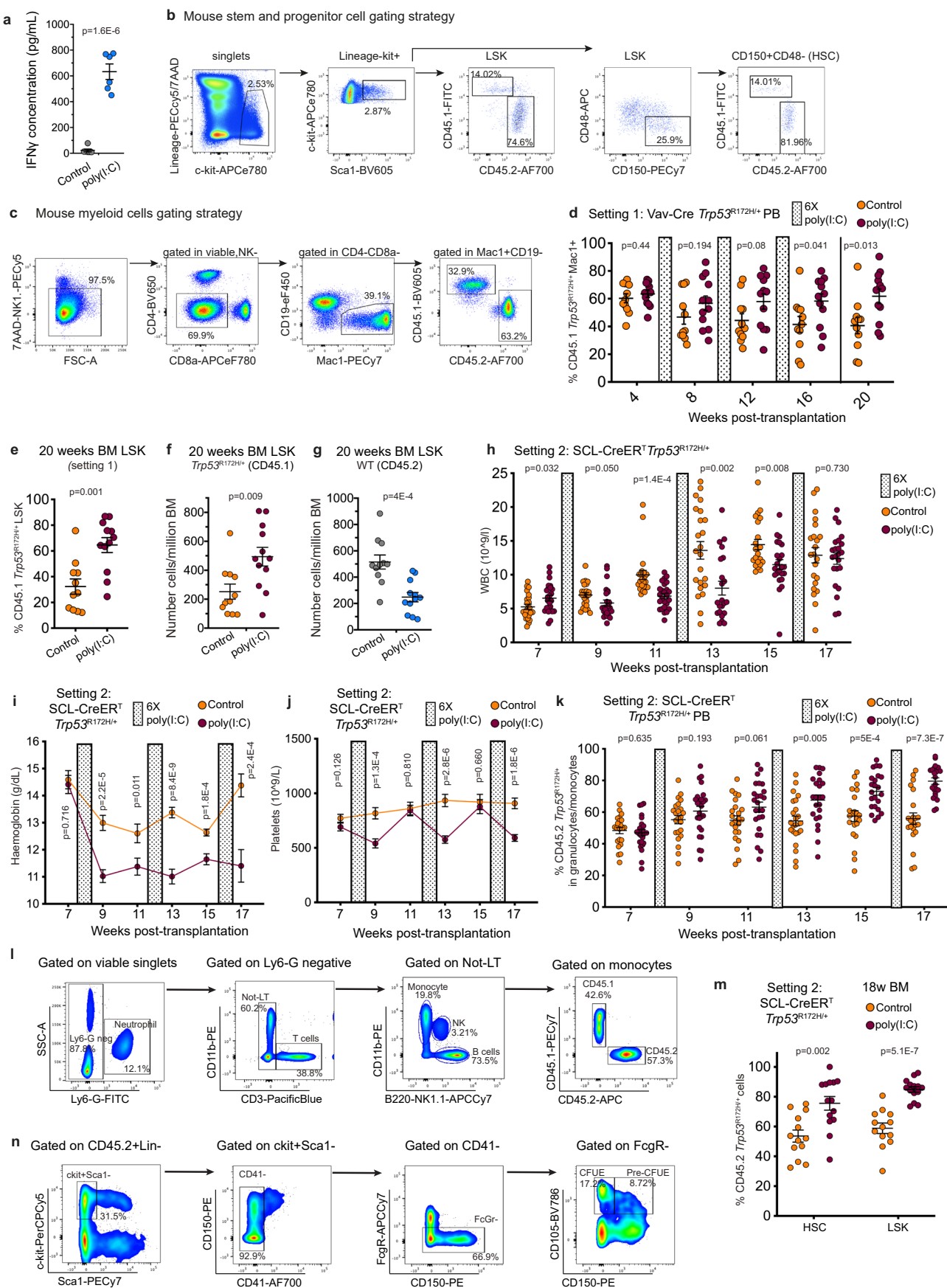

**Extended Data Fig. 10 | See next page for caption.**

**Extended Data Fig. 10 | Analysis of *Trp53*-mutant mice following inflammatory challenge. a**, IFNγ level in spleen serum 4 h after poly(I:C) injection. n = 6 mice per group from 2 independent experiments. Lines indicate mean ± s.e.m. and "p", two-tailed unpaired t-test p-value. **b-c**, Gating strategy for mouse chimaera experiments (Related to Fig. 5) used to quantify BM LSK and HSCs populations **(b)** and myeloid cells in the peripheral blood (PB) **(c)**. **d-g**, Analysis of WT:*Trp53$^{R172H/+}$* chimaera mice treated with 3 cycles of 6 poly(I:C) injections (related to model setting 1, Fig. 5a) with serial readouts of CD45.1 *Trp53$^{R172H/+}$* Mac1$^+$ PB cells **(d)**, percentage of CD45.1 *Trp53$^{R172H/+}$* BM LSK (Lin$^-$Sca-1$^+$ c-Kit$^+$) **(e)**, number of CD45.1 *Trp53$^{R172H/+}$* BM LSK **(f)** and CD45.2 WT BM LSK per million BM cells **(g)** 20 weeks post transplantation. n = 11-12 mice per group from 3 biological replicates in 2 independent experiments.

**h-k**, Analysis of WT:*Trp53$^{R172H/+}$* chimaera mice treated with 3 cycles of 6 poly(I:C) injections (related to model setting 2, Fig. 5h) with serial readouts of white blood cells **(h)**, hemoglobin **(i)** and platelet **(j)** counts measured every 2 weeks, and percentage of CD45.2 *Trp53$^{R172H/+}$* granulomonocytic (Ly6G and/or Mac1 + ) PB cells **(k)**. **l**, Gating strategy for granulomonocytic (neutrophils and monocytes) and lymphoid (T, NK and B cells) populations in WT:*Trp53$^{R172H/+}$* chimaera mice. **m**, Percentage of CD45.2 *Trp53$^{R172H/+}$* BM HSC and LSK at 18 weeks post transplantation. **n**, Gating strategy for CFUE and PreCFUE populations in WT:*Trp53$^{R172H/+}$* chimaera mice. n = 22 control, n = 23 poly(I:C) groups (h-k) or n = 13 control, n = 14 poly(I:C) groups (m) from 2 independent experiments. Bars indicate mean ± s.e.m. and "p", two-tailed unpaired t-test p-value.

# Reporting Summary

## Statistics

For all statistical analyses, confirm that the following items are present in the figure legend, table legend, main text, or Methods section.

| n/a | Confirmed | |
|---|---|---|
| ☐ | ☒ | The exact sample size (*n*) for each experimental group/condition, given as a discrete number and unit of measurement |
| ☐ | ☒ | A statement on whether measurements were taken from distinct samples or whether the same sample was measured repeatedly |
| ☐ | ☒ | The statistical test(s) used AND whether they are one- or two-sided *Only common tests should be described solely by name; describe more complex techniques in the Methods section.* |
| ☐ | ☒ | A description of all covariates tested |
| ☐ | ☒ | A description of any assumptions or corrections, such as tests of normality and adjustment for multiple comparisons |
| ☐ | ☒ | A full description of the statistical parameters including central tendency (e.g. means) or other basic estimates (e.g. regression coefficient) AND variation (e.g. standard deviation) or associated estimates of uncertainty (e.g. confidence intervals) |
| ☐ | ☒ | For null hypothesis testing, the test statistic (e.g. *F*, *t*, *r*) with confidence intervals, effect sizes, degrees of freedom and *P* value noted *Give P values as exact values whenever suitable.* |
| ☐ | ☒ | For Bayesian analysis, information on the choice of priors and Markov chain Monte Carlo settings |
| ☒ | ☐ | For hierarchical and complex designs, identification of the appropriate level for tests and full reporting of outcomes |
| ☒ | ☐ | Estimates of effect sizes (e.g. Cohen's *d*, Pearson's *r*), indicating how they were calculated |

*Our web collection on statistics for biologists contains articles on many of the points above.*

## Software and code

Policy information about availability of computer code

**Data collection**
- Single cell RNA-seq and genotyping libraries were generated using 3'-TARGET-seq (Rodriguez-Meira et al, 2019) at the University of Oxford. Libraries were sequenced in paired-end format on an Illumina HiSeq2000, HiSeq4000 or NextSeq. FASTQ files were generated using bcl2fastq (version 2.20) .
- Targeted myeloid-panel sequencing libraries were generated from bulk genomic DNA using a TruSeq Custom Amplicon panel (Illumina) or a Haloplex Target Enrichment System (Agilent technologies), and were sequenced on a MiSeq (Illumina) instrument.
- SNP-Array data was generated through hybridization of bulk genomic DNA to an Illumina Infinium OmniExpress v1.3 BeadChips Array and SNP-CGH CytoScan HD Array
- Flow cytometry data was collected using BD FACS Diva Software (version 8.0.2).
- Cells were sorted on an Influx (BD Biosciences), a BD Fusion I and BD Fusion II instruments (Becton Dickinson), a SH800S or MA900 (SONY) cell sorters.
- Images for cell morphology were obtained using an AxioPhot microscope (Zeiss).
- Images for M-FISH were obtained with an Olympus BX-51 epifluorescence microscope

Data collection methods are fully described in the manuscript.

**Data analysis**
- Bioinformatic analysis were performed with CentOS Linux 7, R (v4.0.5 and v3.6.1), Python 3.9.13.
Linux packages: bcl2fastq v2.20, Bedtools v2.27.1, BWA v0.7.17, FeatureCounts v1.4.5-p1, GATK v4.1.2.0, Picard v2.3.0, Samtools v.19, SCITE v2016b31, STAR v2.6.1d and v2.4.2a, TrimGalore v0.4.1.
R packages: bigmemory v4.5.36, Biobase v2.50.0, BiocGenerics v0.44.0, cccd v1.5, circlize v0.4.12, clusterProfiler v3.18.1, ComplexHeatmap v2.11.1, data.table v1.14.0, destiny v3.9.0, diffusionMap v1.2.0, DNAcopy v1.60.0, fgsea v1.16.0, GeneOverlap v1.26.0, GenomicRanges v1.42.0, GenomeInfoDb v1.34.9, GISTIC2 v2.023, ggplot2 v3.4.2, ggpubr v0.4.0, glmnet v4.1-1, gridExtra v2.3, igraph v1.2.6, IRanges v2.24.1,

irlba v2.3.3, inferCNV v1.0.4, limma v3.46.0, LinkedMatrix v1.4.0, Matrix v1.5-1, MatrixGenerics v1.2.1, matrixStats v0.58.0, Mocha WDL
pipeline v2021-01-20 (https://software.broadinstitute.org/software/mocha/mocha.20210120.wdl); pbapply v1.4-3, pheatmap v1.0.12, RANN
v2.6.1, RColorBrewer v1.1-2, Rcpp v1.0.7, SHAPEIT v4.1.3.
reshape2 v1.4.4, reticulate v1.22, Rtsne v0.15, S4Vectors v0.28.1, SingCellaR v1.2.0/v1.2.1, SingleCellExperiment v1.12.0, statmod v1.4.36,
SummarizedExperiment v1.20.0, survival v3.5-5, survminer v0.4.9, threejs v0.3.3, umap v0.2.7.0
Python packages: Velocyto v0.17.13
GSEA software version 4.0.3, http://software.broadinstitute.org/gsea
- Targeted myeloid-panel sequencing data : SOPHiA DDM® (Sophia Genetics) and an in-house software GRIO-Dx®.
- SNP Array data : Chromosome Analysis Suite (Affymetrix) v4.1
- M-FISH : Leica Cytovision software version 7.3.1
- Flow cytometry : Kaluza (version 2.1, Beckman Coulter) or FlowJo (version 10.1, BD Biosciences) softwares
- Statistical analysis : GraphPad Prism software (version 7 or later)
- Custom codes are available on https://github.com/albarmeira/p53-transformation

All methods for data analyses are fully described in the methods section of the manuscript.

For manuscripts utilizing custom algorithms or software that are central to the research but not yet described in published literature, software must be made available to editors and reviewers. We strongly encourage code deposition in a community repository (e.g. GitHub). See the Nature Portfolio guidelines for submitting code & software for further information.

## Data

Policy information about availability of data

All manuscripts must include a data availability statement. This statement should provide the following information, where applicable:
- Accession codes, unique identifiers, or web links for publicly available datasets
- A description of any restrictions on data availability
- For clinical datasets or third party data, please ensure that the statement adheres to our policy

Database/datasets used in this study:
- Database: COSMIC (Catalogue Of Somatic Mutations In Cancer, https://cancer.sanger.ac.uk/cosmic) , Pecan Portal, genome version GRCh37 (hg19), National
Cancer Institute (NIH) Genomic Data Commons, cBio Cancer Genomics Portal
- Datasets from the following papers : Papaemmanuil et al, N Engl J Med, 2016; Coombs et al, Cell Stem Cell, 2017;  Desai et al, Nat Med, 2016 ;  Young et al, Nat
Commun, 2016; Psaila et al, Mol Cell, 2020; van Galen et al, Cell, 2019; Fisher et al, Oncogene, 2017; Granja et al, Nature Biotechnology, 2019 ; Ng et al, Nature,
2016.
Two publicly available AML cohorts with genetic mutation and RNA-sequencing data: BeatAML (Tyner et al, Nature, 2018) and The Cancer Genome Atlas (TCGA)
(Ley et al, N Engl J Med, 2013). Subsets of single-cell genotyping and RNA-sequencing data were part of a previously published study (Rodriguez-Meira et al, Mol
Cell, 2019).
Data generated in the study : Single-cell genotyping and RNA-sequencing data generated from this study are publicly available in SRA and GEO with accession
numbers PRJNA930152 and GSE226340, respectively. The dataset generated in this paper is also available as an interactive vignette https://
wenweixiong.shinyapps.io/TP53_MPN_AML_Single_Cell_Atlas/.

## Research involving human participants, their data, or biological material

Policy information about studies with human participants or human data. See also policy information about sex, gender (identity/presentation), and sexual orientation and race, ethnicity and racism.

| Reporting on sex and gender | Sex and gender were not take into account to select the population of interest. |
| Reporting on race, ethnicity, or other socially relevant groupings | Socially relevant groupings were not take into account to select the population of interest. |
| Population characteristics | Patients samples were selected based on their pathology (patients with myeloproliferative neoplasms or acute myeloid leukemia secondary to a myelopoliferative neoplasm, or TP53-sAML) and on TP53 status (mutated or wild-type). Age-matched healthy donors were also used for the study. |
| Recruitment | Samples were collected as part of patients' routine clinical care through previously established research study approvals as detailed below. Patients and normal donors provided written informed consent in accordance with the Declaration of Helsinki for sample collection and use in research. |
| Ethics oversight | Samples were obtained from:<br>- Gustave Roussy (Villejuif, France) and Dijon Hospital (Dijon, France) with the agreement from the Inserm Institutional Review Board Ethical Committee (project C19-73, agreement 21-794, CODECOH n°DC-2020-4324).<br>- from the INForMeD Study (REC: 199833, 26 July 2016, University of Oxford).<br>Patients and healthy donors provided written informed consent in accordance with the Declaration of Helsinki for sample collection and use in research. |

Note that full information on the approval of the study protocol must also be provided in the manuscript.

# Field-specific reporting

Please select the one below that is the best fit for your research. If you are not sure, read the appropriate sections before making your selection.

☒ Life sciences ☐ Behavioural & social sciences ☐ Ecological, evolutionary & environmental sciences

For a reference copy of the document with all sections, see nature.com/documents/nr-reporting-summary-flat.pdf

# Life sciences study design

All studies must disclose on these points even when the disclosure is negative.

| | |
|---|---|
| Sample size | Sample size was determined based on similar studies in the field and availability of samples/numbers of cells per sample. |
| Data exclusions | Data which didn't meet quality control parameters (as detailed in Methods section) were excluded from the analysis. |
| Replication | In vitro and in vivo experiments were repeated to reach 3 biological replicates in at least 2 independent experiments. Attempts at replication were successful. Details on numbers of replicates are provided in the relevant legend and/or methods section. |
| Randomization | Patient samples were separated according to their diagnosis and TP53 mutational status, randomization was not appropriate. Mice were allocated randomly to control or treated groups. |
| Blinding | Blinding was not relevant for single cell data, as the information on mutational status was required for analysis. For mouse experiments, blinding was performed for analysis of FACS data with an anonymized identification number for each mouse. For M-FISH karyotype assessment, the operator was blinded to the genotype and treatment group. |

# Reporting for specific materials, systems and methods

We require information from authors about some types of materials, experimental systems and methods used in many studies. Here, indicate whether each material, system or method listed is relevant to your study. If you are not sure if a list item applies to your research, read the appropriate section before selecting a response.

## Materials & experimental systems

| n/a | Involved in the study |
|---|---|
| ☐ | ☒ Antibodies |
| ☒ | ☐ Eukaryotic cell lines |
| ☒ | ☐ Palaeontology and archaeology |
| ☐ | ☒ Animals and other organisms |
| ☒ | ☐ Clinical data |
| ☒ | ☐ Dual use research of concern |
| ☒ | ☐ Plants |

## Methods

| n/a | Involved in the study |
|---|---|
| ☒ | ☐ ChIP-seq |
| ☐ | ☒ Flow cytometry |
| ☒ | ☐ MRI-based neuroimaging |

## Antibodies

| | |
|---|---|
| Antibodies used | - Antibodies used for human HSPC sorting.<br>CD8-FITC (Lineage), BioLegend, Clone: RPA-T8, Cat#: 301006, 1/100<br>CD20-FITC (Lineage), BioLegend, Clone: 2H7, Cat#: 302304 , 1/150<br>CD66b-FITC (Lineage), BioLegend, Clone: G10F5, Cat#: 305104, 1/15<br>CD10-FITC (Lineage), BioLegend, Clone: HI10a, Cat#: 312208, 1/30<br>CD127-FITC (Lineage), eBioscience, Clone eBioRDR5, Cat#: 11-1278-42, 1/30<br>Human Hematopoietic Lineage Cocktail – FITC (Lineage), eBioscience, NA , Cat# 22-7778-72, 1/15<br>CD123-PECy7, BioLegend, Clone: 6H6, Cat#: 306010, 1/60<br>CD38-PETxRed, Invitrogen, Clone: HIT2, Cat#: MHCD3817, 1/21.5<br>CD90-BV421, BioLegend, Clone: 5E10, Cat#: 328122, 1/30<br>CD45RA-PE, eBioscience, Clone: HI100, Cat#: 12-0458-41, 1/150<br>CD34-APC-eF780, eBioscience, Clone: 4H11, Cat#: 47-0349-42, 1/150<br>CD34-PerCP/Cy5.5, BioLegend, Clone: 562, Cat# 343611, 1/100<br>CD90-PE, BioLegend, Clone: 5E10, Cat# 328109, 1/25<br>CD45RA-FITC, Invitrogen, Clone: MEM56, Cat# MHCD45RA01, 1/150<br>CD2-PE/Cy5, BioLegend, Clone: RPA-2.10, Cat# 300209, 1/300<br>CD3-PE/Cy5, BioLegend, Clone: HIT3a, Cat# 300310, 1/300<br>CD4-PE/Cy5 , BioLegend, Clone: RPA-T4, Cat# 300510, 1/160<br>CD8-PE/Cy5, BioLegend, Clone: RPA-T8, Cat# 301010, 1/300<br>CD10-PE/Cy5, BioLegend, Clone: HI10a, Cat# 312206, 1/80<br>CD11b-PE/Cy5, BioLegend, Clone: ICRF44, Cat# 301308, 1/160<br>CD14-PE/Cy5, Invitrogen, Clone: 61D3, Cat# 15-0149-41, 1/160 |

CD19-PE/Cy5, BioLegend, Clone: HIB19, Cat# AB_314240, 1/300
CD20-PE/Cy5, BioLegend, Clone: 2H7, Cat# AB_314256, 1/200
CD56-PE/Cy5, BD Biosciences, Clone: B159, Cat# 555517, 1/80
CD235a,b-PE/Cy5, BioLegend, Clone: HIR2, Cat# 306606, 1/300
CD117-APC, BD Pharmigen, 104D2, Cat# 333233, 1/30
- Antibodies used for PDX model flow cytometry readout and sorting
CD2-APC-F750, Biolegend, RPA-2.10, 300225, 1/160
CD3-APC-F750, Biolegend, SK7, 344839, 1/160
CD7-APC-F750, Biolegend, CD7-6B7, 343121, 1/160
CD11b-APC-F750, Biolegend, ICRF44 301351, 1/160
CD14-APC-F750, Biolegend, 63D3, 367120, 1/160
CD19-APC-F750, Biolegend, SJ25C1, 363029, 1/160
CD20-APC-F750, Biolegend, 2H7, 302357, 1/160
CD56-APC-F750, Biolegend, 5.1H11, 362553, 1/160
CD235a-APC-F750, Biolegend, HIR2, 306622, 1/160
CD90-PECy7, Biolegend, 5E10, 328124, 1/20
CD34-APC, Biolegend, 581, 343510, 1/100
CD34-PECy7, Beckman Coulter, 581, A21691, 1/100 or 1/50
CD38-BV421, Biolegend, HIT2, 303526, 1/20
CD123-PE, Biolegend, 6H6, 306005, 1/40
CD45RA-BB515, BD Biosciences, HI100, 564552, 1/40
mCD45-BV605, BD Biosciences, 30-F11, 563053, 1/40
mCD45-APC, BD Biosciences, 30-F11 , 559864, 1/100
hCD45-BV786, BD Biosciences, HI30, 563716, 1/40 or 1/50
hCD45-PE, BD Biosciences, HI30, 555483, 1/100
CD41-APC-F750, Biolegend, HIP8, 303749, 1/160
CD235a-PE, BD Biosciences, GA-R2, 555570, 1/50
CD71-BV605, BD Biosciences, M-A712, 743306, 1/50
CD36-APC, BD Biosciences, CB38, 550956, 1/50
CD123-FITC, Biolegend, 6H6, 306013, 1/50
CD117-BV711, Biolegend, 104D2, 313229, 1/50
mCD45-PerCP-Cy5.5, Biolegend, F11 , 103131, 1/50
- Antibodies used for human in vitro differentiation flow cytometry readout.
CD34-PECy7, Beckman Coulter, 581, A21691, 1/100
CD41-AP, BD Biosciences, HIP8, 559777, 1/100
CD42-PE, BD Biosciences, ALMA.16, 558819, 1/100
CD71-BV605, BD Biosciences, M-A712, 743306, 1/100
CD14-AF700, Biolegend, 63D3, 367114, 1/100
CD15-AF700, Biolegend, HI98, 301920, 1/100
CD11b-AF700, Biolegend, ICRF44, 301356, 1/100
CD235a-APC-F750, Biolegend, HIR2, 306622, 1/100
CD34-APC-e780, eBiosciences, 4HI1, 47-0349-42, 1/150
CD41a-APC, eBioscience, HIP8, 17-0419-42, 1/37.5
CD42b-PE, Biolegend, HIP1, 303906, 1/60
CD71-AF700, BD Pharmigen, M-A712, 563769, 1/30
CD117-BV711, Biolegend, 104D2, 313230, 1/60
CD33-PECy7, Biolegend, P67.6, 366618, 1/150
CD235a-BV421, BD Horizon, GA-R2, 562938 , 1/300
CD14-FITC, eBiosciences,  61D3, 11-0149-42, 1/75
CD11b-FITC, eBioSciences, ICRF44, 11-0118-42, 1/75
- Antibodies used for mouse peripheral blood and bone marrow flow cytometry readouts.
CD45.1-BV605, BioLegend, A20, 110738, 1/100
CD45.2 AF700, BioLegend, 104, 109822, 1/100 BM, 1/400 PB
Mac1 PE-Cy7, BioLegend, M1/70, 101215, 1/1600
CD19 eF450, eBioscience, eBio1D3, 48-0193-82, 1/150
CD4 BV650, BioLegend, RM4-5, 100546, 1/200
CD8a APCeF780, eBioscience, 53-6.7, 47-0081-82, 1/400
NK1.1-PE-Cy5, BioLegend, PK136, 108716, 1/400
Ly6G FITC, BioLegend, 1A8, 127606, 1/100
CD11b PE, BioLegend, M1/70, 101208, 1/100
CD3 Pacific Blue, BioLegend, 17A2, 100214, 1/100
B220 APC-Cy7 , BioLegend, RA3-6B2, 103224, 1/100
NK1.1 APC-Cy7, BioLegend, PK136, 108724, 1/100
CD45.1 PE-Cy7, BioLegend, A20, 110730, 1/100
CD45.2 APC, BioLegend, 104, 109814, 1/100
CD4 PE-Cy5, BioLegend, RM4-5, 100514, 1/400
B220 PE-Cy5, BioLegend, RA3-6B2, 103210, 1/200
Gr1 PE-Cy5, BioLegend, RB6-8C5, 108410, 1/800
CD5 PE-Cy5, BioLegend, 53-7.3, 100610, 1/800
Mac1 PE-Cy5, BioLegend, M1/70, 101210, 1/800
CD8a PE-Cy5, BioLegend, 53-6.7, 100710, 1/1200
cKIT APC-eF780, eBioscience, 2B8, 47-1171-82, 1/400
Sca1 BV05, BioLegend, D7, 108133, 1/200
CD150 PE-Cy7, BioLegend, TC15-12F12.2, 115914, 1/400
CD48 APC, BioLegend, HM48-1, 103412, 1/600
CD45.1 FITC, BioLegend, A20, 110706, 1/400

```
Sca1 PE-Cy7, BioLegend, E13-161.7, 122514 , 1/600
Ter119 PE-Cy5, BioLegend, TER-119, 116210, 1/400
Ter119, BioLegend, TER119, 116202, 5 µL / 25.106 cells
B220, BioLegend, RA3-6B2, 103202, 5 µL / 25.106 cells
Gr1, BioLegend, RB6-8C5, 108402, 6.25 µL / 25.106 cells
CD4, BioLegend, GK1.5, 100402, 1.25 µL / 25.106 cells
CD8, BioLegend, 53-6.7, 100702, 1.25 µL / 25.106 cells
CD11b, BioLegend, M1/70, 101202, 2.5 µL / 25.106 cells
B220 APC, BioLegend, RA3-6B2, 103212, 1/100
Ter119 APC, BioLegend, TER119, 116212, 1/100
Gr1 APC, BioLegend, RB6-8C5, 108412, 1/100
CD3 APC, BioLegend, 17A2, 100236, 1/100
CD11b APC, BioLegend, R1/70, 101212, 1/100
c-Kit PerCP-Cy5.5, BioLegend, 2B8, 105824, 1/50
Sca1 PE-Cy7, BioLegend, D7, 108114 , 1/50
CD150 PE, BioLegend, TC15-12F12.2 , 115904, 1/50
FcGR APC-Cy7, BioLegend, 93, 101327, 1/50
CD41 AF700, BioLegend, MWReg30, 133926, 1/50
CD105 BV786, BDHorizon, MJ7/18, 564746, 1/50
CD48 BV711, BioLegend, HM48-1, 103441, 1/50
CD45.1 BUV 395, BDHorizon, A20, 565212, 1/25
CD45.2 BUV737, BDHorizon, 104, 612778, 1/25
AnnexinV  FITC, BioLegend, NA, 640906, 1/100
Ki67 FITC, BioLegend, 16A8, 652410, 1/25
```

| | |
|---|---|
| Validation | Human and mouse antibodies were already validated, titrated and referenced  in peer-reviewed publications, as described on the suppliers' websites (Biolegend, eBiosciences, BD horizon, BD Biosciences, BD Pharmingen, Beckman Coulter). Combination of antibodies for human hematopoietic stem cells and progenitors have been already tested in our previous publication (Rodriguez-Meira et al, Mol Cell, 2019). |

# Animals and other research organisms

Policy information about studies involving animals; ARRIVE guidelines recommended for reporting animal research, and Sex and Gender in Research

| | |
|---|---|
| Laboratory animals | Mouse housing was carried out in individually ventilated cages (19-24°C, humidity 40-65%, 12/12 light dark cycle). Enrichment was done with nesting and bedding material. Mice were fed on standard croquettes, and supplemented with nutritionally complete gel diet after irradiation and in case of weight loss. Mice were maintained on a specific and opportunistic pathogen free health status.<br><br>- NOD.CB17-Prkdcscid IL2rgtm1/Bcgen mice (B-NDG, Envigo), female, 8 weeks-old (PDX experiments)<br>- C57/BL6 wild type mice CD45.1 (11-17 weeks old) or CD45.2 (6-8 weeks old), male and female (for chimera experiments with poly(I:C) and LPS challenge)<br>- C57/BL6 Trp53tm2Tyj Commd10Tg(Vav1-icre)A2Kio (referred to as Trp53R172H/+) CD45.1, 5-6 weeks old, male and female (for chimera experiments with poly(I:C) and LPS challenge)<br>- C57/BL6 Trp53tm2Tyj Tg(Tal1-cre/ERT)42-056Jrg (referred to as Trp53LSL-R172H/+) CD45.2, 8-13 weeks old, males (for chimera experiments with poly(I:C) challenge) |
| Wild animals | This study did not involve wild animals. |
| Reporting on sex | - NOD.CB17-Prkdcscid IL2rgtm1/Bcgen mice (B-NDG, Envigo) : female<br>- C57/BL6 wild type mice CD45.1 or CD45.2 : male and female<br>- C57/BL6 Trp53tm2Tyj Commd10Tg(Vav1-icre)A2Kio (referred to as Trp53R172H/+) CD45.1 :  male and female<br>- C57/BL6 Trp53tm2Tyj Tg(Tal1-cre/ERT)42-056Jrg (referred to as Trp53LSL-R172H/+) CD45.2 : males |
| Field-collected samples | Study did not involve field-collected samples. |
| Ethics oversight | Mouse experiments were approved by the French National Ethical Committee on Animal Care (n° 2020-007-23589) and by the UK University of Oxford Animal Welfare and Ethical Review Body (Project License P2FF90EE8 ). |

Note that full information on the approval of the study protocol must also be provided in the manuscript.

# Flow Cytometry

## Plots

Confirm that:

☒ The axis labels state the marker and fluorochrome used (e.g. CD4-FITC).

☒ The axis scales are clearly visible. Include numbers along axes only for bottom left plot of group (a 'group' is an analysis of identical markers).

☒ All plots are contour plots with outliers or pseudocolor plots.

☒ A numerical value for number of cells or percentage (with statistics) is provided.

## Methodology

| | |
|---|---|
| Sample preparation | Human or mouse samples were stained in IMDM + 10% FCS or PBS + 5% FCS (respectively) with several antibodies, incubated during 20min at RT and washed before being analyzed. Staining for apoptosis was done in  Annexin V binding buffer 1X (BD Biosciences). Cell cycle was assessed after fixation and permeabilization (BD Cytofix/Cytoperm and Permeabilization Buffer Plus, BD Biosciences).<br><br>All methods for sample preparation are fully described in the methods section of the manuscript. |
| Instrument | Cells were analyzed on a FACSCanto II or a BD Fortessa X20 (BD Biosciences) instrument.<br>Cells were sorted on a Influx Cell sorter (BD Biosciences), a  BD Fusion I or Fusion II instruments (Becton Dickinson)  or SH800S or MA900 sorters (SONY). |
| Software | Analysis of the flow cytometry data was performed using Kaluza (Beckman Coulter) or FlowJo (version 10.1, BD Biosciences) softwares. |
| Cell population abundance | Human and mouse haematopoietic stem and progenitor (HSPC) populations represent minor cell types (in the majority of cases, less than 1-5% of the total sample), except when they display a competitive advantage in the context of leukemic transformation. Sorting was performed in purity mode for bulk experiments and single-cell mode for single-cell sorting experiments. Post-sort purify was checked by sorting 100 cells from selected HSPC fractions (e.g. Lin-CD34+CD38- cells for human experiments and Lin-Sca1+ckit+ for mouse experiments) into an eppendorf tube containing 100 uL sorting buffer and analyzing the number of cells included within the same immunophenotype. Post-sort purity was consistently above 95%. |
| Gating strategy | Gating strategies are outlined in Extended Data Fig.2a (human HSPCs), Fig.5a (human cells from PDX models) and Fig.10b-c, l-n  (mouse bone marrow and peripheral blood). |

☒ Tick this box to confirm that a figure exemplifying the gating strategy is provided in the Supplementary Information.

