## [Peer Review File · Nature Genetics]

Peer Review Information

Manuscript Title: Single-Cell Multi-Omics Identifies Chronic Inflammation as a Driver of TP53 mutant Leukaemic Evolution

Corresponding author name(s): Professor Adam Mead, Dr Alba Rodriguez-Meira, Dr Iléana Antony-Debré

Reviewer Comments & Decisions:

Decision Letter, initial version:

16th Mar 2022

Dear Adam,

Your Article, "Deciphering TP53 mutant Cancer Evolution with Single-Cell Multi-Omics" has now been seen by 4 referees. You will see from their comments copied below that while they find your work of considerable potential interest, they have raised quite substantial concerns that must be addressed. In light of these comments, we cannot accept the manuscript for publication, but would be very interested in considering a revised version that addresses these serious concerns.

In summary, all four reviewers appreciated the technical quality of the work, and these reports suggest there is a clear path to publication.

Reviewer #1 is the most positive for your work, saying it is "cardinal addition to our understanding of TP53 pathology". Their requests are minor.

Reviewer #2, while also appreciative of your work's strengths, has one major - and, in our mind, vital - comment: further characterisation of how, exactly, the chronic inflammation leads to sAML development.

Reviewer #3 says that your findings are "timely and highly relevant", but raise questions about a number of your results. Most notably, they suggest that the pI:C stimulus used in your murine experiments may not be the most appropriate way to induce chronic inflammation. They provide clear guidance on how to address these issues.

Reviewer #4 also sounds positive, but also makes a few requests.

We note there are several overlapping concerns, but we think the most important one is that raised by Reviewer #2 (and echoed by Reviewers #3 and #4): further detail on how chronic inflammation leads

to progression to sAML is needed, which will require further experimental work in vivo. We note that Reviewer #3's comment #2 suggests that the stimuli used in these murine studies should also be carefully considered.

We hope you will find the referees' comments useful as you decide how to proceed. If you wish to submit a substantially revised manuscript, please bear in mind that we will be reluctant to approach the referees again in the absence of major revisions.

To guide the scope of the revisions, the editors discuss the referee reports in detail within the team, including with the chief editor, with a view to identifying key priorities that should be addressed in revision and sometimes overruling referee requests that are deemed beyond the scope of the current study. We hope that you will find the prioritised set of referee points to be useful when revising your study. Please do not hesitate to get in touch if you would like to discuss these issues further.

If you choose to revise your manuscript taking into account all reviewer and editor comments, please highlight all changes in the manuscript text file. At this stage we will need you to upload a copy of the manuscript in MS Word .docx or similar editable format.

*2) If you have not done so already please begin to revise your manuscript so that it conforms to our Article format instructions, available [here](http://www.nature.com/ng/authors/article_types/index.html). Refer also to any guidelines provided in this letter.

[redacted]

Note: This URL links to your confidential home page and associated information about manuscripts you may have submitted, or that you are reviewing for us. If you wish to forward

this email to co-authors, please delete the link to your homepage.

If you wish to submit a suitably revised manuscript we would hope to receive it within 6 months. If you cannot send it within this time, please let us know. We will be happy to consider your revision so long as nothing similar has been accepted for publication at Nature Genetics or published elsewhere. Should your manuscript be substantially delayed without notifying us in advance and your article is eventually published, the received date would be that of the revised, not the original, version.

Thank you for the opportunity to review your work.

Sincerely,

Michael Fletcher, PhD
Associate Editor, Nature Genetics

ORCID: 0000-0003-1589-7087

Referee expertise:

Referee #1: bone marrow fibrosis, MDS, leukaemia.

Referee #2: AML development and genomics; clonal haematopoiesis.

Referee #3: HSPCs, stress, leukaemia.

Referee #4: genomics and evolution of myeloid leukaemias.

Reviewers' Comments:

Reviewer #1:

Remarks to the Author:

Rodriguez-Meira et al. performed allelic resolution single-cell multi-omic analysis of HSPCs in patients with MPN who transform to secondary acute myeloid leukemia (AML) in their manuscript "Deciphering TP53 mutant Cancer Evolution with single-cell multi-omics".

This manuscript and data provided are a cardinal addition to our understanding of TP53 pathology. The elegant studies performed in the manuscript provide high granularity on the clonal evolution of TP53 mutant clones in MPN on its progression to sAML even including functional validation of findings. The recognition that extrinsic suppression promotes TP53-mutant transformation provides new opportunities to prevent clonal expansion.

In particular they show complex genetic intratumoral heterogeneity in TP53-sAML. TP53 multi-hit clones were in particular enriched in progenitor populations (HSPCs) but rather rare in primitive HSCs. Interestingly, acquisition of TP53 is quickly followed by acquisition of chromosomal abnormalities. Certain chromosomal abnormality patterns, especially chromosome 7 loss, are collectively required for leukemic stem cell expansion.

Integration of single cell transcriptomes and diffusion maps of HSPCs from TP53-sAML showed distinct clustering compared to TP53-WT pre-LSCs. Unexpectedly, the TP53 mutant clones were rather enriched in erythroid-biased populations. They elegantly follow up on this finding by functional experiments and show a direct effect of TP53 knock-down on increased erythroid and decreased myeloid differentiation.

A "stemness score" was developed which translated into a 51-gene "p53LSC-signature". A high score was strongly associated with poor survival, providing a powerful new tool to aid risk stratification in AML.

Another provocative finding from their data suggests a role for inflammatory signaling, in TP53 heterozygous cells which progress to blast phase (compared to those in chronic phase). This finding is validated in competitive transplantation of Trp53 mutant cells compared to WT cells under poly(IC) stimulus.

These findings provide a crucial conceptual advance in clonal evolution but also selection of TP53 mutant cells which has also high relevance for other cancer types.

I only have a few minor comments:

- 1) The inflammation-related clonal dominance in the competitive transplantation is very interesting. After 20 weeks the chimerism for Trp53 mutant cells is around 60% in the poly(IC) treated group - did the authors see any effects on blood counts? Was there a myeloid or lymphoid expansion?
- 2) Chronic inflammation impairs the erythroid differentiation. Did the authors see the erythroid bias in the Trp53 mutant clones? Was there a rescue of an erythroid differentiation defect? Did some mice develop a leukemic phenotype?
- 3) Using index sorting of HSPCs populations, the authors describe that TP53 multi-hit clones are enriched in progenitor populations (as shown in Extended Figure 3a), although no quantification is provided (only two representative patients are presented).
- 4) Further investigations from the study points towards an erythroid bias of these clones. Does the index sorting data confirm this finding?
- 5) In Extended Figure 9a and b, the authors present the HSPC immunophenotype across chronic phase TP53-MPN, pre TP53-sAML and TP53-sAML, and conclude that this phenotype is comparable in TP53-MPN and pre TP53-sAML. A quantification would be helpful, to rule out that FACS profiling cannot help predicting progression to sAML.
- 6) In Figure 4j, the authors present a scheme summarizing their results and proposed model. This

representation may suggest that inflammation suppresses MPN clones, while it seems to do so only in a competition model. This may be clarified.

Reviewer #2:

Remarks to the Author:

Impression/summary

This is a well-conducted study using precious primary patient samples to investigate the basis of progression of MPN to TP53-mutated sAML (TP53-sAML). The authors harness their leading skills in single cell sequencing by coupling single cell genotyping (TARGETseq) with index-sorted ss-RNaseq to study cells with relevant genotypes in informative MPN patients and healthy donors. They use in vitro human cells and mouse models to gain further insights/validate their observations. Their main findings are:

- i. Confirmation that stepwise bi-allelic TP53 mutation/loss is central to TP53-sAML progression
- ii. Evidence that CNAs/karyotypic abnormalities drive TP53-sAML leukemogenesis
- iii. Demonstration that TP53-sAMLs showed enrichment for HSC and early erythroid population + display aberrant erythroid-biased differentiation trajectories (mirroring the frequency of TP53 mutations in primary erythroleukemia)
- iv. Derivation of a new p53LSC-signature that associates with poor prognosis in independent AML datasets
- v. Evidence that TP53-WT HSCs may be suppressed/blocked by cell-extrinsic inflammatory signals in vivo (but are able to overcome this suppression ex-vivo and differentiate to mature lineages when cultured ex vivo)
- vi. Evidence that TP53-mutant cells in MPN patients who went on to progress (vs not progress) to sAML displayed inflammatory gene signatures and were resistant to IFN γ in vitro.

The manuscript is technically excellent and makes insightful contributions to our understanding of the cellular and molecular basis for progression of MPN to TP53-sAML, whilst also deriving a new prognostic signature that correlates with AML prognosis. The most significant advance is the finding that chronic inflammation appears to suppress TP53-WT HSPC whilst enhancing the fitness advantage of TP53-mutant cells. The authors propose that this phenomenon is important in the progression of TP53-mutant MPN to AML.

Assessment

Experiments and analysis are of high quality and I have few criticisms/comments. Also, interpretation of the findings is sound in the main. However, the most impactful conclusion of the manuscript, i.e. that chronic inflammation has a role in the development of TP53-sAML requires additional clarification and experimental support.

Major criticism

1. The authors present evidence that pre-TP53-sAML stem cells are resistant to inflammation. However, it is not clear how this leads to leukemic progression, as progression happens only after acquisition of bi-allelic TP53 mutations. How does this advantage of heterozygous TP53-mut cells facilitate a second hit (Fig 4j)? Simply through increased cell numbers or through increased mutation rates? Evidence for either of these would strengthen the manuscript.

A plausible alternative hypothesis is that the resistance to inflammatory signaling in pre-TP53-sAML stem cells is greatly augmented by a second hit (something that would not be true TP53-mutant MPN stem cells lacking this signaling), as evidenced by the loss of a bi-allelic clone in one CP TP53-MPN

patient – proposing that acquisition of second hit mutations is not rate-limiting). This alternative hypothesis is testable experimentally and if found correct would strengthen this manuscript, as would any evidence for the molecular basis of the inflammatory signaling (which admittedly may be more difficult to derive).

Minor Criticisms

1. The precise nature of TP53 mutations including VAFs should be included in Table S1.

Reviewer #3:

Remarks to the Author:

In this manuscript, the authors performed allelic resolution analysis on TP53 at the single cell level of hematopoietic stem and progenitor cells (HSPCs) from MPN patients that developed into secondary AML (sAML). By single cell transcriptomics analysis, the authors demonstrate that the dominant clones upon transformation carry multiple TP53 mutations and exhibit leukemia stem cell (LSC) or erythroid gene expression signatures. The TP53-LSC signature identified AML patients with poor prognosis in two independent cohorts independent of TP53 mutational status, providing a novel risk stratification strategy. By longitudinal analyses of patient samples at chronic phase MPN (CP-MPN) and subsequent sAML stages, the authors identify augmented inflammation as a possible mediator of transformation of CP-MPN to sAML. This notion was further supported by mouse studies, which showed that Trp53-R172H/+ heterozygous cells expanded in recipient mice in response to inflammatory stimuli.

Overall, the study presents a comprehensive analysis of how TP53 mutations contributes to the clonal evolution of sAML. The TP53-LSC signature provides an improved risk stratification strategy over the highly regarded LSC17 score. The finding that inflammation promotes TP53 mutant HSPCs to outcompete TP53 WT cells in vivo is timely and highly relevant in the field of clonal hematopoiesis (CH) and myeloid malignancies, as many new studies are finding the essential role it has in clonal evolution of CH mutant clones. It is also novel as most studies on inflammation and CH have focused on TET2 or DNMT3A mutations whereas most studies on TP53 have focused on DNA damaging agents and chemotherapies (eg. Bolton et al). Some concerns were noted in the definition of preLSCs, how general inflammation is involved in TP53 mutant clonal expansion, and the significance of the erythroid fraction in TP53 AML.

Major critiques:

1. The authors describe that TP53 WT preLSCs were enriched in HSC associated genes but exhibit reduced clonogenicity, retained expression of CD34, and reduced proliferation in short-term cultures (Fig3j and Ext. Data Fig7). The authors also describe that 60% of the preLSCs had MPN-related mutations and 40% were wild-type for all mutations. If the population carried no mutations, how can the authors define them as “preLSCs”? Aren’t they the residual normal HSCs? This possibility is consistent with the authors notion that the “preLSC were strikingly enriched in the phenotypic HSC compartment”. Additionally, the authors attribute the defective proliferation in short term culture assays to cell extrinsic effects from the leukemic microenvironment, implying no cell intrinsic defects in the “preLSCs”.

Related to this question, the authors should provide a detailed breakdown of the 880 preLSCs in terms

of the patients (which of the 9 patients did they come from?) and the type of evolution they followed (Fig1b-e). Are the 532 cells (60%) that had MPN mutations from patterns b, c, d in Fig1 (thus, are the antecedent of TP53 mutant sAML cells) or from pattern Fig1e that exhibited independent evolution from TP53 mutant sAML cells? A concern is whether the conclusion was derived from cells that were collected from only few patients, skewing the results to a biology of clones that followed a specific evolution path and not broadly applicable to TP53 mutant MN/sAML in general.

2. Poly (I:C) has been used widely to induce the type I interferon response (interferon alpha/beta) (Nature. 458:904, Nat Med. 15:696). While poly (I:C) can induce the type II interferon (interferon gamma) response, it is not a commonly used agent to induce IFN γ . To establish the specific role of IFN γ in promoting p53 mutant cell expansion, the authors should treat mice with IFN γ (as in Nature. 465:793), use additional inflammatory stimuli that induces IFN γ , or assess the genetic dependency on the IFN γ receptor. Alternatively, if the model is that a variety of inflammatory signals (albeit excluding IFN α as in Figure 4i) promote p53 mutant expansion, this should be experimentally established by using different inflammatory stimuli (e.g. LPS) in vivo.

3. Single cell RNA-seq analysis revealed that TP53 mutant sAML cells can be separated into two fractions that have either high LSC or erythroid scores. Interestingly, pseudotime analysis shows that the differentiation trajectory of CD34+ HSPCs is directed towards the erythroid fraction. The significance of the erythroid fraction is weak as is currently presented. Whether the erythroid fraction is downstream of the LSC fraction should be assessed and discussed. For example, will TP53 mutant HSPCs (which seems to be enriched in LMPP or myeloid progenitors) exhibit erythroid differentiation in transplantation or in culture?

4. Related to the discovery of the erythroid fraction in TP53 mutant sAML, some clinical significance of the erythroid fraction should be explored and discussed. Given the seemingly linear relation between LSC score high cells and erythroid score high cells (Fig. 2a-b), will the erythroid signature (extracted similarly to the LSC signature in Fig2k) also identify AML patients with poor prognosis? Given the known counteracting functions of p53 and GATA1 (see below), will the erythroid score or GATA1 expression identify AML patients that have suppressed p53 target gene expression?

5. The authors mention that erythroleukaemia is uncommon in TP53-sAML. Of note, TP53 mutations are frequently found in acute erythroid leukaemias (e.g. Leukemia. 27:1940–1943) and GATA1 and p53 interact to inhibit each other (Blood 114:165). This should be discussed.

6. The authors describe that known p53-pathway genes were enriched in TP53 multi-hit HSPCs compared to TP53 wild-type cells (Ext Data Fig. 4b). This is not apparent in the heatmap provided, as it is not clear if one of the quadrants show enriched expression of TP53 target genes. Can the authors provide a GSEA or a similar analysis on p53 target genes?

Minor critique:

1. Line 254. The authors indicate 6 CP-MPN patients but the figure and line 263 describe this as 4 CP-MPN patients.

Reviewer #4:

Remarks to the Author:

Rodriguez-Meira et al set out to understand how TP53 mutant clones transform from MPN to secondary AML. They used a single cell sequencing strategy that integrates transcriptomic and mutational data on cd34+ cells from these samples. Overall, these samples acquire successive hits to the p53 locus over time result in a TP53 hemizygous or biallelic mutant state upon transformation to sAML. Consistent with previous reports, TP53 secondary AML was associated with complex cytogenetic evolution. They find that TP53 mutant cells have higher erythroid-associated transcriptional program scores relative to P53 WT cells from these same samples. This was further associated with increased GATA1 expression, indicating that there may be enhanced erythroid priming. Signatures indicative of impaired differentiation, however they maintained functional differentiation potential in vitro. Next, they compare pre-transformation MPN->sAML samples to chronic phase MPN samples—both with p53 mutations. These pre-sAML p53 mutated cells demonstrated upregulation of interferon response genes. In a murine model of IFN challenge p53 mutant cells have greater fitness than their WT counterparts. Collectively this suggests that IFN associated with inflammation may enable the outgrowth of p53 mutant clones.

Overall, this is an elegant manuscript with complementary single cell multiomic and functional studies. A few points that may strengthen the manuscripts are below:

Figure 2a. Does the p53 LSC signature simply pick out the p53 mutated cases from Beat-AML and TCGA, which would already be predicted to have inferior overall survival, or does it also identify P53 WT cases that cluster with P53-mutant cases, and thus allow for improved stratification over P53 mutant status alone? The discussion states that this signature is predictive regardless of p53 status, but this is hard to appreciate when reading the results.

Figure 4c. The downregulation of TNFa/TGFB signatures is a bit hard to find in the supplemental tables, perhaps it would help the reader to add the NES value to the text. Figure 4c shows the NES for IFNgamma at 1.67, but not IFN alpha at 1.91, which is a bit more impressive. It might be good to show both, along with the associated GSEA plots as this is such a key point in the paper.

How might higher IFN gene expression signatures in pre-TP53 AML cells correspond to increased survival under IFNy challenge? One could imagine that an increase in IFN associated gene signatures could actually be an indication of vulnerability to inflammatory challenge. Some additional explanation might be helpful here.

Is there any potential connection between erythroid bias and increased fitness under IFNy challenge? Relatedly might GATA1 or CEBPA be known to participate in any of these response processes?

Minor

-Line 50, a very brief explanation of the TARGET seq methodology would be helpful to the reader here.

A little interpretation about TNFa being down but IFN up with respect to inflammatory processes would be helpful to the reader.

Author Rebuttal to Initial comments**Response to Reviews: Deciphering TP53 mutant Cancer Evolution with Single-Cell Multi-Omics (NG-A59372R)****Reviewer #1:**

Remarks to the Author:

Rodriguez-Meira et al. performed allelic resolution single-cell multi-omic analysis of HSPCs in patients with MPN who transform to secondary acute myeloid leukemia (AML) in their manuscript "Deciphering TP53 mutant Cancer Evolution with single-cell multi-omics".

This manuscript and data provided are a cardinal addition to our understanding of TP53 pathology. The elegant studies performed in the manuscript provide high granularity on the clonal evolution of TP53 mutant clones in MPN on its progression to sAML even including functional validation of findings. The recognition that extrinsic suppression promotes TP53-mutant transformation provides new opportunities to prevent clonal expansion.

In particular they show complex genetic intratumoral heterogeneity in TP53-sAML. TP53 multi-hit clones were in particular enriched in progenitor populations (HSPCs) but rather rare in primitive HSCs. Interestingly, acquisition of TP53 is quickly followed by acquisition of chromosomal abnormalities. Certain chromosomal abnormality patterns, especially chromosome 7 loss, are collectively required for leukemic stem cell expansion.

Integration of single cell transcriptomes and diffusion maps of HSPCs from TP53-sAML showed distinct clustering compared to TP53-WT pre-LSCs. Unexpectedly, the TP53 mutant clones were rather enriched in erythroid-biased populations. They elegantly follow up on this finding by functional experiments and show a direct effect of TP53 knock-down on increased erythroid and decreased myeloid differentiation.

A "stemness score" was developed which translated into a 51-gene "p53LSC-signature". A high score was strongly associated with poor survival, providing a powerful new tool to aid risk stratification in AML.

Another provocative finding from their data suggests a role for inflammatory signaling, in TP53 heterozygous cells which progress to blast phase (compared to those in chronic phase). This finding is validated in competitive transplantation of Trp53 mutant cells compared to WT cells under poly(IC) stimulus.

These findings provide a crucial conceptual advance in clonal evolution but also selection of TP53 mutant cells which has also high relevance for other cancer types.

RESPONSE: We are very grateful to the reviewer for their positive

comments. I only have a few minor comments:

- 1) The inflammation-related clonal dominance in the competitive transplantation is very interesting. After 20 weeks the chimerism for *Trp53* mutant cells is around 60% in the poly(I:C) treated group - did the authors see any effects on blood counts? Was there a myeloid or lymphoid expansion?

RESPONSE: To determine how poly(I:C) treatment might alter blood counts, we established an inducible SCL-CreER^T *Trp53*^{R172H/+} mouse model as described in the results section lines 256-258 and new Fig.5h. Poly(I:C) treatment led to inflammation-associated changes in blood cell parameters, including anaemia, leucopenia and thrombocytopenia as now described in the results section lines 258-260 and in associated new figures (Extended Data Fig.10h-j). Poly(I:C) treatment was also associated with a myeloid bias in peripheral blood leucocytes specifically associated with *Trp53*-mutation as described in the results section lines 260-264 and Fig.5i,j.

- 2) Chronic inflammation impairs the erythroid differentiation. Did the authors see the erythroid bias in the *Trp53* mutant clones? Was there a rescue of an erythroid differentiation defect? Did some mice develop a leukemic phenotype?

RESPONSE: As described in response to the above point, we observed an expected inflammation-induced anemia with chronic poly(I:C) treatment. We also performed phenotypic analysis of BM erythroid progenitors (preCFU-e and CFU-E : Lin⁻Sca-1⁻c-Kit⁺CD41⁻FcgRII/III⁻CD105⁺) after chronic poly(I:C) treatment. Numbers of wild-type competitor erythroid progenitors were reduced upon poly(I:C) treatment (as expected), whereas *Trp53*-mutation was associated with an increase in erythroid progenitors that was not impacted by inflammation. These data are described on lines 265-269 and in the new Fig.5k. We did not observe a leukaemic phenotype in any of the *Trp53*^{R172H/+} mice within the timecourse of these experiments. As *Trp53* mutant mice are prone to develop T-cell leukaemia and lymphomas (e.g. Loizou et al, DOI: 10.1158/2159-8290.CD-18-1391), we deliberately planned the final readout of these experiments at an early timepoint to avoid the later confounding effect of lymphoid malignancies.

- 3) Using index sorting of HSPCs populations, the authors describe that TP53 multi-hit clones are enriched in progenitor populations (as shown in Extended Figure 3a), although no quantification is provided (only two representative patients are presented).

RESPONSE: We now provide quantification of index sorting data as an additional panel in Extended Data Fig.3a. These data show the expected high clonal burden of *TP53*-multi-hit cells in progenitor populations, with more frequent preLSCs in HSC and MPP populations.

- 4) Further investigations from the study points towards an erythroid bias of these clones. Does the index sorting data confirm this finding?

RESPONSE: To confirm the erythroid bias of *TP53*-multi-hit cells, we projected phenotypically

defined MEP (using index sorting data and associated single-cell genotyping) on the diffusion map from Fig.2a. This analysis shows that the MEP population maps to *TP53*-multi-hit cells harboring a high erythroid score (see Extended Data Fig.5f).

5) In Extended Figure 9a and b, the authors present the HSPC immunophenotype across chronic phase *TP53*-MPN, pre *TP53*-sAML and *TP53*-sAML, and conclude that this phenotype is comparable in *TP53*-MPN and pre *TP53*-sAML. A quantification would be helpful, to rule out that FACS profiling cannot help predicting progression to sAML.

RESPONSE: We quantified HSPC populations in total CD34+ cells from healthy donors, chronic phase-*TP53*-MPN and pre-*TP53*-sAML samples. Results are presented in Extended Data Fig.9c. These data show no significant differences in the progenitor cell distribution between chronic phase-*TP53*-MPN and pre-*TP53*-sAML samples. Analysis of paired pre-*TP53*-sAML and *TP53*-sAML samples (prior to and at transformation, respectively) showed strong variations in progenitor cell distribution, with decreased HSCs and increased progenitors, and patient-to-patient heterogeneity in the specific progenitor population expanded (Extended Data Fig.9d).

6) In Figure 4j, the authors present a scheme summarizing their results and proposed model. This representation may suggest that inflammation suppresses MPN clones, while it seems to do so only in a competition model. This may be clarified.

RESPONSE: We thank the reviewer for making this interesting point and we agree that it may be misleading to suggest that inflammation suppresses the MPN clone. Indeed, our mouse model data does not test this possibility. We emphasize in the text that the proposed model is consistent with inflammation “suppressing wild-type haematopoiesis” (lines 287-288) and we have edited the proposed model (now in Fig.6e) so this does not imply that inflammation suppresses the MPN clone.

Finally, as is routine for all manuscripts in our laboratory at the revision stage, we have also carried out an extensive, independent and systematic recheck of all the data and analyses/scripts in the paper against the original raw data. This involved careful re-analysis of all data from raw data

files, to further ensure the highest possible integrity of the data as well as their accessibility to the readers. As part of this process, we identified some minor errors which we have corrected, resulting in some changes in the numbers of cells included and gene lists. Importantly, this has not resulted in any changes to any of the findings described or conclusions reached in our manuscript, although the appearance of some of Figures has been slightly altered by this re-analysis without any meaningful difference in the data. We would obviously be happy to clarify specific details of this process and the resulting changes at your request.

Reviewer #2:

Remarks to the Author:

Impression/summary

This is a well-conducted study using precious primary patient samples to investigate the basis of progression of MPN to TP53-mutated sAML (TP53-sAML). The authors harness their leading skills in single cell sequencing by coupling single cell genotyping (TARGETseq) with index-sorted ss-RNAseq to study cells with relevant genotypes in informative MPN patients and healthy donors. They use in vitro human cells and mouse models to gain further insights/validate their observations. Their main findings are:

- i. Confirmation that stepwise bi-allelic TP53 mutation/loss is central to TP53-sAML progression
- ii. Evidence that CNAs/karyotypic abnormalities drive TP53-sAML leukemogenesis
- iii. Demonstration that TP53-sAMLs showed enrichment for HSC and early erythroid population + display aberrant erythroid-biased differentiation trajectories (mirroring the frequency of TP53 mutations in primary erythroleukemia)
- iv. Derivation of a new p53LSC-signature that associates with poor prognosis in independent AML datasets
- v. Evidence that TP53-WT HSCs may be suppressed/blocked by cell-extrinsic inflammatory signals in vivo (but are able to overcome this suppression ex-vivo and differentiate to mature lineages when cultured ex vivo)
- vi. Evidence that TP53-mutant cells in MPN patients who went on to progress (vs not progress) to sAML displayed inflammatory gene signatures and were resistant to IFN γ in vitro.

The manuscript is technically excellent and makes insightful contributions to our understanding of the cellular and molecular basis for progression of MPN to TP53-sAML, whilst also deriving a new prognostic signature that correlates with AML prognosis. The most significant advance is the finding that chronic inflammation appears to suppress TP53-WT HSPC whilst enhancing the fitness advantage of TP53-mutant cells. The authors propose that this phenomenon is important in the progression of TP53-mutant MPN to AML.

RESPONSE: We are very grateful to the reviewer for their positive comments.

Assessment

Experiments and analysis are of high quality and I have few criticisms/comments. Also, interpretation of the findings is sound in the main. However, the most impactful conclusion of the manuscript, i.e. that chronic inflammation has a role in the development of TP53-sAML requires additional clarification and experimental support.

Major criticism

1. The authors present evidence that pre-TP53-sAML stem cells are resistant to inflammation. However, it is not clear how this leads to leukemic progression, as progression happens only after acquisition of bi-allelic TP53 mutations. How does this advantage of heterozygous TP53-mut cells facilitate a second hit (Fig 4j)? Simply through increased cell numbers or through increased mutation rates? Evidence for either of these would strengthen the manuscript.

RESPONSE: We thank the reviewer for this excellent suggestion to improve the manuscript. Exit from dormancy induced by a variety of inflammatory cytokines, including those induced by poly(I:C) and LPS, is known to cause DNA-damage-induced attrition in HSCs (e.g. Walter et al, Nature 2015; DOI: 10.1038/nature14131). We therefore propose that *TP53* mutation rescues HSCs that would otherwise undergo DNA-damage-induced attrition, ultimately leading to the accumulation of HSCs which have acquired DNA damage, thus promoting genetic evolution that underlies disease progression. In order to test this possibility, we established an inducible *Trp53*^{R172H/+} model (Fig. 5h) and demonstrated that *Trp53*^{R172H/+} LSK cells are resistant to inflammation induced apoptosis in comparison with the expected increase in apoptosis we observed in wild-type counterparts (Pietras et al, JEM, 2014; DOI: 10.1084/jem.20131043), whereas the cell cycle is increased by poly(I:C) in both WT and *Trp53* mutated cells (described in the results section lines 269-274 and new Fig.5l and 5m). To confirm that this ultimately leads to genetic evolution of *Trp53* mutated cells (gain of karyotypic aberrations), we carried out M-FISH karyotype analysis of CD45.2+ *Trp53*^{+/+} LSK cells expanded *in vitro* from mice following poly(I:C) treatment and CD45.1+ *Trp53*^{R172H/+} LSK cells from mice with or without poly(I:C) treatment. We observed a striking increase in karyotypic abnormalities in *Trp53* mutated LSK cells upon poly(I:C) treatment, including frequent acquisition of complex abnormalities (3 or more aberrations per cell) compared to both their poly(I:C) treated WT counterparts and *Trp53* mutated LSK cells without poly(I:C) treatment. Collectively, these data suggest that leukaemic progression under chronic inflammation is mediated both by increased cell number (through increased cell cycling with resistance to apoptosis in *Trp53* mutant cells) and increased genetic aberrations as observed in patients and in the mouse models. These new data are presented in a new figure (Fig.6a-d) and described in the results section on lines 276-289. We believe that these data address the reviewer's comment and strengthen the manuscript. To emphasize this point, we have added additional text in the discussion on lines 332-343: "We provide evidence that *TP53* mutant HSCs showing dysregulated inflammation-associated gene expression are enriched in patients who go on to develop *TP53*-sAML. We propose that HSC that would

otherwise undergo inflammation- associated and DNA-damage-induced attrition, are rescued by *TP53* mutation, ultimately leading to the accumulation of HSCs which have acquired DNA damage, thus promoting genetic evolution that underlies disease progression. Further studies are required to characterize this, and also the key inflammatory mediators and molecular mechanisms involved, which we believe are unlikely to be restricted to a single axis, with a myriad of inflammatory mediators overexpressed in MPN.”

A general note for *the in vivo* experiments. We performed the experiments in 2 laboratories with different mouse colonies (in Oxford and Paris), and we consequently used 2 different models for the induction of *Trp53* mutation: the Vav-iCre system presented at the 1st submission (Fig5.a) and the SCL-Cre-ER^T system we used to perform some additional experiments to address the reviewers' comments (Fig.5h). We clarify in each figure which model was used for the presented experiment and, importantly, we confirmed selection of the *Trp53* mutated cells upon poly(I:C) treatment in the SCL-Cre-ER^T system (Ext Data Fig. 10k, m).

A plausible alternative hypothesis is that the resistance to inflammatory signaling in pre-TP53-sAML stem cells is greatly augmented by a second hit (something that would not be true TP53-mutant MPN stem cells lacking this signaling, as evidence by the loss of a bi-allelic clone in one CP TP53-MPN patient – proposing that acquisition of second hit mutations is not rate-limiting). This alternative hypothesis is testable experimentally and if found correct would strengthen this manuscript, as would any evidence for the molecular basis of the inflammatory signaling (which admittedly may be more difficult to derive).

RESPONSE: To test this hypothesis, we compared selection of *Trp53* R172 heterozygous and hemizygous HSC upon chronic inflammation. As hemizygous *Trp53* R172 mice are particularly vulnerable to T-cell leukaemia and lymphomas, we were only able to carry out a shorter period of poly(I:C) treatment in the hemizygous mice (12 injections versus 18 for heterozygous mice) as some hemizygous mice began to succumb to lymphoid malignancy. Despite the shorter period of poly(I:C) treatment, as shown in the figure below, we observed a similar fitness advantage of *Trp53* R172 hemizygous HSC compared to the heterozygous cells with longer poly(I:C) treatment. This is consistent with both heterozygous and hemizygous *Trp53* R172 mutations conferring resistance to inflammation-associated attrition of HSPCs, with a stronger effect in hemizygous mice. As we were unable to make this comparison with the same length of poly(I:C) treatment, we have not included these new data in the manuscript, but would be happy to do so at the reviewer's request. This model is further supported by a new analysis showing enhanced inflammation-associated transcriptional signatures in multi-hit *TP53* HSPC in pre-*TP53*-sAML patients (compared to heterozygous *TP53* mutant cells) as described on lines 234-236 (new Extended Data Fig.9o). This is also in line with previous evidence studying competitive advantage of single versus multi-hit *Trp53* mutation in response to irradiation-induced DNA damage, where loss of the wild-type allele augments the fitness advantage of *Trp53* mutant HSPCs following low dose irradiation (Boettcher et al, Science, 2019; DOI: 10.1126/science.aax3649). Accordingly, we have added an additional sentence to make this important point in the discussion on lines 343-346.

Minor Criticisms

1. The precise nature of TP53 mutations including VAFs should be included in Table S1.

RESPONSE: The nature of *TP53* mutations including the VAFs are now included in Table S1.

Finally, as is routine for all manuscripts in our laboratory at the revision stage, we have also carried out an extensive, independent, and systematic recheck of all the data and analyses/scripts in the paper against the original raw data. This involved careful re-analysis of all data from raw data files, to further ensure the highest possible integrity of the data as well as their accessibility to the readers. As part of this process, we identified some minor errors which we have corrected, resulting in some changes in the numbers of cells included and gene lists. Importantly, this has not resulted in any changes to any of the findings described or conclusions reached in our manuscript, although the appearance of some of Figures has been slightly altered by this re-analysis without any meaningful difference in the data. We would obviously be happy to discuss specific details of this process and the resulting changes at your request.

Reviewer #3:

Remarks to the Author:

In this manuscript, the authors performed allelic resolution analysis on TP53 at the single cell level of hematopoietic stem and progenitor cells (HSPCs) from MPN patients that developed into secondary AML (sAML). By single cell transcriptomics analysis, the authors demonstrate that the dominant clones upon transformation carry multiple TP53 mutations and exhibit leukemia stem cell (LSC) or erythroid gene expression signatures. The TP53-LSC signature identified AML patients with poor prognosis in two independent cohorts independent of TP53 mutational status, providing a novel risk stratification strategy. By longitudinal analyses of patient samples at chronic

phase MPN (CP-MPN) and subsequent sAML stages, the authors identify augmented inflammation as a possible mediator of transformation of CP-MPN to sAML. This notion was further supported by mouse studies, which showed that Trp53-R172H/+ heterozygous cells expanded in recipient mice in response to inflammatory stimuli.

Overall, the study presents a comprehensive analysis of how TP53 mutations contributes to the clonal evolution of sAML. The TP53-LSC signature provides an improved risk stratification strategy over the highly regarded LSC17 score. The finding that inflammation promotes TP53 mutant HSPCs to outcompete TP53 WT cells in vivo is timely and highly relevant in the field of clonal hematopoiesis (CH) and myeloid malignancies, as many new studies are finding the essential role it has in clonal evolution of CH mutant clones. It is also novel as most studies on inflammation and CH have focused on TET2 or DNMT3A mutations whereas most studies on TP53 have focused on DNA damaging agents and chemotherapies (eg. Bolton et al).

RESPONSE: We are very grateful to the reviewer for their positive comments.

Some concerns were noted in the definition of preLSCs, how general inflammation is involved in TP53 mutant clonal expansion, and the significance of the erythroid fraction in TP53 AML.

Major critiques:

1. The authors describe that TP53 WT preLSCs were enriched in HSC associated genes but exhibit reduced clonogenicity, retained expression of CD34, and reduced proliferation in short-term cultures (Fig3j and Ext. Data Fig7). The authors also describe that 60% of the preLSCs had MPN-related mutations and 40% were wild-type for all mutations. If the population carried no mutations, how can the authors define them as “preLSCs”? Aren’t they the residual normal HSCs? This possibility is consistent with the authors notion that the “preLSC were strikingly enriched in the phenotypic HSC compartment”. Additionally, the authors attribute the defective proliferation in short term culture assays to cell extrinsic effects from the leukemic microenvironment, implying no cell intrinsic defects in the “preLSCs”.

RESPONSE: As pointed out by the reviewer, under our term “preLSC” we have included both cells carrying MPN-driver mutations as well as non-mutated residual HSCs. We now specifically clarify this in the text in the Results section (lines 172-175) as follows: “...TP53 wild-type cells, referred to as preLSCs, which include both residual HSPC that were wild-type for all mutations analyzed, as well as HSPC which form part of the antecedent MPN clone”. We now quantify the progenitor distribution of preLSCs, which are strikingly enriched in the HSC and MPP compartments (Ext Data Fig.3a, new right panel). We would prefer to retain the current terminology as the focus of our paper is on the impact of TP53 mutation and even cells which are wild-type for all mutations tested are not necessarily residual normal HSC. For example, we looked at the HSC-associated gene signature in preLSCs-WT and preLSCs-mutant MPN cells and identified an increased HSC gene score in both populations compared to healthy donors, as shown below. Although the increase was more marked in MPN cells, these results suggest that

that WT cells are also affected (this can be mediated through non genetic extrinsic or intrinsic effects).

Related to this question, the authors should provide a detailed breakdown of the 880 preLSCs in terms of the patients (which of the 9 patients did they come from?) and the type of evolution they followed (Fig1b-e). Are the 532 cells (60%) that had MPN mutations from patterns b, c, d in Fig1 (thus, are the antecedent of TP53 mutant sAML cells) or from pattern Fig1e that exhibited independent evolution from TP53 mutant sAML cells? A concern is whether the conclusion was derived from cells that were collected from only few patients, skewing the results to a biology of clones that followed a specific evolution path and not broadly applicable to TP53 mutant MN/sAML in general.

RESPONSE: We now provide detail of the 880 preLSC in term of patients' origin and type of evolution, the latter also according to presence or absence of MPN driver mutation (Ext Data Fig.7a). We hope this reassures the reviewer that the analysis was done on a mixture of patients (n=9), showing 4 types of evolution, without major skewing, suggesting that our finding could be broadly applicable to TP53 mutant sAML.

2. Poly (I:C) has been used widely to induce the type I interferon response (interferon alpha/beta) (Nature. 458:904, Nat Med. 15:696). While poly (I:C) can induce the type II interferon (interferon gamma) response, it is not a commonly used agent to induce IFN γ . To establish the specific role of IFN γ in promoting p53 mutant cell expansion, the authors should treat mice with IFN γ (as in Nature. 465:793), use additional inflammatory stimuli that induces IFN γ , or assess the genetic dependency on the IFN γ receptor. Alternatively, if the model is that a variety of inflammatory signals (albeit excluding IFN α as in Figure 4i) promote p53 mutant expansion, this should be experimentally established by using different inflammatory stimuli (e.g. LPS) in vivo.

RESPONSE: The reviewer is correct that our proposed model is that a variety of inflammatory

stimuli can promote p53 mutant expansion. Accordingly, as suggested by the reviewer, we carried out an additional *in vivo* experiment with LPS (new Fig. 5a) and demonstrate that this also leads to selection of *Trp53*^{R172H/+} myeloid cells and LSK (Fig.5f and g). As we now clarify in the discussion, lines 330-332: “Here, we demonstrate a hitherto unrecognized effect of TP53 mutations, which conferred a marked fitness advantage to HSPC in the presence of chronic inflammation induced with both poly(I:C) as well as LPS.”

Exit from dormancy induced by a variety of inflammatory cytokines, including those induced by poly(I:C) and LPS, is known to cause DNA-damage-induced attrition in HSCs (e.g. Walter et al, Nature 2015; DOI: 10.1038/nature14131). We therefore propose that *TP53* mutation rescues HSCs that would otherwise undergo DNA-damage-induced attrition, ultimately leading to the accumulation of HSCs which have acquired DNA damage, thus promoting genetic evolution that underlies disease progression. In order to test this possibility, we established an inducible *Trp53*^{R172H/+} model (Figure 5h) and demonstrated that *Trp53*^{R172H/+} LSK cells are resistant to inflammation induced apoptosis in comparison with the expected increase in apoptosis we observed in wild-type counterparts (Pietras et al, JEM, 2014; DOI: 10.1084/jem.20131043), whereas the cell cycle is increased by poly(I:C) in both WT and *Trp53* mutated cells (described in the results section lines 269-274 and new Fig.5l and 5m). To confirm that this ultimately leads to genetic evolution of *Trp53* mutated cells (gain of karyotypic aberrations), we carried out M-FISH karyotype analysis of CD45.2+ *Trp53*^{+/+} LSK cells expanded *in vitro* from mice following poly(I:C) treatment and CD45.1+ *Trp53*^{R172H/+} LSK cells from mice with or without poly(I:C) treatment. We observed a striking increase in karyotypic abnormalities in *Trp53* mutated LSK cells upon poly(I:C) treatment, including frequent acquisition of complex abnormalities (3 or more aberrations per cell) compared to both their poly(I:C) treated WT counterparts and *Trp53* mutated LSK cells without poly(I:C) treatment. Collectively, these data suggest that leukaemic progression under chronic inflammation is mediated both by increased cell number (through increased cell cycling with resistance to apoptosis in *Trp53* mutant cells) and increased genetic aberrations as observed in patients and in the mouse models. These new data are presented in a new Figure (Fig.6a-d) and described in the results section on lines 276-289. We believe that these data address the reviewer’s comment and strengthen the manuscript. To emphasize this point, we have added additional text in the discussion on lines 332-343: “We provide evidence that *TP53* mutant HSCs showing dysregulated inflammation-associated gene expression are enriched in patients who go on to develop *TP53*-sAML. We propose that HSC that would otherwise undergo inflammation- associated and DNA-damage-induced attrition, are rescued by *TP53* mutation, ultimately leading to the accumulation of HSCs which have acquired DNA damage, thus promoting genetic evolution that underlies disease progression. Further studies are required to characterize this, and also the key inflammatory mediators and molecular mechanisms involved, which we believe are unlikely to be restricted to a single axis, with a myriad of inflammatory mediators overexpressed in MPN.”

A general note for *the in vivo* experiments. We performed the experiments in 2 laboratories with different mouse colonies (in Oxford and Paris), and we consequently used 2 different models for the induction of *Trp53* mutation: the Vav-iCre system presented at the 1st submission (Fig5.a) and the SCL-Cre-ER^T system we used to perform some additional experiments to address the reviewers’ comments (Fig.5h). We clarify in each Figure which model was used for the presented

experiment and, importantly, we confirmed selection of the *Trp53* mutated cells upon poly(I:C) treatment in the SCL-Cre-ER^T system (Ext Data Fig. S10).

3. Single cell RNA-seq analysis revealed that TP53 mutant sAML cells can be separated into two fractions that have either high LSC or erythroid scores. Interestingly, pseudotime analysis shows that the differentiation trajectory of CD34⁺ HSPCs is directed towards the erythroid fraction. The significance of the erythroid fraction is weak as is currently presented. Whether the erythroid fraction is downstream of the LSC fraction should be assessed and discussed. For example, will TP53 mutant HSPCs (which seems to be enriched in LMPP or myeloid progenitors) exhibit erythroid differentiation in transplantation or in culture?

RESPONSE: To explore if the erythroid fraction is downstream of the LSC fraction, we transplanted sorted human CD34⁺ from TP53-sAML patients in immunodeficient mice, and observed that these cells exhibited erythroid differentiation potential, as evidenced by differentiation into both erythroid progenitors (hCD45⁺Lin⁻CD235a⁻CD123⁻CD36⁺CD71⁺) and more mature erythroid cells (hCD45⁺Lin⁻CD235a⁺) cells, with patient-to-patient heterogeneity. Additionally, TP53-sAML patient cells can give rise to erythroid differentiation after *in vitro* culture, as shown by the presence of CD235a⁺ cells in erythroid-promoting (+EPO) culture conditions. These data, now presented in Ext. Data Fig 5a-c, and are described on lines 113-115. To further strengthen the observation relating to the erythroid bias of TP53-multi-hit cells, we projected phenotypically defined MEP (using index sorting data and associated single-cell genotyping) on the diffusion map from Fig.2a. This analysis shows that the MEP population maps to TP53-multi-hit cells harboring a high erythroid score (see Ext Data Fig.5f). These erythroid cells show downregulation of LSC transcriptional modules (Extended Data Fig.4d). These data are complemented by our observations in independent cohorts that TP53 mutation is associated with increased erythroid transcriptional score. We would also draw the reviewer's attention to the new data from the mouse model relating to erythroid differentiation. We performed phenotypic analysis of BM erythroid progenitors (preCFU-E and CFU-E : Lin⁻Sca-1^c-Kit⁺CD41⁻FcgRII/III⁺CD105⁺) after chronic poly(I:C) treatment. Numbers of wild-type competitor erythroid progenitors were reduced upon poly(I:C) treatment (as expected), whereas *Trp53*-mutation was associated with an increase in erythroid progenitors that was not impacted by inflammation. These data are described on lines 265-269 and in the new Fig.5k.

4. Related to the discovery of the erythroid fraction in TP53 mutant sAML, some clinical significance of the erythroid fraction should be explored and discussed. Given the seemingly linear relation between LSC score high cells and erythroid score high cells (Fig. 2a-b), will the erythroid signature (extracted similarly to the LSC signature in Fig2k) also identify AML patients with poor prognosis? Given the known counteracting functions of p53 and GATA1 (see below), will the erythroid score or GATA1 expression identify AML patients that have suppressed p53 target gene expression?

RESPONSE: We thank the reviewer for this excellent suggestion. We applied the erythroid signature (p53Ery signature), derived as for LSC signature, to the BeatAML cohort and identified that a high erythroid score predicts poor overall survival in the BeatAML cohort. However, this

result is not reproducible in the TCGA cohort (as shown in the figures below). Therefore, we have decided to not include these new results in the revised version of our manuscript, but we would be happy to do so at the reviewer's request.

BeatAML, HR: 5.26, p-value < 2E-16

TCGA, HR: 0.95, p-value: 0.814

(Time is in years).

As suggested by the reviewer, we carried out an additional analysis and show that high erythroid score correlates with lower p53 target gene expression (now included in Ext. Data Fig.5k), indicating that a high erythroid score can identify AML patients with reduced p53 target gene expression. With regards to GATA1, we already show that GATA1 expression is increased in *TP53* mutant cells in our data (Fig.2g) and in *TP53* mutant AML (Ext. Data Fig .5l). As shown below, we also found a trend for lower p53 target gene expression in BeatAML cases with high GATA1 expression, but the comparison was not statistically significant (p=0.09). We have not included this figure in the revised manuscript but would be happy to do so at the reviewer's request.

5. The authors mention that erythroleukaemia is uncommon in TP53-sAML. Of note, TP53 mutations are frequently found in acute erythroid leukaemias (e.g. *Leukemia*. 27:1940–1943) and GATA1 and p53 interact to inhibit each other (*Blood* 114:165). This should be discussed.

RESPONSE: We have included these references in the discussion with an additional comment with regards to the interaction between GATA1 and p53 on lines 310-312: “Notably, *CEBPA* knockout or mutation is reported to cause a myeloid to erythroid lineage switch with increased expression of *GATA1* and, in addition, *GATA1* associates with and inhibits p53.”

6. The authors describe that known p53-pathway genes were enriched in TP53 multi-hit HSPCs compared to TP53 wild-type cells (Ext Data Fig. 4b). This is not apparent in the heatmap provided, as it is not clear if one of the quadrants show enriched expression of TP53 target genes. Can the authors provide a GSEA or a similar analysis on p53 target genes?

RESPONSE: We carried out an additional analysis which revealed that differentially expressed genes in TP53 multi-hit HSPC showed a highly significant ($p=2.03e-05$) overlap with canonical p53 target genes derived from meta-analysis (Fischer, *Oncogene*, 2017; DOI: 10.1038/onc.2016.502). This analysis is shown in Extended Data Fig.4c.

Minor critique:

1. Line 254. The authors indicate 6 CP-MPN patients but the figure and line 263 describe this as 4 CP-MPN patients.

RESPONSE: 6 patients profiled by bulk sequencing but only 4 by TARGET-seq, we have clarified this in the text (lines 218-220) as follows: “All 5 pre-TP53-sAML samples and 4 of the 6 CP TP53-MPN were then analysed by TARGET-seq (Fig.4a).”

Finally, as is routine for all manuscripts in our laboratory at the revision stage, we have also carried

out an extensive, independent and systematic recheck of all the data and analyses/scripts in the paper against the original raw data. This involved careful re-analysis of all data from raw data files, to further ensure the highest possible integrity of the data as well as their accessibility to the readers. As part of this process, we identified some minor errors which we have corrected, resulting in some changes in the numbers of cells included and gene lists. Importantly, this has not resulted in any changes to any of the findings described or conclusions reached in our manuscript, although the appearance of some of Figures has been slightly altered by this re-analysis without any meaningful difference in the data. We would obviously be happy to clarify specific details of this process and the resulting changes at your request.

Reviewer #4:

Remarks to the Author:

Rodriguez-Meira et al set out to understand how TP53 mutant clones transform from MPN to secondary AML. They used a single cell sequencing strategy that integrates transcriptomic and mutational data on cd34+ cells from these samples. Overall, these sample acquire successive hits to the p53 locus over time result in a TP53 hemizygous or biallelic mutant state upon transformation to sAML. Consistent with previous reports, TP53 secondary AML was associated with complex cytogenetic evolution. They find that TP53 mutant cells have higher erythroid-associated transcriptional program scores relative to P53 WT cells from these same samples. This was further associated with increased GATA1 expression, indicating that there may be enhanced erythroid priming. agnatures indicative of impaired differentiation, however they maintained functional differentiation potential in vitro. Next, they compare pre-transformation MPN->sAML samples to chronic phase MPN samples—both with p53 mutations.

These pre-sAML p53 mutated cells demonstrated upregulation of interferon response genes. In a murine model of IFN challenge p53 mutant cells have greater fitness than their WT counterparts. Collectively this suggests that IFN γ associated with inflammation may enable the outgrowth of p53 mutant clones.

Overall, this is an elegant manuscript with complementary single cell multiomic and functional studies.

RESPONSE: We are very grateful to the reviewer for their positive

comments. A few points that may strengthen the manuscripts are below:

1. Figure 2a. Does the p53 LSC signature simply pick out the p53 mutated cases from Beat-AML and TCGA, which would already be predicted to have inferior overall survival, or dose it also identify P53 WT cases that cluster with P53-mutant cases, and thus allow for improved stratification over P53 mutant status alone? The discussion states that this signature is predictive regardless of p53 status, but this is hard to appreciate when reading the results.

RESPONSE: We show that the p53LSC score predicts for adverse outcome in TP53 WT patients

from the BeatAML and TCGA datasets (Extended Data Fig.6d and e). To address the reviewer's point, we have now included an additional analysis (Extended Data Fig.6a,b) to show the distribution of p53-LSC score according to *TP53* mutation status in both BeatAML and TCGA cohorts. As expected, *TP53* mutant samples showed a high p53LSC score. However, some *TP53* wild-type AML samples also showed a high p53LSC score. We describe this new analysis in the results section on lines 160-162.

2. Figure 4c. The downregulation of TNFa/TGFb signatures is a bit hard to find in the supplemental tables, perhaps it would help the reader to at the NES value to the text. Figure 4c shows the NES for IFNgamma at 1.67, but not IFN alpha at 1.91, which is a bit more impressive. It might be good to show both, along with the associated GSEA plots as this is such a key point in the paper.

RESPONSE: We have now added a new panel c in Fig.4 showing the enriched and downregulated pathways (including TNFa/TGFb) along with the NES value.

3. How might higher IFN gene expression signatures in pre-TP53 AML cells correspond to increased survival under IFN challenge? One could imagine that an increase in IFN associated gene signatures could actually be an indication of vulnerability to inflammatory challenge. Some additional explanation might be helpful here.

RESPONSE: Our proposed model is that a variety of inflammatory stimuli can promote p53 mutant expansion. Exit from dormancy induced by a variety of inflammatory cytokines, including interferon (also induced indirectly by poly(I:C)) and LPS, is known to cause DNA-damage-induced attrition in HSCs (e.g. Walter et al, Nature 2015; DOI: 10.1038/nature14131). We therefore propose that *TP53* mutation rescues HSCs that would otherwise undergo DNA-damage-induced attrition, ultimately leading to the accumulation of HSCs which have acquired DNA damage, thus promoting genetic evolution that underlies disease progression. In order to test this possibility, we established an inducible *Trp53^{R172H/+}* model (Figure 5h) and demonstrated that *Trp53^{R172H/+}* LSK cells are resistant to inflammation induced apoptosis in comparison with the expected increase in apoptosis we observed in wild-type counterparts (Pietras et al, JEM, 2014; DOI: 10.1084/jem.20131043), whereas the cell cycle is increased by poly(I:C) in both WT and *Trp53* mutated cells (described in the results section lines 269-274 and new Fig.5l and 5m). To confirm that this ultimately leads to genetic evolution of *Trp53* mutated cells (gain of karyotypic aberrations), we carried out M-FISH karyotype analysis of CD45.2+ *Trp53^{+/+}* LSK cells expanded *in vitro* from mice following poly(I:C) treatment and CD45.1+ *Trp53^{R172H/+}* LSK cells from mice with or without poly(I:C) treatment. We observed a striking increase in karyotypic abnormalities in *Trp53* mutated LSK cells upon poly(I:C) treatment, including frequent acquisition of complex abnormalities (3 or more aberrations per cell) compared to both their poly(I:C) treated WT counterparts and *Trp53* mutated LSK cells without poly(I:C) treatment. Collectively, these data suggest that leukaemic progression under chronic inflammation is mediated both by increased cell number (through increased cell cycling with resistance to apoptosis in *Trp53* mutant cells) and increased genetic aberrations as observed in patients and in the mouse models. These new data are presented in a new Figure (Fig.6a-d) and described in the results section on lines 276-

289. We would speculate that the increased interferon-associated gene expression observed in pre-*TP53*-sAML cells may represent survival of *TP53* mutant cells that would otherwise undergo apoptosis when exposed to chronic inflammation in the presence of wild-type p53, thereby enriching for cells showing upregulated interferon associated gene expression. We believe that these data address the reviewer's comment and strengthen the manuscript. To emphasize this point, we have added additional text in the discussion on lines 332-343: "We provide evidence that *TP53* mutant HSCs showing dysregulated inflammation-associated gene expression are enriched in patients who go on to develop *TP53*-sAML. We propose that HSC that would otherwise undergo inflammation-associated and DNA-damage-induced attrition, are rescued by *TP53* mutation, ultimately leading to the accumulation of HSCs which have acquired DNA damage, thus promoting genetic evolution that underlies disease progression. Further studies are required to characterize this, and also the key inflammatory mediators and molecular mechanisms involved, which we believe are unlikely to be restricted to a single axis, with a myriad of inflammatory mediators overexpressed in MPN."

A general note for *the in vivo* experiments. We performed the experiments in 2 laboratories with different mouse colonies (in Oxford and Paris), and we consequently used 2 different models for the induction of *Trp53* mutation: the Vav-iCre system presented at the 1st submission (Fig5.a) and the SCL-Cre-ER^T system we used to perform some additional experiments to address the reviewers' comments (Fig.5h). We clarify in each Figure which model was used for the presented experiment and, importantly, we confirmed selection of the *Trp53* mutated cells upon poly(I:C) treatment in the SCL-Cre-ER^T system (Extended Data Fig.10k,m).

4. Is there any potential connection between erythroid bias and increased fitness under IFN γ challenge? Relatedly might GATA1 or CEBPA be known to participate in any of these response processes?

RESPONSE: Chronic inflammation is known to disrupt erythroid differentiation. We observed an expected inflammation-induced anemia with chronic poly(I:C) treatment. We also performed a new phenotypic analysis of BM erythroid progenitors (preCFU-E and CFU-E : Lin⁻Sca-1⁻c-Kit⁺CD41⁻FcgRII/III⁻CD105⁺) after chronic poly(I:C) treatment. Numbers of wild-type competitor erythroid progenitors were reduced upon poly(I:C) treatment (as expected), whereas *Trp53*-mutation was associated with an increase in erythroid progenitors that was not impacted by inflammation. These data are described on lines 265-269 and in the new Fig.5k. We would speculate that the relative levels of GATA1 and CEBPA, and known interactions with p53, might contribute to the erythroid-associated transcription we observed in *TP53*-sAML. With regards to GATA1, we already show that GATA1 expression is increased in *TP53* mutant cells (Fig.2g) and in *TP53* mutant AML cohorts (Ext. Data Fig.5i). Furthermore, we demonstrate an altered CEBPA/GATA1 ratio in association with *TP53* mutation (Fig.2g and h). We have made an additional comment in the discussion on lines 310-312 to emphasize the potential role of this interaction between CEBPA, GATA1 and p53: "Notably, *CEBPA* knockout or mutation is reported to cause a myeloid to erythroid lineage switch with increased expression of *GATA1* and, in addition, *GATA1* associates with and inhibits p53."

Minor

-Line 50, a very brief explanation of the TARGET seq methodology would be helpful to the reader here.

RESPONSE: We have added additional text as requested (lines 54-55), including relevant references which describe the method in detail.

A little interpretation about TNFa being down but IFN up with respect to inflammatory processes would be helpful to the reader.

RESPONSE: Relating to interferon signatures, as described in our response to point 3, we propose the *TP53* mutation protects the HSC from DNA-damage-induced attrition in association with inflammation-induced proliferation. In line with this, TNFa (and TGFb) are both cytokines that are associated with attrition of HSC through apoptosis, we have now included relevant references in this regard (results section, lines 230-231).

Finally, as is routine for all manuscripts in our laboratory at the revision stage, we have also carried out an extensive, independent and systematic recheck of all the data and analyses/scripts in the paper against the original raw data. This involved careful re-analysis of all data from raw data files, to further ensure the highest possible integrity of the data as well as their accessibility to the readers. As part of this process, we identified some minor errors which we have corrected, resulting in some changes in the numbers of cells included and gene lists. Importantly, this has not resulted in any changes to any of the findings described or conclusions reached in our manuscript, although the appearance of some of Figures has been slightly altered by this re-analysis without any meaningful difference in the data. We would obviously be happy to clarify specific details of this process and the resulting changes at your request.

Decision Letter, first revision:

17th Apr 2023

Dear Adam,

Thank you for submitting your revised manuscript "Deciphering TP53 mutant Cancer Evolution with Single-Cell Multi-Omics" (NG-A59372R1). It has now been seen by the original referees and their comments are below. The reviewers find that the paper has improved in revision, and therefore we'll be happy in principle to publish it in Nature Genetics, pending minor revisions to satisfy the referees' final requests and to comply with our editorial and formatting guidelines.

We are now performing detailed checks on your paper and will send you a checklist detailing our editorial and formatting requirements soon. Please do not upload the final materials and make any

revisions until you receive this additional information from us.

Sincerely,

Michael Fletcher, PhD
Senior Editor, Nature Genetics

ORCID: 0000-0003-1589-7087

Reviewer #1 (Remarks to the Author):

I would like to congratulate the authors on the revision of their manuscript which addressed all my (minor) concerns. These questions were answered extensively by adding new experimental data (comparing SCL-Cre to Mx1Cre) which allowed to address the role of inflammatory stress on the selection of mutant TP53 MPN clones even better. The schematic (now Figure 6e) nicely summarized the findings of the study.

I do not have additional questions.

The authors provide high granularity on the clonal evolution of TP53 mutant clones in MPN on its progression to sAML. The functional validation is sound and provide a crucial conceptual advance in clonal evolution but also selection of TP53 mutant cells.

Reviewer #2 (Remarks to the Author):

The authors have adequately addressed my comments and those of other reviewers. In doing so they have provided robust evidence for their conclusions.

Reviewer #3 (Remarks to the Author):

The authors addressed all concerns this reviewer raised and put together a highly interesting, provocative, and well-executed study.

Reviewer #4 (Remarks to the Author):

Authors do a very comprehensive job responding to reviewer comments. They have satisfied all of my concerns. The new Figure 6 is super interesting and brings the whole story full circle mechanistically. I recommend publication.

Final Decision Letter:

20th Jul 2023

Dear Adam,

I am delighted to say that your manuscript "Single-Cell Multi-Omics Identifies Chronic Inflammation as a Driver of TP53 mutant Leukaemic Evolution" has been accepted for publication in an upcoming issue of Nature Genetics.

Your paper will be published online after we receive your corrections and will appear in print in the next available issue. You can find out your date of online publication by contacting the Nature Press Office (press@nature.com) after sending your e-proof corrections. Now is the time to inform your Public Relations or Press Office about your paper, as they might be interested in promoting its publication. This will allow them time to prepare an accurate and satisfactory press release. Include your manuscript tracking number (NG-A59372R2) and the name of the journal, which they will need when they contact our Press Office.

Please note that Nature Genetics is a Transformative Journal (TJ). Authors may publish their research with us through the traditional subscription access route or make their paper immediately

open access through payment of an article-processing charge (APC). Authors will not be required to make a final decision about access to their article until it has been accepted. [Find out more about Transformative Journals](https://www.springernature.com/gp/open-research/transformative-journals)

Authors may need to take specific actions to achieve [compliance with funder and institutional open access mandates](https://www.springernature.com/gp/open-research/funding/policy-compliance-faqs). If your research is supported by a funder that requires immediate open access (e.g. according to [Plan S principles](https://www.springernature.com/gp/open-research/plan-s-compliance)) then you should select the gold OA route, and we will direct you to the compliant route where possible. For authors selecting the subscription publication route, the journal's standard licensing terms will need to be accepted, including [those licensing terms will supersede any other terms that the author or any third party may assert apply to any version of the manuscript](https://www.nature.com/nature-portfolio/editorial-policies/self-archiving-and-license-to-publish).

Please note that Nature Portfolio offers an immediate open access option only for papers that were first submitted after 1 January, 2021.

If you have not already done so, we invite you to upload the step-by-step protocols used in this manuscript to the Protocols Exchange, part of our on-line web resource, natureprotocols.com. If you complete the upload by the time you receive your manuscript proofs, we can insert links in your article that lead directly to the protocol details. Your protocol will be made freely available upon publication of your paper. By participating in natureprotocols.com, you are enabling researchers to more readily

reproduce or adapt the methodology you use. Natureprotocols.com is fully searchable, providing your protocols and paper with increased utility and visibility. Please submit your protocol to <https://protocolexchange.researchsquare.com/>. After entering your nature.com username and password you will need to enter your manuscript number (NG-A59372R2). Further information can be found at <https://www.nature.com/nature-portfolio/editorial-policies/reporting-standards#protocols>

Sincerely,

Michael Fletcher, PhD
Senior Editor, Nature Genetics

ORCID: 0000-0003-1589-7087